# Pyrite-lined shells as indicators of inefficient bioirrigation in the Holocene-Anthropocene stratigraphic record

Adam Tomašových[1], Michaela Berensmeier[2], Ivo Gallmetzer[2], Alexandra Haselmair[2], Martin Zuschin[2]

[1]Earth Science Institute, Slovak Academy of Sciences, Bratislava, 84005, Slovakia
[2]University of Vienna, Department of Palaeontology, Althanstrasse 14, 1090 Vienna, Austria

**Correspondence to:** Adam Tomašových (geoltoma@savba.sk)

**Abstract.** Although the depth of bioturbation can be estimated on the basis of ichnofabric, the time scale of sediment mixing (reworking) and irrigation (ventilation) by burrowers that affects carbonate preservation and biogeochemical cycles is difficult to estimate in the stratigraphic record. However, pyrite linings on the interior of shells can be a signature of slow and shallow irrigation. They indicate that shells of molluscs initially inhabiting oxic sediment pockets were immediately and permanently sequestered in reduced, iron-rich microenvironments within the mixed layer. Molluscan biomass stimulated sulfate reduction and pyrite precipitation was confined to the location of decay under such conditions. A high abundance of pyrite-lined shells in the stratigraphic record can thus be diagnostic of limited exposure of organic tissues to $O_2$ even when the seafloor is inhabited by abundant infauna disrupting and age-homogenizing sedimentary fabric as in the present-day northern Adriatic Sea. Here, we reconstruct this sequestration pathway characterized by slow irrigation (1) by assessing preservation and postmortem ages of pyrite-lined shells of the shallow-infaunal and hypoxia-tolerant bivalve *Varicorbula gibba* in sediment cores and (2) by evaluating whether an independently-documented decline in the depth of mixing, driven by high frequency of seasonal hypoxia during the 20th century, affected the frequency of pyrite-lined shells in the stratigraphic record of the northern Adriatic Sea. First, at prodelta sites with high sedimentation rate, linings of pyrite framboids form rapidly in the upper 5-10 cm as they appear already in interiors of shells younger than 10 years and occur preferentially in well-preserved and articulated shells with periostracum. Second, increments deposited in the early 20th century contain <20% of shells lined with pyrite at the Po prodelta and 30-40% at the Isonzo prodelta, whereas the late 20th century increments possess 50-80% of shells lined with pyrite at both locations. At sites with slow sedimentation rate, the frequency of pyrite linings is low (<10-20%). Surface sediments remained well mixed by deposit and detritus feeders even in the late 20th century, thus maintaining the suboxic zone with dissolved iron. The upcore increase in the frequency of pyrite-lined shells thus indicates that the oxycline depth was reduced and bioirrigation rates declined during the 20th century. We hypothesize that the permanent preservation of pyrite linings within the shells of *V. gibba* in the subsurface stratigraphic record was enabled by slow recovery of infaunal communities from seasonal hypoxic events, leading to the dominance of surficial sediment modifiers with low irrigation potential. The presence of very young and well-preserved pyrite-lined valves in the uppermost zones of the mixed layer indicates that rapid obrution by episodic sediment deposition is not needed for preservation of pyrite linings when sediment irrigation is impersistent and background sedimentation rates are not low (here, exceeding ~0.1 cm/y) and infaunal organisms die at their living position within the sediment. Abundance of well-preserved shells lined by pyrite exceeding ~10% per assemblage in apparently well-mixed sediments in the deep-time stratigraphic record can be an indicator of inefficient bioirrigation. Fine-grained prodelta sediments in the northern Adriatic Sea deposited since the mid-20th century, with high preservation potential of reduced microenvironments formed within a mixed layer, can represent taphonomic and early-diagenetic analogues of deep-time skeletal assemblages with pyrite linings.

## 1. Introduction

Slow and shallow bioturbation (including biomixing and bioirrigation, Kristensen et al., 2012; Wrede et al., 2018) can reflect oxygen depletion, toxicity or other environmental stresses that limit ecosystem functioning and nutrient recycling (Rhoads and Germano, 1986; Nilsson and Rosenberg, 2000; Rosenberg et al., 2001; Solan and Kennedy, 2002). It can be also conditioned by evolutionary factors such as the lack of innovations for deep burrowing (Thayer, 1983) or by extinction of burrowers after mass extinctions (Pruss et al., 2004; Buatois and Mangano, 2011). Although biomixing (reworking of sedimentary particles) and bioirrigation (exchanges of pore water and solutes by burrow ventilation) can be decoupled to some degree (Woodin et al., 2010), estimating the mixing rate and depth on the basis of the stratigraphic record can be used to assess the response and recovery of benthic ecosystems to stress and disturbances and to evaluate long-term trends in ecosystem functioning and biogeochemical cycling (Droser et al., 2002; Canfield and Farquhar, 2009; Tarhan et al., 2015; Gougeon et al., 2018; Buatois et al., 2020). The mixing rate, the mixing depth, and the sedimentation rate represent three parameters that determine the residence time of sedimentary and organic particles in the mixed layer (Meysman et al., 2003). Assuming that biomixing positively relates to bioirrigation, this residence time determines preservation, recycling and burial efficiency of organic matter, carbonates and redox-sensitive minerals because it also controls their exposure time to $O_2$ in near-surface sediment zones (Aller, 1982, 1994; Hartnett et al., 1998; Meile and van Cappellen, 2005; Boyle et al., 2014; Aller and Cochran, 2019) and to borers, microbes and undersaturated pore waters in the so-called taphonomic active zone (TAZ; Davies et al., 1989; Walker and Goldstein, 1999). If sedimentation rate does not change over a given duration of deposition, dividing the depth of the mixed layer by sedimentation rate can be used to estimate the residence time of particles in the mixed layer, and thus their potential exposure to bioirrigation (Wheatcroft, 1990; Tomašových et al., 2019a). However, first, exposure time of particles to $O_2$ is not a simple function of their residence time in the mixed layer and also depends on the size and feeding mode of burrowers and on the persistence and magnitude of their pumping rate (Sandnes et al., 2000; Lohrer et al., 2004; Gingras et al., 2008; Volkenborn et al., 2012; Norkko et al., 2013; Kristensen et al., 2018; Renz et al., 2018; Wrede et al. 2018). For example, deposit feeders mainly promote mixing whereas suspension-feeders and chemosymbionts mainly induce irrigation (Christensen et al., 2000; Michaud et al., 2005). Second, long-term estimates of net sedimentation rate cannot be interpolated to shorter time scales (Jerolmack and Sadler, 2007) and the time scale of mixed-layer formation is similarly confounded by internal temporal variability in bioturbation over shorter time scales (Teal et al., 2008). For example, a 20-cm thick layer deposited under 1 cm/y may be mixed (or irrigated) permanently over 20 years or may remain unmixed for 19 years and then instantaneously mixed just in a single year. Both the instantaneous mixing and the slower mixing will generate an age-homogeneous distribution of sedimentary particles but will differ in the duration of mixing and in recycling and burial efficiency of redox-sensitive sedimentary particles. Therefore, although the depth of mixing in the stratigraphic record can be estimated on the basis of ichnofabric and trace fossils (Bromley and Ekdale, 1986; Droser and Bottjer, 1988; Savrda and Ozalas, 1993), it is unclear (i) whether mixing leading to age homogenization of sedimentary particles and to the loss of sedimentary fabric was also associated with efficient irrigation and (ii) whether the ichnofabric characterizing a mixed layer in the stratigraphic record developed over yearly, decadal or longer time scales. This uncertainty differs from temporally-explicit estimates of penetration of sediment by $O_2$ and other solute tracers, from estimates of apparent redox potential discontinuity, or from the estimates of the mixed layer thickness based on [234]Th that can be measured in present-day environments (Andersson et al., 2006; Maire et al., 2008; Germano et al., 2011; Gerwing et al., 2018; Solan et al., 2019; Borger et al., 2020).

One criterion that can be used to constrain the exposure of particles to $O_2$ in the mixed layer in the *ancient* stratigraphic record includes the detection of rapid authigenic mineralization associated with the decay of organic tissues in the absence of $O_2$ (Allison, 1988; Briggs et al., 1991, 1996). The lack of bioturbation that limits $O_2$ exposure and allows

preservation of intact, articulated, multi-element skeletal remains under anaerobic degradation of organic matter can induce early pyritization, phosphatization or silicification of organic tissues (Gabbott et al., 2004; Zhu et al., 2005; Cai et al., 2012; Saleh et al., 2019; 2020; Schiffbauer et al., 2014; Novek et al., 2016). Early pyritization or silicification induced by reactive organics may also generate death masks that stabilize and cement sediment and blanket benthic organisms (Gehling, 1999; Strang et al., 2016; Tarhan et al., 2016; Gibson et al., 2018; Liu et al., 2019; Slagter et al., 2021). Similar pathways stimulated by the decay of labile organics or further sustained by microbial and mucus-forming processes in anaerobic microenvironments (microniches or pockets; Emery and Rittenberg, 1952; Jorgensen, 1977; Borkow and Babcock, 2003; Stockdale et al., 2010; Anderson et al., 2011; Virtasalo et al., 2010, 2013; Lehto et al., 2014) can lead to distinctive clusters of authigenic minerals concentrated in intra-skeletal voids initially filled with organic tissues in otherwise well-preserved skeletal remains (Brett et al., 2012a, b). These authigenic minerals do not replace soft tissues but form linings or infills located at the sediment-skeletal boundaries in buried fossil assemblages (Gaines et al., 2008). For example, intraskeletal pores (initially containing organic tissues and microbes associated with their decay) lined by framboids or larger pyrite crystals preserved in the stratigraphic record, as observed in interiors of foraminifers, cephalopods, articulated brachiopods or bivalves, indicate that reduced microenvironments within shells were not re-exposed to $O_2$ by irrigation or exhumed to the sediment-water interface (Hudson, 1982; Schieber and Baird, 2001; Hunda et al., 2006; Schieber, 2012; Machain-Castillo et al., 2019). Such microenvironments thus remained permanently sequestered after their initial formation and can be indicative of spatially- and temporally-limited bioirrigation, leading to a shallow depth of the oxycline. However, with the exception of seep environments (Powell et al. 2012) or studies focusing on pyrite infills in burrows (Virtasalo et al., 2010; Gingras et al., 2014; Harazim et al., 2015; Kędzierski et al., 2015; Baucon et al., 2020), pyrite-lined skeletal remains are rarely observed in actualistic studies assessing preservation of organisms with durable skeletal elements in Holocene or Anthropocene marine environments (Best and Kidwell, 2000). This rarity may not be surprising because the preservation of pyrite framboids nucleating at the sites of organic decay (initiated in reduced microenvironments) is limited in habitats with a well-irrigated mixed layer where most of the sulfide produced by sulfate reducers in microenvironments is rapidly reoxidized (Canfield et al., 1993; Wijsman et al., 2002; Canfield and Farquhar, 2009). However, taphonomic and early-diagenetic pathways that affect skeletal remains remain poorly explored in environments affected by eutrophication, stratification and oxygen depletion that tend to be characterized by limited bioirrigation and by high burial efficiency of iron (Carstensen et al., 2014; Kristensen et al., 2014; Lenstra et al., 2019).

The decline in the functioning of marine soft-bottom benthic ecosystems driven by eutrophication and deoxygenation over the past centuries (Schaffner et al., 1992; Valente and Cuomo, 2005; Petersen et al., 2008; Zillén et al., 2008) and affected by changes in biogeochemical cycling (Villnäs et al., 2012; Kristensen et al., 2014; Jokinen et al., 2018) thus represents a unique opportunity (1) to assess the nature of pathways that lead to the preservation of pyrite-lined shells (especially the steps associated with the initial formation of reducing, iron-rich microenvironments and with the subsequent lack of their reoxidation by irrigation and/or by exhumation to the sediment-water interface) and thus can provide taphonomic and early-diagenetic analogues for pyrite-lined shells in the deep-time stratigraphic record, and (2) to test whether the general decline in bioturbation and the associated shift in the composition of soft-bottom benthic communities detected in sediment cores over the past decades influenced the frequency of pyrite-lined shells in the stratigraphic record. On one hand, the decline in activities of infauna can limit bioirrigation, thus reducing the penetration of sediment by $O_2$ (with more frequent formation of reducing microenvironments where pyrite framboids can precipitate) and decreasing the potential of pyrite framboids to be reoxidized. On the other hand, the decline in activities of infauna can lead to the decline in sediment mixing, thus reducing an iron-based redox shuttle and limiting the development of the suboxic zone rich in dissolved iron that is necessary for confinement of $H_2S$ to the location of carcass decay (Canfield et al., 1993; van de Velde and Meysman 2016; Beam et al., 2018). Here, we focus on Holocene-Anthropocene sediment cores from the northern

Adriatic Sea which harbors diverse types of infaunal and epifaunal benthic communities (Fedra et al., 1976; Zuschin et al., 1999; Zuschin and Stachowitsch, 2009). This sea was affected by eutrophication and by an increase in frequency of seasonal hypoxia in the 20[th] century (Justic, 1991; Degobbis et al., 2000), with frequent mucilage events (Cozzi et al., 2004; Giani et al., 2005; Precali et al., 2005) and several spatially-extensive seasonal anoxic events inducing mass mortality of benthic fauna in the 70s-90s (Stefanon and Boldrin, 1982; Faganeli et al., 1985; Stachowitsch, 1991). Oligotrophication was detected in the northern Adriatic Sea since 1990s (Mozetič et al. 2010; Djakovac et al., 2012, 2015), although habitats close to Po prodelta were regularly affected by seasonal hypoxia during the 1977-2008 monitoring (Alvisi and Cozzi, 2016) and hypoxic events were detected in the Gulf of Trieste also in the early 21[st] century (Kralj et al., 2019). Paleoecological records in sediment cores collected in the Po prodelta and in the Gulf of Trieste showed that the effects of increasing frequency of seasonal oxygen depletion (and other changes related to pollution, sediment turbidity and trawling) translated into a 20[th] century regime shift in the composition of soft-bottom molluscan communities (at 10-30 m water depths, Tomašových et al., 2020). Although this shift was characterized by a decline in the depth of the mixed layer during the 20[th] century, as indicated by an upcore increase in preservation of flood layers and in temporal resolution of molluscan assemblages in sediment cores (Tomašových et al., 2018), the uppermost centimetres of soft-bottom sediments remain relatively well-mixed by burrowers during the late 20[th] century.

Pyrite can precipitate rapidly at the location of decay of tissues of hosts or at the location of intra-skeletal organic matrices among crystals (we call such pyrite *primary linings*, Boekschoten, 1966; Brown, 1966) but can also form at the location of decay of secondary inhabitants of shells that were exposed to redox fluctuations in the TAZ for longer (coelobionts, encrusters, or borers; *secondary linings*, Kobluk and Risk, 1977) or can precipitate in skeletal pores as disseminated nanopyritic inclusions (Brand and Morrison, 1987), leading to darkened and stained valves (Tomašových et al., 2021). However, only primary linings represent a unique evidence of limited postmortem exposure time of such shells to $O_2$ from irrigation (with pyrite framboids confined to the location of carcass decay) prior to their transit below the mixed layer into the final stratigraphic record. Here, to assess whether abundant pyrite-lined shells of the hypoxia-tolerant bivalve *Varicorbula gibba* are informative about bioirrigation and to distinguish primary from secondary linings (and from other types of pyrite associated with skeletal remains that forms over longer time scales), we evaluate (1) the rate of formation of pyrite linings and the depth of their formation (in surface or subsurface sediment zones) on the basis of postmortem age- and depth-frequency distributions of dated specimens of *V. gibba* in sediment cores, (2) whether pyrite-lined valves possess more frequently a periostracum and higher shell organic content than valves without pyrite (controlling for shell postmortem age), (3) whether the increment-level frequency of pyrite-lined valves correlates negatively with other types of alteration such as disarticulation or bioerosion, (4) whether the frequencies of pyrite-lined valves change upcore within the Holocene-Anthropocene stratigraphic record, and whether they track independently-documented changes in community composition, using abundance and shell size of the hypoxia-tolerant bivalve *V. gibba*.

## 2. Setting

The northern Adriatic Sea is a relatively low-energy and low-relief, ramp-like shelf environment with narrow tidal range, with counterclockwise surface circulation, mainly driven by the freshwater input of the River Po (Kemp et al., 1999; Boldrin et al., 2009; Brush et al., 2021). The cyclonic circulation normally deflects nutrient-rich water masses off the Po Delta southwards. However, this circulation slows down during the summer when a local cyclonic gyre can spread such eutrophic waters also towards the eastern Adriatic shelf normally characterized by oligotrophic surface conditions. The decline in circulation leads to the development of a pycnocline, and in combination with high phytoplankton production, to the

accumulation of marine snow and mucilages and to bottom-water oxygen depletion in some years across large parts of the northern Adriatic Sea (Stachowitsch, 1991; Penna et al., 1993; Manini et al., 2000; Cozzi et al., 2004; Precali et al., 2005). Sediments associated with silt and clay-rich sediments are relatively rich in iron in deltaic environments (3-4%, Spagnoli et al., 2014) and less so in carbonate sediments off Istria (>1.5-3% in the Gulf of Trieste, Faganeli et al., 1991; Faganeli et al., 1994; Dolenec et al., 1998). Iron is bound mainly by iron oxides or by pyrite (as framboids or foram infillings). Pyrite forms

2-5% of the sediments in the Gulf of Trieste (Faganeli and Ogrinc, 2009), and typically less than 1% in Holocene cores on the Po Plain (Amorosi et al., 2002). Biogeochemical studies performed in the 1990s showed that anaerobic decomposition of organics (mainly sulfate reduction) is a dominant pathway in organic-rich sediments at the Po prodelta (Barbanti et al., 1995). In contrast, less-organic rich sites in the NW Adriatic Sea with slower sedimentation display a higher importance of aerobic respiration and iron reduction (Barbanti et al., 1995; Hammond et al., 1999). This gradient in sedimentation rate

leads to lower recycling of organic carbon at the Po prodelta and higher recycling at sites distal relative to river-borne sediments. A similar situation was also observed in the Gulf of Trieste, with higher burial efficiency of organic carbon and nitrogen at muddy sites with high sedimentation rate at the Isonzo prodelta and with higher recycling at sandy sites with slow sedimentation rate in the central and southern parts of the Gulf of Trieste (with iron bound with oxides rather than with sulfides, Hines et al., 1997; Arcon et al., 1999; Faganeli and Ogrinc, 2009).


## 3. Methods

### 3.1 Sediment cores

Sediment cores collected at five sites in the northern Adriatic Sea (Fig. 1) described and geochronologically-dated in our former papers were used to assess stratigraphic changes in the proportion of specimens lined with pyrite. Two sites are

located at the Po prodelta (Po 3 and Po 4 at 21 m, Tomašových et al., 2018), one site in the northern Gulf of Trieste, in the Bay of Panzano at 12 m at the Isonzo prodelta (Gallmetzer et al., 2017; Tomašových et al., 2017), one site in the southern Gulf of Trieste off Piran at 23 m (Mautner et al., 2018; Tomašových et al., 2019b), and one site on the southern tip of Istria at Brijuni at 44 m (Schnedl et al., 2018). Sediment cores collected in 2013 are ~1.5 m long (Gallmetzer et al., 2016) and were initially split into 2 cm-thick increments in the upper 20 cm and into 5 cm-thick increments below. The core diameter was 9

cm at the site Brijuni and 16 cm at sites Po, Panzano, and Piran. All increments were sieved with 1 mm and, with some exceptions (at Piran), all individuals of *Varicorbula gibba* were picked from residues. At Piran, the increments between 10 to 35 cm were split into quarters and increments between 35 to 45 cm were split to half owing to high shell abundance (Gallmetzer et al., 2019). Van Veen grabs with surface death assemblages were collected in 2014 (Haselmair et al., 2021). Core geochronology, the estimates of sedimentation rates, and the depth of the mixed layer are based (i) on the profiles in the

excess activity of [210]Pb and (ii) on the stratigraphic distribution of median bivalve shell ages based on amino acid racemization (AAR) calibrated by [14]C published formerly in studies devoted to individual sites (at Panzano in Tomašových et al., 2017, at Po in Tomašových et al., 2018, at Piran in Mautner et al., 2018 and Tomašových et al., 2019b, and at Brijuni in Schnedl et al., 2018 and Gallmetzer et al., 2019).

Although these sites are located at similar water depths, they differ in their proximity to clastic input from rivers,

grain size, net sedimentation rate, sediment organic enrichment and iron content. Stratigraphic profiles in grain size and in concentrations of $CaCO_3$, TOC and TN, and Fe concentrations (in 63 μm sediment fraction, documented in Gallmetzer et al., 2017; Mautner et al., 2018; Schnedl et al., 2018) show that the three prodelta sites are exclusively muddy and rich in iron (~3%) whereas the two locations off Istria are sandy, iron-poor (1-2%), and rich in carbonate skeletal material (molluscs,

echinoderms, bryozoans, coralline algae) (Fig. 2). The two Po cores are characterized by high sedimentation rate (~1-2 cm/y), the Panzano core was deposited under intermediate sedimentation rate (~0.2-0.5 cm/y, occassionally up to 1 cm/y), and the Piran and Brijuni sites are sediment-starved (~0.01-0.03 cm/y). At Piran a 25 cm-thick skeletal shell bed occurs at 8 cm below the seafloor (Tomašových et al., 2019b) and at Brijuni a 20 cm-thick sandy mud in the core-top overlies a coarse bryozoan-rich molluscan muddy sand (Tomašových et al., 2021). Upcore changes in median shell age show that sedimentation rates moderately oscillated through time and do not show any increase in the uppermost levels that correspond to the late 20[th] century. The within-site variability in sedimentation rates is smaller than the marked variability among sites. Sedimentation rates fluctuated between ~1 and 2.4 cm/y during the 20[th] century at the Po sites (Tomašových et al., 2018) and between ~0.2-1 cm/y over the past 500 years at Panzano (Gallmetzer et al., 2017; Tomašových et al., 2017). Sedimentation rates at Piran and Brijuni were persistently one or two orders of magnitude lower both during the transgressive and highstand phase (~0.01-0.03 cm/y) than at Po and Panzano (Tomašových et al., 2019b, 2021), and we thus refer to Po and Panzano as sites with high and to Piran and Brijuni as sites with low sedimentation rate. The estimates of sedimentation rates based on the slope of the $^{210}$Pb profiles below the mixed layer are similar to those based on downcore changes in median shell ages at both Po sites and at Panzano (Fig. 3). However, the $^{210}$Pb segments located below the fully-mixed layer at Piran and Brijuni are still steepened by biomixing (and thus overestimate sedimentation rates), as is typical of conditions when the rate of biomixing exceeds the rate of sedimentation (Johannessen and Macdonald 2012). The thickness of the surface well-mixed layer, based on homogeneity of median per-increment shell ages (amino acid racemization calibrated by $^{14}$C, Fig. 3A) is 20 cm at Po and at Brijuni, 5 cm at Panzano, and 8 cm at Piran (where a coarse skeletal shell bed occurs at 8-35 cm below the seafloor). The cores at Piran and Brijuni can be subdivided to units deposited during the transgressive phase characterized by rapid increase in accommodation space (transgressive systems tract, TST), during the time of maximum ingression (maximum flooding zone, MFZ), and during the highstand phase characterized by very slow increase in accommodation space (highstand systems tract, HST, prior to the 20[th] century). The uppermost zones contain a mixture of late-highstand and 20[th] century sediments (Fig. 2, Mautner et al., 2018; Schnedl et al., 2018). The net sedimentation rates at Piran and Brijuni (affected by negligible clastic sediment input and by significant contribution of in situ heterozoan carbonate production) were very slow during the highstand phase and did not increase relative to the condition during the transgressive phase (i.e., prograding sediment bodies did not form during the highstand phase). The core at Panzano captures about 500 years and the 20[th] century sediments occur in the upper 35 cm. The cores at Po consist of sediments deposited during the early and the late 20[th] century (and the earliest 21[st] century), as described by Gallmetzer et al. (2017) and Tomašových et al. (2018, 2019b).

The thickness of the surface, well-mixed layer, based on the vertical extent of uniform or irregular segments of profiles in $^{210}$Pb excess is ~16 cm at Po, and 6 cm at Panzano, Piran, and Brijuni (Fig. 3B). With the exception of Brijuni, the estimates of the mixed-layer depths based the $^{210}$Pb profiles and the $^{14}$C-based shell age profiles are thus similar. At Brijuni, the 6 cm-thick mixed layer based on the $^{210}$Pb profiles relative to the 20 cm-thick mixed layer based on $^{14}$C probably reflects the shorter (multi-decadal) half-life of $^{210}$Pb relative to the longer time needed to mix the upper 20 cm of sediment and their bioclasts. Downcore increase in the mottling of sedimentary fabric and in time averaging of *V. gibba* assemblages in 5 cm increments further indicate that the mixed layer extended to more than ~20 cm at Po and Panzano prior to the late 20[th] century (Tomašových et al., 2018). These estimates of the depth of the mixed layer do not necessarily correspond to depths of instantaneous mixing. They apply to the depths that are fully-mixed at yearly and decadal scales at Po and Panzano but at millennial scales at Piran and Brijuni. Some incomplete mixing by burrowers also occurs below these depths, as suggested by the presence of very young shells at 100 cm at the Po sites (Tomašových et al., 2018).

We target specimens of the bivalve *V. gibba* because this species is common in all cores. *V. gibba* is a small-sized (<15 mm), shallow-infaunal, poorly mobile bivalve that has short siphons and thus lives in the upper 3 cm of sediment (Faresi et al.,

2012), with the posterior margin located at or slightly below the sediment-water interface. It feeds on suspended
        phytoplankton but also exploits benthic diatoms, bacteria and organic detritus at the sediment-water interface (Yonge 1946).
        This species has higher tolerance to reduced oxygen levels relative to other molluscs and invertebrate groups (Holmes and
        Miller, 2006; Riedel et al., 2012) and can survive for several days and even weeks in anoxic conditions (Christensen 1970).
        In the northern Adriatic Sea, it increases in abundance in the wake of short-term, seasonal anoxic or hypoxic events to more
than 1,000 individuals/m$^2$ (Hrs-Brenko 2006, Nerlović et al. 2011), and grows to 7-8 mm during the first year, achieving the
        maximum size of ~15 mm in two years (Hrs-Brenko 2003). It is classified as an opportunistic species in the assessments of
        benthic ecosystem health in the Mediterranean Sea (Simboura and Zenetos, 2002; Moraitis et al. 2018). Soft-bottom
        molluscan assemblages at water depths below the seasonal thermocline in the northern Adriatic became dominated by this
        species during the late 20$^{th}$ century Sea (Tomašových et al., 2018). The shells of this species consist of two unequal-sized
valves that are formed by an outer (with a cross-lamellar structure) and an inner (with a cross-lamellar and complex cross-
        lamellar structure) layer (Fig. 4A). Both valves possess one or more conspicuous internal, 10-100 μm-thick conchiolin layers
        (Fig. 4B-C), with embedded aragonite nodules (< 5 μm) (Lewy and Samtleben, 1979; Kardon, 1998). The internal conchiolin
        layer in the right valve terminates on the internal surface in a groove between the pallial line and the ventral margin (Fig. 4).
        The margin of the left valve is pressed against this groove so that the left valve is almost perpendicular relative to the right
valve, and the external periostracum of the left valve overlaps with the ventral external margin of the right valve. These traits
        allow a tight closure of articulated valves. The conchiolin in the internal layers is more refractory and insoluble (Krampitz et
        al., 1983) than labile organics initially lining the interior of valves. Its intrinsic attributes with (1) tightly-closed valves with
        overlapping periostracum and a small internal ligament (Yonge, 1946), reducing the potential for postmortem disarticulation,
        and (2) a conchiolin layer located within valves can favor the formation of reduced microenvironments both within shells
and within individual valves (Fig. 4).

### 3.2 Age distributions of pyrite-lined valves

        Ratios of D- and L-isomers (D/L) of eight amino acids (aspartic, glutamic, serine, alanine, valine, phenylalanine, isoleucine,
        and leucine), and their concentrations in valves of *V. gibba* were measured at Northern Arizona University using reverse-
phase high-pressure liquid chromatography (RP-HPLC) and the procedures of Kaufman and Manley (1998). D/L ratios
        measure the extent of razemization and thus represent a geochronological tool (Kosnik et al., 2008; Allen et al., 2013). These
        data were measured in 252 valves from Po 3, in 243 valves from Po 4, in 311 valves from Panzano, and in 232 valves from
        Piran, and were presented with age calibrations of aspartic acid D/L by $^{14}$C-dated valves by Tomašových et al. (2017, 2018)
        and Mautner et al. (2018). To constrain the rate and the location of pyrite formation on the basis of postmortem age data, we
evaluated (1) whether valves of the same postmortem age (binned to 10 years at Po and to 50 years at Panzano), with and
        without pyrite linings, differ in their mean stratigraphic depth or whether pyrite-lined valves are located deeper, (2) whether
        the valves with and without pyrite linings differ in their burial rate below the present-day mixed layer (with pyrite-lined
        valves expected to be buried earlier to deeper zones with reducing conditions), and (3) whether valves of the same age with
        and without pyrite linings differ in their total content of amino acids. We assess the differences in depth and age between
valves with and without pyrite by comparing their mean values and 95% bootstrapped confidence intervals. We estimate
        burial rates of valves below the mixed layer by fitting age distributions of valves with and without pyrite linings from Po and
        Panzano to two models that assume that the input of dead shells to the death assemblage is constant during their residence in
        the mixed layer (and thus over the duration of time averaging). The parameters estimated by these models are related to the
        burial rate and depend on the steepness of age distributions in the mixed layer. However, they are also determined by the
disintegration rate within the mixed layer (Tomašových et al., 2014). First, a simple model with temporally-constant loss rate

of valves ($\lambda$) from the mixed layer (with loss occurring by disintegration and/or burial) predicts that age distributions can be well-fitted by the exponential distribution (disintegration-burial model). Second, a more complex (sequestration) model where disintegration rate declines from $\lambda_1$ to $\lambda_2$ at some sequestration rate $\tau$ predicts that the resulting age distributions are heavy-tailed, typically owing to exhumation of older valves to the sediment surface, with $\lambda_1$ corresponding to the

disintegration rate of young valves, $\lambda_2$ corresponding to the reduced disintegration rate of older valves, and $\tau$ to the sequestration rate that can correspond to the burial rate (Tomašových et al., 2014). The Akaike Information Criterion corrected for small sample size shows similar support for these two models at Po and Panzano both in valves with and without pyrite linings, with either a lower AICs for the simple model or a slightly higher support for the sequestration model that does not exceed 2-3 units, and small differences between $\lambda_1$ and $\lambda_2$ in the sequestration model (Table S1). The $\lambda$

parameter estimated by the simple model at Po and at Panzano is similar both to [210]Pb-based estimates and to [14]C shell-based estimates of the sedimentation rate (1-2 cm/y at Po and 0.2 cm at Panzano). Therefore, we infer that this $\lambda$ parameter at Po and Panzano corresponds to burial rates and can be used to compare burial rates of valves with and without pyrite linings.

### 3.3 Taphonomic scoring

To assess the nature and types of pyrite linings, we evaluated preservation of all specimens of *V. gibba* at light-microscope

scale at 10-20x magnification. We investigated several specimens with scanning electron microscope (SEM) and with backscattered electrons (BSE), using electron probe microanalyzer at 100-1,000x magnification. The chemical composition of pyrite and iron oxides was validated with energy-dispersive X-ray spectroscopy. All valves of *V. gibba* that were dated by amino-acid racemization at four sites (Po 3, Po 4, Panzano M28, and Piran M53) were scored. At Po 3 (replicate cores M12, M13 and M14), Po 4 (cores M20 and M21), Panzano (cores M28 and M29), all additional specimens were scored under a

light microscope. At Brijuni, all specimens of *V. gibba* from every second increment were scored. Nine alteration variables were scored on specimens of *V. gibba* at light-microscope magnification (10-20x): (1) presence of black pyrite framboids that form strings, clusters or continuous coatings on internal valve surfaces, (2) loss of periostracum and/or of the internal conchiolin layer (if either external periostracum and/or internal conchiolin layer are visible, periostracum is scored as being preserved), (3) disarticulation, (4) internal fine-scale surface dissolution (shallow pits), (5) internal bioerosion (generated by

algae and sponges, excluding predatory drilling), (6) external encrustation, (7) intense surface wear on external and internal valve surfaces (extensive pitting and wear, leading to loss of external ornamentation), and (8) penetrative blueish or dark gray staining (induced by nanopyritic inclusions that fill valve microporosity). All data are available in the Supplement.

### 3.4 Geographic and stratigraphic differences in the frequency of pyrite-lined valves

To evaluate stratigraphic and geographic changes in preservation of *V. gibba* at the cm-scale of stratigraphic increments, we computed relative frequencies of specimens with a given alteration relative to the total number of specimens in 4 cm (pooling 2 cm increments in the upper 20 cm in each core) and 5 cm-thick increments at five sites. The frequency of disarticulated shells is computed as the number of all articulated specimens in a given increment divided by the minimum number of individuals (i.e., the sum of articulated shells plus the higher number of left or right valves). Although the

estimates of disarticulation are biased upward owing to disarticulation that takes place during sieving, we assume that this bias affects all increments equally. All increments were plotted and analyzed in stratigraphic analyses. However, the downcore stratigraphic profiles reflect (1) downcore changes in early-diagenetic conditions within the mixed layer driven by the depth of the oxycline and by the depth of iron and sulfate reduction and (2) chronological changes in biogeochemical processes (e.g., driven by eutrophication-induced changes in bioirrigation). Therefore, assuming that preservation of valves

below the mixed layer is permanent, our inferences about chronological changes in preservation are based on the subsets of stratigraphic profiles below the mixed layer (and on increments with more than 10 scored individuals). In analyses assessing bivariate and multivariate relationships among all alteration variables, we extracted the increments located below the mixed layer with more than 10 specimens, and measured Spearman rank correlations between per-increment frequencies of pyrite-lined valves and other types of alteration. We used principal coordinate analysis (PCO) and Manhattan distances to

investigate (1) geographic differences in preservation of *V. gibba* among all sites and (2) differences in preservation of *V. gibba* between the late 20$^{th}$ century increments (that primarily capture the main eutrophication phase) and increments that are older (corresponding to the early 20$^{th}$ century at the Po prodelta and the 16-19$^{th}$ centuries at the Isonzo prodelta) at the three prodelta sites with high sedimentation rates.

**3.5 Relationship with macrobenthic community composition**

To assess whether shell preservation responds to the ecosystem shift in the 20$^{th}$ century, we rank correlated the frequencies of pyrite-lined valves with two indicators of macrobenthic community states typical of the eutrophication phase (marked by high abundance and size of the hypoxia-tolerant bivalve *V. gibba*), including (1) proportional abundances of *V. gibba* in 5 cm increment (relative to the total molluscan abundance), and (2) maximum shell size of *V. gibba* measured by the 95$^{th}$

percentile of length distributions per increment. The chronological increase in abundance and size of this poorly-mobile species can reflect an increase in the contribution of surficial sediment modifiers to the functioning of benthic communities (*V. gibba* belongs to surface modifiers in the classification of mixing potential of benthic species, Queirós et al., 2013), with a concomitant decline in abundance of biodiffusors. Although proportional abundances are affected by between-core variability in time averaging and in net sedimentation rate, stratigraphic trends in proportional abundance within cores can be

robust because sedimentation rates remained relatively constant during the deposition of individual cores.

**4. Results**

**4.1 Preservation of pyrite and valves at light-microscope scales**

The preservation of *V. gibba* valves varies from well-preserved valves with (Fig. 5A-B) or without periostracum (Fig. 5C-

D), with well-preserved pallial line and the internal conchiolin groove (pl and clg in Fig. 5), to valves with signs of fine-scale dissolution and delaminated into two distinct valve layers (originally separated by the internal conchiolin layer, Fig. 5E-F and 5G-H). This delamination of valves into two layers is a characteristic signature of *V. gibba* preservation driven by the decay of internal conchiolin layer at all sites (Fig. 5E-H, 6C-H). The internal conchiolin layer (icl) located ventrally below the pallial line can be exposed in fragments (Fig. 5I) or in delaminated valves (Fig. 5F). The interiors of valves at Po and

Panzano are frequently lined by finely-disseminated microcrystals and framboids (Fig. 5J-K) or by strings (Fig. 5L), continuous (Fig. 5M-N) or patchy coatings (Fig. 5O-P) formed by pyrite framboids. In contrast, internal pyrite linings are rare on valves at Piran and Brijuni (Fig. 6).

In thin sections, well-preserved valves without any pyrite framboids represent the initial preservation state prior to any alteration (Fig. 7A). The pyrite linings in thin sections are visible as linings formed by pyrite framboids dispersed on

interior valve surfaces (black grains in Fig. 7B-D) and in the conchiolin layer (Fig. 7E). Pyrite linings are frequently associated with internal fine-scale dissolution and can co-occur with isolated reddish grains of Fe oxides at all sites. However, the fine-scale dissolution of valves at Po and Panzano is minor and shallow, and is still associated with some

surviving portions of periostracum or conchiolin layer. Specimens from Piran and Brijuni are bored and dissolved but are rarely lined by continuous coatings of pyrite framboids (Fig. 6). However, valves at Piran and Brijuni can contain dispersed framboids that are mainly concentrated in borings (Fig. 7F). Some relatively pristine valves from Po and Panzano show a blueish-or gray-colored staining on the interior and exterior, still with well-preserved external ornamentation (Fig. 5Q-T). In contrast, gray-stained valves at Piran and Brijuni are worn, degraded by bioerosion, and encrusted by bryozoans, serpulids or coralline algae on interior sides (Fig. 6). In thin sections, macroborings produced by sponges are very rare in valves from Po and Panzano, and bioerosion is primarily limited to simple borings that are few microns thick. In contrast, valves from Piran and Brijuni exhibit dense borings (Fig. 7F).

## 4.2 Preservation of pyrite at 100-1,000x magnification

On one hand, BSE images show that pyrite framboids preserved on the interiors of well-preserved valves vary in shape, size (5-10 μm) and packing, ranging from densely- and regularly-packed microcrystals within spherical framboids up to loosely and irregularly-packed microcrystals within irregularly-shaped framboids (Fig. 8A-G). Framboids also partly fill pores between aragonite nodules (arrows in Fig. 8I) within the conchiolin layer (that is ~10-100 μm thick) where they can be partly altered by Fe oxide rims (Fig. 8H-I). Larger euhedral pyrite crystals were not observed on valve interiors. At Piran and Brijuni, pyrite framboids located in borings (secondary linings) occur within strongly bored and stained valves (Fig. 8L). On the other hand, in addition to framboidal pyrite that forms primary or secondary linings, another type of pyrite preservation can be detected at high magnification and is characteristic of stained valves. In BSE images, the valves that are stained contain disseminated or dispersed inclusions of nanopyrite (< 1 μm), located in primary inter-crystalline skeletal pores or in secondary pits formed by dissolution or bioerosion in the inner or the outer layer; they are not located on valve surfaces or within the conchiolin layer (Fig. 8J-L). Valve portions with high abundance of nanopyritic inclusions are charaterized by blueish or grayish color.

SEM images show that pyrite framboids on surfaces of well-preserved, non-bored or unworn valves at the Po and Isonzo prodeltas are attached to the original interior surface (primary linings). Dispersed microcrystals and subspherical to spherical framboids consisting of microcrystals uniformly cover inner, well-preserved and smooth surfaces or fill irregularities close to the groove around the termination of the internal conchiolin layers (Fig. 9A-D). Microcrystals and framboids co-occur and can be distributed across the whole interior surfaces in clusters (Fig. 9E), in strings (Fig. 9F) and locally co-occur with Fe oxides and gypsum (Fig. 9F). Pristine valves show no or weak bioerosion by larger borers (sponges), although they can be penetrated by simple non-branching borings with micrometric diameters. In contrast, pyrite framboids are infrequent on interiors of valves that are affected by dense borings produced by sponges, forming complicated galleries, with or without sediment infill (Fig. 9G-H). SEM and BSE observations thus show that pyrite framboids directly attached to interior surfaces of well-preserved and weakly-bored valves represent primary linings, whereas pyrite framboids occurring on highly-altered valves are located in borings and represent secondary linings.

## 4.3 Age and depth distributions

Age distributions of valves with and without pyrite linings show right-skewed shapes in the mixed layer at the prodelta sites, with median age equal to 7-10 (without pyrite) and 7-18 years (with pyrite linings) at Po (in the upper 20 cm) and to 11 (without pyrite) and 15 years (with pyrite linings) at Panzano (in the upper 6 cm), respectively (Fig. 10A-F). The increments in the mixed layer are thus dominated at these sites by recentmost cohorts younger than ~10 years. Therefore, the pyrite framboids form rapidly at yearly scales at Po and Panzano. The simple preservation model with temporally-constant burial outperforms or is equally efficient than more complex sequestration models both for valves with and without pyrite in the

mixed layer at Po and Panzano. Both types of valves exhibit the same distribution shape well-fitted by the exponential distribution, with a similar steepness of the right tail (black lines in Fig. 10A-F, Table S1). The support for the simple disintegration-burial model thus indicates that both types of valves are buried at a similar rate below the mixed layer at Po and Panzano (with overlapping 95% confidence intervals, Fig. 10A-F). Age distributions of *V. gibba* valves without pyrite linings in the mixed layer at Piran are right-skewed but heavy-tailed and contain valves that are older than 1,000 years (median age = 1,100 years; only one 140 years-old valve was lined by pyrite, figure not shown), indicating slower rates of burial than at Po or Panzano and more complex preservation dynamic associated with exhumation. $\lambda_2$ is two orders of magnitude smaller than $\lambda_1$, and $\tau$ is comparable to $^{14}$C shell-based estimates of net sedimentation rate (~0.01 cm/y), indicating much longer residence times of valves in the mixed layer and a temporal decline in their disintegration rate. At the scale of the whole cores, median ages of pyrite-lined valves and valves without pyrite are also comparable at Po (27 and 33 years with and without pyrite, respectively) and Panzano (100 and 124 years with and without pyrite linings, respectively) (Fig. 10G-L). However, the whole-core age distributions of valves at Po and Panzano show signs of multimodality and the frequency with pyrite-lined valves peaks at 1986 AD at Po and at 1963 AD at Panzano. Equally-old pyrite-lined valves and valves without pyrite at Po and Panzano tend to be located at similar depths within the mixed layer (Fig. 10M-O). Multimodal whole-core age distributions and stratigraphic changes in abundance indicate that input rates of *V. gibba* were not constant over the duration of core deposition at all sites and especially increasing in abundance after~1950 AD, (Tomašových et al., 2018). However, the burial-rate parameter based on the exponential model (assuming the temporally-constant input of *V. gibba*) can be realistic when based on the topcore increments characterized by yearly to decadal time averaging at Po and Panzano (i.e., these increments were deposited after the mid-20$^{th}$ century increase in dominance by *V. gibba*). The mean depth of valves younger than 10 years is 20 cm for specimens both with and without pyrite at Po, and pyrite-lined valves are located deeper (at 60 cm) than valves without pyrite (at 40 cm) in the 20-year cohort.

### 4.4 Preservation of pyrite-lined valves relative to other types of alteration

Controlling for postmortem age, valves lined by pyrite contain higher concentrations of amino acids than valves without pyrite at Po and Panzano (Fig. 11). Median concentrations of amino acids in valves with pyrite linings typically exceed those in valves without pyrite within the same age cohort (Fig. 11A-C), and the log-log relationship between postmortem age and amino acid concentrations show that intercepts of valves lined by pyrite exceed intercepts of valves without pyrite by a factor of ~2 (Fig. 11D-F). At the scale of individual specimens, pyrite-lined valves possess some relicts of periostracum or conchiolin layer more frequently than valves without pyrite linings (Fig. 11G-I), all articulated shells of *V. gibba* from Po and Panzano that were opened had black strings or coatings of pyrite framboids on the valve interiors. At the scale of increments, the relations between preservation of pyrite-lined valves and other types of alteration exhibit three patterns. First, per-increment frequencies of pyrite-lined valves rank correlate *negatively* with the frequencies of disarticulated shells (Fig. 12A), with the frequencies of valves without any relicts of periostracum or conchiolin layer (Fig. 12B-C), and with the frequencies of bioerosion, staining, and encrustation (Fig. 12D-F). Second, the rank correlation between per-increment frequencies of pyrite-lined valves and frequencies of dissolved valves across all sites is low. However, partitioning the sites by high and low sedimentation rate, frequencies of pyrite-lined valves correlate with frequencies of dissolved valves positively at Po and Panzano and negatively at Piran and Brijuni (Fig. 12G). Third, with the exception of pyrite linings, all other types of alterations rank correlate *positively* (Fig. 12H-O). These correlations are determined by the among-site gradient in sedimentation rate (associated with the gradient in grain size and carbonate content): *V. gibba* valves at sites with high sedimentation are well preserved and rarely bored, encrusted, or abraded (Fig. 5). In contrast, *V. gibba* valves at sites with slow sedimentation show a broader range of preservational signatures, with high abundance of bored, encrusted, worn and stained specimens (Fig. 6). Valves with periostracum or conchiolin layer are abundant at Po and Panzano (with 10-80%

specimens per increment), whereas such specimens are extremely rare at Piran and Brijuni (<1%). Principal coordinate analysis of increments from all sites shows a major separation in preservation of *V. gibba* among increments deposited at low and high sedimentation rate (Fig. 13A-B). The first PCO axis correlates negatively with frequencies of pyrite linings and positively with all other alteration variables (with the exception of fine-scale dissolution).

## 4.5 Stratigraphic trends in the frequency of pyrite-lined valves within the mixed layer

Within the mixed layer, the frequency of pyrite-lined valves does not show any consistent downcore trends at the Po prodelta (Fig. 13D). Although valves with pyrite increase in abundance below the mixed layer at the scale of 10 cm increments from 15-24% at 10-20 cm to 35-40% at 20-90 cm at at Po 4, there is no clear downcore trend across this transition at Po 3 where the frequency of pyrite-lined valves is ~30-40% both in the mixed layer and at 20-90 cm. The uppermost 5 cm at Po 3 and Po 4 already contain more than 20% and 40% of pyrite-lined valves, respectively (Fig. 13D). The abundances of pyrite-lined valves below the mixed layer increase from 10-20% at 4-8 cm to 55-70% at 12-20 cm at Panzano, with the increase coinciding with the base of the mixed layer at 6 cm. At Piran and Brijuni, the frequencies of pyrite-lined valves within the mixed layer do not show any obvious trend.

## 4.6 Stratigraphic trends in the frequency of pyrite-lined valves below the mixed layer

At sites with high sedimentation rate, the frequency of pyrite-lined valves is highest (50-80%) in the late 20[th] century increments in the upper part of cores (at 20-80 cm at Po and at 8-20 cm at Panzano, Fig. 13D), coinciding with the eutrophication and high frequency of hypoxia. These increments at Po are also characterized by lower mottling, by higher preservation of flood layers, and by lower time averaging (median interquartile age range ranges up to 15 years, Tomašových et al., 2018) when compared to increments in the lower part of cores (at 80-150 cm, with median IQR equal to 25-28 years, Tomašových et al., 2018). The maximum frequency of pyrite-lined valves declines downcore to 20% at the Po prodelta (at 80-150 cm) and to 30-40% at Panzano (at 20-150 cm) in increments deposited prior to the late 20[th] century. The frequencies of pyrite-lined valves do not change markedly at sites with slow sedimentation and are typically less than 20% at Piran and Brijuni. The frequencies of pyrite-lined valves at Piran and Brijuni further decline when they are limited to valves not affected by bioerosion (black circles in Fig. 13D). Focusing on these sites with high sedimentation rate, PCO shows consistent separation between the latest HST and the late 20[th] century increments (Fig. 13C), reflecting the chronological increase in the frequency of pyrite linings. This within-core increase in the frequency of pyrite linings is coupled with an upcore increase in the frequency of valves without the periostracum (Fig. 14A), in the frequency of articulated valves, and in the frequency of valves with fine-scale dissolution (Fig. 14B-C). At sites with slower sedimentation and coarser, carbonate-rich sediments, the frequency of pyrite-lined valves remains constant upcore (<20%) and the frequency of pyrite-lined valves not affected by borers is <10%. The frequency of bored specimens gradually increases from 20-30% in the lower (transgressive systems tract) increments to 70-80% in the upper (late-highstand systems tract) increments at Piran (Fig. 14). In contrast, the frequency of bored specimens gradually declines from 90-100% in the lower (transgressive systems tract) increments to ~40% in the upper highstand increments at Brijuni (Fig. 14). Although the frequency of pyrite linings is low, valves at Piran and Brijuni show another mode of preservation with pyrite: they frequently contain disseminated nanopyritic inclusions within borings, leading to their darkening and staining (> 20%).

## 4.8 Relationship between pyrite-linings and the composition of benthic communities

Per-increment maximum shell size of *V. gibba* correlates positively with its proportional abundance ($r = 0.3$, $p = 0.0006$) across all sites. Proportional abundance of *V. gibba* per increment correlates weakly with the per-increment frequency of valves lined with pyrite framboids ($r$ [prop. abundance] $= 0.21$, $p = 0.016$, Fig. 15A). In contrast to regional-scale

relationships that are affected by between-core differences in time averaging, the positive relationships between proportional abundance and the frequency of pyrite-lined valves are moderarely high at Po ($r$ [Po 3] = 0.61, $p$ = 0.0015, $r$ [Po 4] = 0.6, $p$ = 0.003, $r$ [Po pooled] = 0.56, $p$ < 0.0001, Fig. 15A). The relationships between maximum shell size of *V. gibba* and the frequency of pyrite linings are significantly positive at the scale of individual cores at sites with high sedimentation rate ($r$ [Po 3] = 0.65, p = 0.0007, $r$ [Po 4] = 0.49, $p$ = 0.017, $r$ [Po pooled] = 0.61, $p$ < 0.0001, $r$ [Panzano] = 0.46, $p$ = 0.0002, $r$ [Piran] = 0.55, $p$ = 0.16, r [Brijuni] = 0.11, $p$ = 0.71, Fig. 15B). Within-core stratigraphic series at Po and Panzano show autocorrelation in the frequency of pyrite-lined valves and in abundance and maximum shell size at lag 1 (stratigraphic series at Brijuni and Piran do not show autocorrelation in these traits). Although differencing the stratigraphic series to remove autocorrelation reduces the strength of the bivariate relationships, rank correlations between the frequency of pyrite-lined valves and maximum size of *V. gibba* remain significantly positive at Panzano ($r$ [size] = 0.32, $p$ = 0.017; $r$ [biomass] = 0.34, $p$ = 0.02) and of borderline significance at Po if increments from Po 3 and Po 4 are pooled ($r$ [size] = 0.3, $p$ = 0.047; $r$ [biomass] = 0.28, $p$ = 0.033). Therefore, the upcore increase in the frequency of pyrite-lined shells directly parallels ecological changes characterized by an abrupt increase in maximum shell size and abundance of *V. gibba*, contributing to increasing importance of shallow-infaunal groups that mix uppermost sediment zones but have weaker bioirrigation potential.

## 5. Discussion

### 5.1 Effects of net sedimentation rate and grain size on preservation of pyrite linings

Our observation that the pyrite-lined valves are preserved at high abundance in the subsurface stratigraphic record at the Po and Isonzo prodeltas (with sedimentation rates exceeding ~0.2 cm/y) whereas they are infrequent at Piran and Brijuni (with sedimentation rates < ~0.02 cm/y) indicates that the first-order condition for their permanent sequestration below the mixed layer is enhanced when net sedimentation rate is high and sediments are fine-grained, with residence time of sedimentary particles (including valves) in the mixed layer (when they may be exposed to $O_2$ from bioirrigation) not exceeding more than several years or few decades. Some pyrite framboids on valves at coarse-grained sites with slow sedimentation represent secondary linings because they directly occur within borings in heavily-bored valves (i.e., the difference between frequencies of pyrite linings in all and non-bored valves is large at Brijuni, Fig. 13D). This preservation contrasts with well-preserved pyrite-lined valves with well-developed pyrite linings (not limited to borings) at sites with high sedimentation where exclusion of bored valves does not reduce their frequency (i.e., the difference between frequencies of pyrite linings in all and non-bored valves is small at Po and Panzano, Fig. 13D). The preservation of primary pyrite linings is thus largely limited to sites with high sedimentation rate (> 0.2 cm/y).

First, larger grain size and sediment porosity of coarse-grained sediments enhance advective pore-water fluxes driven by bottom currents, hydraulic sediment reworking, or by bioirrigation (Taylor et al., 2003; Mermillod-Blondin, 2011; Ahnerkamp et al., 2017). They deepen the oxycline and promote higher exposure of organics within shells to $O_2$ and stronger iron recycling (Aller, 1994) relative to fine-grained sediments where pore water fluxes are dominated by diffusion (Mermillod-Blondin and Rosenberg, 2006; Meysman et al., 2006). Muddy sediments are on average oxygenated to less than 1-2 cm in the northern Adriatic Sea (Epping and Helder, 1997; Moodley et al., 1998, 2000), and zones with high concentrations of dissolved iron tend be limited to the uppermost 5 cm in the NW Adriatic Sea (Hammond et al., 1999) and in the Gulf of Trieste (Faganeli and Ogrinc, 2009). In contrast, sandy sediments exhibit stronger bioirrigation in the upper 20 cm (Faganeli and Ogrinc, 2009). Such conditions can inhibit the initial formation of reduced microenvironments. Second, the

lower concentrations of iron in sediments can reduce the probability of initial pyritization at coarse-grained sites rich in carbonate sediments even in reduced microenvironments, with $H_2S$ diffusing away from the decay sites owing to the low availability of dissolved iron (Raiswell et al., 1993; Allen, 2002). Third, at sites with slow sedimentation rates (~0.01 cm/y) and coarse and permeable sediments less rich in iron and organic carbon, regardless of bioturbation, valves will be exposed to redox fluctuations in the TAZ for longer than at sites with high sedimentation rate, leading to intense alteration of valves by borers, encrusters, or by pore waters that tend to be undersaturated with respect to aragonite in the upper 10-20 cm at the Po prodelta (Hammond et al. 1999) and in the Gulf of Trieste (Ogrinc and Faganeli, 2003). This effect will minimize both the initial formation of reduced microenvironments and their susceptibility to reoxidation. Higher levels of bioerosion, encrustation, ornamentation loss and staining at Piran and Brijuni than at Po and Panzano (Fig. 12-13) indicate that slow sedimentation rates (~0.01-0.03 cm/y) reduce the preservation potential of framboidal linings by increasing their exposure to bioirrigation or to exhumation. In contrast to very low frequency of pyrite-lined valves at sites with low sedimentation rate, frequency of valves stained by nanopyritic inclusions is higher at these sites, probably reflecting long (millennial-scale) subsurface exposure of less reactive iron-bearing minerals (relative to highly reactive iron oxides and oxyhydroxides) to sulfidic conditions (Raiswell and Canfield, 1996). A similar relationship with darkened specimens enriched in minerals possessing iron sulfides being more damaged by disarticulation, bioerosion and encrustation, in contrast to specimens less affected by discoloration, was observed by Kolbe et al. (2011) in Ordovician brachiopods. Low sedimentation rates associated with coarse-grained sediments thus exacerbate not only residence time of skeletal remains in the shallowest sediment zones but also their net exposure to $O_2$. Higher recycling of organic carbon, nitrogen and sulfur at sites with slow sedimentation in the northern Adriatic Sea is supported by pore-water biogeochemical profiles that show their higher burial efficiencies at the Po prodelta relative to sites with slower sedimentation in the NW Adriatic Sea (Giordani et al., 1992, 2002; Hammond et al., 1999). Similarly, recycling efficiency of organic carbon and nitrogen is about 85% at Piran and in the central parts of the Gulf of Trieste and less at proximal deltaic locations off the Isonzo prodelta (Faganeli and Ogrinc, 2009). High sedimentation rates typical of deltaic environments can limit mixing and irrigation via high substrate instability (MacEachern et al., 2005; Bhattacharya et al., 2020). Benthic communities at the Po Delta are indeed affected by short-term seasonal variability in sediment input and reworking (Ambrogi et al., 1990; Paganelli et al., 2012). However, habitats deeper than ~20 m at Po prodelta are largely beyond the reach of proximal deposition of thicker flood deposits (Palinkas and Nittrouer, 2007; Tesi et al., 2012). The benthic communities inhabiting these habitats seem to be mainly limited by the frequency of hypoxic events rather than by substrate instability (Crema et al., 1991; Simonini et al., 2004; Tomašových et al., 2020). Although high background sedimentation rate (or rapid episodic burial) is a necessary condition for preservation of pyrite linings, we argue below that it is not a sufficient condition, and that potential for reoxidation must be also reduced by disturbances (such as hypoxia), which limit the abundances of irrigating infauna.

**5.2 Pyrite-lined valves in subsurface zones as signatures of limited bioirrigation.** Pyrite framboids formed soon after, or concurrent with, the decay of tissues of *V. gibba* and associated microbes. This is indicated by the preferential occurrence of pyrite linings in valves with periostracum and with higher organic content (with median age < 15 years in the surface mixed layer) and the negative correlation of pyrite-lined valves with the frequencies of other types of alteration at the scale of increments (framboids arranged in filaments or strings in Fig. 5L or 7F resemble bacterial relicts, Westall, 1999; Wilson and Taylor, 2017). Two steps are necessary for preservation of pyrite-linings in the subsurface stratigraphic record, including their initial formation in reduced microenvironments rich in dissolved iron and the subsequent lack of their reoxidation.

**(1) Initial sequestration in reducing, iron-rich zones.** The early formation of concentrations of pyrite framboids that cluster on valve surfaces requires (1) that the organic tissues within shells do not decay or are not scavenged under aerobic conditions and (2) that sediment is rich in iron so that sulfides produced by sulfate reduction are confined to the decay

location (Fisher and Hudson, 1987; Raiswell et al., 1993; Farrell et al., 2009; Schiffbauer et al., 2014). Such early-sequestration conditions can occur under the decay of organics in suboxic conditions in non-sulfidic, iron-dominated porewaters (Berner, 1969; Briggs et al., 1996; Allen, 2002; Raiswell et al., 2008; Stockdale et al., 2010): (i) when freshly-dead shells are episodically buried under a deposition of new sediment (obrution, Brett et al., 1997) sourced by river floods or storms, and can decay in reducing conditions beyond the reach of burrowers; the thickness of the obrution deposits needs to be sufficiently high so that pyrite is not reoxidized by burrowing infauna later (Allison, 1988; Brett et al., 2012a, b; Schiffbauer et al., 2014) and/or (ii) when bioirrigation is shallow or intermittent and biomass of decaying infaunal organisms remains tightly enclosed within shells (as in *V. gibba*) or within burrows (Thomsen Vorren, 1984; Hansen et al., 1996), generating reducing microenvironments even without obrution (Jorgensen, 1977). Although these two pathways can be complementary, they may also act independently.

The first scenario with obrution is frequently invoked in the deep-time stratigraphic record because it explains the short exposure of organic remains to $O_2$ and their rapid sequestration below the mixed layer into the historical layer (Brett et al., 2012a). Distinct layers deposited by major decadal floods preserved in cores at Po prodelta (Tesi et al., 2012; Tomašových et al., 2018) may have triggered episodic burial of benthic communities with *V. gibba*, but similar flood-event layers were not detected at Panzano. However, first, the major discharge events that lead to the deposition of flood deposits occur at decadal scales at the Po prodelta (seven events during the 20[th] century, Zanchettin et al., 2008), whereas age distributions of pyrite-lined valves indicate that pyrite linings form continuously at yearly scales. The frequency of flood events did not increase during the 20[th] century, in contrast to the increasing frequency of pyrite-lined valves in the upper parts of cores at Po and Panzano. Second, pyrite-lined valves younger than 10 years old occur at high abundance in the uppermost 5-10 cm of the mixed layer close to the sediment-water interface, indicating that they do not represent transient valves that were just recently exhumed from deeper zones. Third, equally-old valves with and without pyrite linings occur at similar depths and their age distributions at Po and Panzano are similar (Fig. 9A-F), indicating that valves with and without pyrite were buried below the mixed layer at a similar rate. A single episodic burial pulse that mixes living or recently-dead shells (with decaying biomass and potential for framboid formation) with older shells (without biomass) will generate one age distribution of valves with pyrite linings dominated by younger cohorts and another age distribution of valves without pyrite linings dominated by older cohorts. Age distributions of valves with pyrite linings generated by such episodic burial should be steeper and their median ages should be lower relative to valves without linings, in contrast to our observations both at Po and Panzano. We thus suggest that the pyrite framboids did not preferentially form within valves that were rapidly buried deeper in sediments (either by new sediment deposition or by burrowers) as predicted by the obrution scenario and rather precipitated in near-surface sediment zones naturally inhabited by *V. gibba*.

In the second scenario, shells can be located in reducing microenvironments close to the sediment-water interface even without any fast episodic burial by obrution when irrigation is persistently patchy and a large portion of dead shells is exposed to reducing conditions in sediment pockets. The uppermost sediments become oxygen-depleted during late-summer hypoxic events in the northern Adriatic Sea (Stachowitsch, 1984; Cermelj et al., 2001), with the oxygen demand of sediments increased by the decay of phytoplankton and high-biomass benthic communities during late-summer mucilage and mass mortality events (Stachowitsch, 1984; Herndl et al., 1987, 1989; Nebelsick et al., 1997). Such organic enrichment can lead to porewater sulfidization and to exhaustion of highly reactive iron from porewaters, but organic-rich sites at the Po prodelta still show high concentrations of dissolved iron in the uppermost few cm (Barbanti et al., 1995). Sediment mixing by (weakly-irrigating) infauna tends to counteract the exhaustion of dissolved iron and the potential buildup of $H_2S$ in porewaters because burrowers transfer particles with iron oxides from the sediment-water interface to reducing conditions in subsurface zones (Dhakar and Burdige, 1996; Faganeli and Ogrinc, 2009; van de Velde and Meysman, 2016). Several lines

of evidence indicate that this second scenario, the spatially-limited bioirrigation coupled with the inherent potential of decaying tissues in articulated shells of *V. gibba* to generate reducing microenvironments in near-surface muddy sediments with iron-dominated porewaters, is important. First, qualitative observations indicate that pyrite linings are very rare in valves of *Nucula* that co-occur with pyrite-lined *V. gibba* valves, and pyrite linings occur rarely in apertures of gastropods. The tightly-articulated shells of *V. gibba* can be intrinsically susceptible to the formation of reducing conditions owing to (i) the overlapping periostracum, (ii) the internal groove that can lock valves to some degree, and (iii) low opening moments of a small internal ligament that can be insufficient to open valves against the sediment pressure (as in other members of the Myoidea, Trueman, 1954; Yonge, 1982). Second, the high abundance of very young and well-preserved pyrite-lined valves in the uppermost 5-10 cm of the mixed layer suggests that they form in the uppermost sediment zones and that the oxycline thus tends be shallow. With the exception of burial of *V. gibba* to their position just below the sediment-water interface during their life, postmortem obrution by episodic sediment deposition is not needed for the initial formation of pyrite linings when sediment irrigation is limited (but biomixing still allows iron recycling to generate zones rich in dissolved iron).

**(2) The lack of reoxidation.** Most of the sulfide is reoxidized in marine environments with bioirrigation and sediment exhumation by burrowers (Canfield et al., 1993; Thamdrup et al., 1994a). Reduced conditions with pyrite linings within shells or burrows thus will be transient when sedimentation rate is low and/or when bioirrigation and sediment exhumation by burrowers oxidize such microenvironments (Bertics and Ziebis, 2010). The positive relation between pyrite and fine-scale dissolution, and transformation of some pyrite grains to iron oxides and gypsum indicates that some pyrite-lined valves at Po or Panzano were briefly exposed to oxygenated pore waters and were partly dissolved by sulfuric acid (Cai et al., 2006; Pirlet et al., 2010; Hu et al., 2011). However, this type of replacement was limited in extent because it did not fully remove the pyrite linings, and many valves at Po and Panzano were still buried at sufficiently high rate relative to the time scale of pyrite oxidation so that a significant subset of pyrite-lined valves survived into the subsurface record. The conditions that ultimately allow the transit of reduced microenvironments formed by pyrite-lined shells at sites with relatively high background sedimentation rate into the subsurface stratigraphic record can occur when bioirrigation is slow or temporally-impersistent. These conditions can occur when the recovery of infaunal communities in the wake of anoxic or hypoxic events (as in the northern Adriatic Sea) is slow or non-existent owing to hysteresis (when a community is pushed into an alternative stable state by hypoxia and remains in this state even when hypoxic conditions abated) (Bentley and Nittrouer, 1999; Solan et al., 2004; Elliott et al., 2007; Borja et al., 2010; Duarte et al., 2015).

Depending on the frequency and duration of hypoxic events or other disturbances limiting the recovery of benthic fauna, two preservation scenarios can be envisioned (Fig. 16): (i) If the frequency of hypoxic events is low and infaunal organisms both efficiently mixing and irrigating sediments rapidly recover, articulated shells will disarticulate and disintegrate when exposed to scavengers, borers and degradation of organics in the TAZ. For example, maceration of the conchiolin layer triggers delamination of valves into inner and outer layers (Fig. 5E-H). Although a thick suboxic zone can develop under such conditions, the depth of the oxycline within the mixed layer will also increase (Fig. 16A). First, pyrite framboids will not nucleate on valves anymore if the labile biomass and microbes coating the decaying tissues were degraded during the earlier phase of aerobic decomposition. Second, although bacterial sulfate reduction in microenvironments surrounded by iron-dominated pore waters triggers initial formation of pyrite linings in some shells, oxidation induced by bioirrigation will catch up with pyrite-lined shells prior to their deep burial and thus inhibit their subsurface sequestration. (ii) If the frequency of hypoxic events is high relative to the recovery time of the burrowing community with efficient bioirrigators, the early recovery of deposit and detritus feeders that mix the uppermost sediments can support the formation of the ferruginous suboxic zone (Aller et al., 1986; van de Velde and Meysman, 2016), thus allowing the precipitation of pyrite framboids at the location of decaying molluscs. If subsequent recovery of bioirrigation-

inducing burrowers is slow or interrupted by another hypoxic event or if communities with infrequent bioirrigators are locked by hysteresis effects (Kemp et al., 2009; Duarte et al., 2015), the oxycline can permanently remain close to the sediment-water interface (Fig. 16B), and some subset of pyrite-lined shells can remain preserved. These conditions with reduced bioirrigation are typical of eutrophied environments (Karlson et al., 2007; Lehtoranta et al., 2009). If the nutrient-fueled eutrophication or other sources of sediment organic enrichment lead to permanent anoxia of bottom waters, the concentrations of pyrite framboids within shells can be prohibited because mixing of sediments by burrowers that underlies the iron-based redox shuttle is aborted and sulfide production by bacterial sulfate reduction in organic-rich sediments can exceed the availability of reactive iron oxides, with $H_2S$ diffusing away (Raiswell and Berner, 1985; Schenau et al., 2002; Raiswell et al., 2008; Farrell et al., 2009), or ferrous iron can be released from sediments to the non-sulfidic water column (Pakhomova et al., 2007). The absence of pyrite linings in anoxic sediments documented in the deep-time stratigraphic record (e.g., Brett et al., 1997) indicates that in marine environments with persistent bottom-water anoxia, the window for the early and rapid formation of pyrite linings (e.g., within shells of nektonic groups such as cephalopods that fell on the anoxic seafloor) will be closed when (a) the suboxic zone is not induced by biomixing and (b) organic matter degradation in surface sediments leads to the excess of hydrogen sulfide (Middelburg and Levin, 2009).

## 5.3 Temporal changes in bioirrigation generate stratigraphic trends in frequency of pyrite-lined valves

The upcore increase in the frequency of pyrite-lined valves at prodelta sites and the modality of whole-core age distributions of pyrite-lined valves hint to chronological changes in the preservation of pyrite linings. First, the stratigraphic increase in the frequency of pyrite-lined valves and the multivariate segregation in preservation of *V. gibba* valves between the early and the late 20[th] century increments at Po and Panzano indicate that $O_2$ exposure of shells in the mixed layer was longer prior to and during the early than during the late 20[th] century. Any effects on early diagenetic pathways determined by changes in grain size can be excluded because sediment grain size remains constant downcore at Po and Panzano. Second, a general feature of the whole-core age distributions is the dominance of valves younger than 10 years, generated under active input of new dead shells into the mixed layer. Older modes in these distributions are indicative about temporal changes in conditions that favored or inhibited the preservation of pyrite linings. At Po, pyrite-lined valves show a secondary mode at ~30-50 years that is formed by cohorts from the late 20[th] century, and most valves older than ~100 years do not have pyrite linings (Fig. 10E-F). Similarly, at Panzano, pyrite-lined valves form a mode that is represented by cohorts from the late 20th century (Fig 10G-H). Although paleoecological records of foraminifers and dinoflagellates indicate that eutrophication affects the northern Adriatic Sea ecosystems since the 19[th] century (Barmawidjaja et al. 1995; Sangiorgi and Donders 2004), a main shift in the composition of macrobenthic communities and size of the bivalve *V. gibba* occurs at the same stratigraphic levels where pyrite linings increase in frequency, i.e., at 80-90 cm at Po and at 12-20 cm Panzano (Fig. 15, Tomašových et al., 2018, 2020). These levels correspond to the mid-20[th] century at both prodeltas, and coincide with an increase in nutrient load and with an increase in seasonal hypoxia and mucilage frequency in the northern Adriatic Sea (Marchetti et al., 1989; Justic, 1991). Ecological surveys performed in the late 20[th] century also indicate that a major shift in the composition of soft-bottom benthic communities (towards the dominance by *V. gibba*) in the NW Adriatic Sea took place in response to eutrophication and hypoxic events after the early 20[th] century (Crema et al., 1991). Two observations support the hypothesis that the stratigraphic shift towards higher pyrite frequency of pyrite-lined shells coincides with changes in the composition of macrobenthic communities towards states that still produced a well-mixed layer but were characterized by smaller bioirrigation efficiency in the late 20[th] century at Po and Isonzo prodeltas. First, although the thickness of the mixed layer at the Po prodelta as estimated on the basis of sediment cores probably exceeded 20 cm prior to the late 20[th] century (Tomašových et al., 2018), abundant burrows of shrimps (Fig. 1D) and the 16-20 cm-thick mixed layer at Po sites observed

at the time of sampling in 2013 and 2014 indicate that local-scale biomixing remains active in the uppermost sediment zones. The thickness of the mixed layer documented here and by other studies (Frignani and Langone, 1991; Frignani et al., 2005; Alvisi et al., 2006, Alvisi, 2009) is typical of the mixed-layer thickness observed in other marine shelf environments (Moodley et al., 1998; Teal et al., 2010) and the depth of mixing is also not smaller than the mixed-layer thickness (< 5 cm) typical of persistently-hypoxic environments on continental slopes (Meadows et al. 2000; Smith et al. 2000; Levin et al.

2003). Second, our inference about the role of inefficient bioirrigation in sequestration of pyrite linings in the subsurface stratigraphic record, as envisioned in the scenario in Fig. 16B, is supported by changes in the functional composition in benthic communities. At the Po and Isonzo prodeltas, the late 20th century infauna is dominated by shallow-burrowing deposit- and detritus-feeders that modify the uppermost centimetres of surface sediments, including *Owenia fusiformis, Varicorbula gibba* or *Ampelisca diadema* (Occhipinti-Ambrogi et al., 2005; Solis-Weiss et al., 2007; N'Siala et al., 2008).

These patterns differ from early 20th century ecological surveys (Vatova, 1949; Crema et al., 1991; Schinner, 1993; Schinner et al., 1997) and the early 20th century increments in cores (Gallmetzer et al., 2017; Tomašových et al., 2018) that indicate that pre-eutrophication benthic communities were dominated by biodiffusors with relatively high irrigation potential, such as *Ova, Amphiura* or *Turritella*. The ophiuroid *Amphiura filiformis* extensively reworks and irrigates sediments (Solan and Kennedy 2002; Vopel et al., 2003), although predation pressure can reduce its irrigation effects (Wrede

et al. 2017) or the gastropod *Turritella communis* are not abundant at prodelta sites and tend to occur at deeper sites with slower sedimentation rates in the late 20th century surveys (Chiantore et al., 2001). In contrast to *Amphiura*, the ophiuroid *Ophiura albida* that is abundant at the Po sites belongs to the functional group of slow biodiffusors (Queirós et al., 2013).

## 6. Implications for the fossil record: inferring slow and patchy bioturbation and limited residence time

The pyrite framboids lining intra-skeletal pores (originally filled with organic tissues) in well-preserved shells represent a unique indicator of slow, spatially- and temporally-limited irrigation. Such conditions can be produced by delayed recoveries from hypoxic events and/or by community states with low bioirrigation potential unable to recover anymore even when the frequency of hypoxic events returns to pre-impact levels (Steckbauer et al., 2011). These conditions can also characterize marine benthic ecosystems prior to the appearance of adaptations that allow deep bioirrigation or marine benthic ecosystems

that were impoverished in functioning in the aftermath of mass extinctions. In permanently-normoxic environments with intense bioturbation where most labile biomass degrades within the aerobic zone and any early pyrite is oxidized, the frequency of shells with shell-lined pyrite transferred into the permanent record will be negligible. This index can be also used to track the net $O_2$ exposure of skeletal remains and recycling efficiency of iron and sulfides in the deep-time stratigraphic record because pyrite-lined shells represents a distinct taphonomic and diagenetic signature of fossil

assemblages preserved in fine-grained sediments (Kobluk and Risk, 1977; Hudson, 1982; Bjerreskov, 1991; Underwood and Bottrell, 1994; Farrell et al., 2009; Brett et al., 2012a), especially in organisms with internal skeletal cavities that do not immediately open after their death (Hudson, 1982; Fisher, 1986; Loope and Watkins, 1989; Jin et al., 2007). On one hand, preservation of well-preserved, frequently articulated skeletal remains of organisms with otherwise fragile elements, coupled with pyrite linings, is a typical taphonomic feature of assemblages preserved in fine-grained sediments in Paleozoic (Brett et

al., 2012a, b) or in Mesozoic successions (Hudson, 1982; Fernández-López et al., 2000; Paul et al., 2008; Reolid, 2014). On the other hand, actualistic studies assessing skeletal alteration of molluscs, brachiopods or echinoderms rarely record this type of preservation in surface Holocene sediments (Staff and Powell, 1990; Kowalewski et al., 1994; Kidwell et al., 2005; Best et al., 2007). It is possible that the concentration of actualistic studies on the taphonomic processes in the uppermost, typically-well-irrigated portions of the mixed layer and on environments with slow sedimentation rates underestimate this

type of preservation. However, here, we suggest that prodelta sediments in the northern Adriatic Sea affected by the late 20th century eutrophication can represent analogue conditions that lead to preservation of well-preserved and pyrite-lined shells

in the deep-time stratigraphic record. In contrast to obrution models invoking rapid burial to explain the initial sequestration of shells (so that they do not disarticulate and decay under reducing conditions), the pathway observed in prodelta sediments in the northern Adriatic Sea probably occurs without episodic burial of valves (by burrowers or by new sediment) because (1) very young pyrite-lined valves (< 10 years old) occur in the uppermost zones, (2) stochastic burial by burrowers would lead to deeper location of pyrite-lined valves and some differences in the depth of valves with and without pyrite, and (3) mixing smears out the depth location of shells after the deposition of thin flood layers, so that the depth of shell burial initiated by obrution does not remain constant. However, background net sedimentation rates need to be sufficiently high so that the time for burial of skeletal remains is shorter than the time to irrigate the mixed layer.

**7. Conclusions**

Preservation of pyrite-lined shells as a function of rapid and permanent sequestration of shells in pockets rich in dissolved iron is an indicator of inefficient bioirrigation at sites with high background sedimentation rates in the northern Adriatic Sea. Although sediment mixing affects the uppermost 10-20 cm of sediments in the northern Adriatic Sea, irrigation is reduced relative to pre-eutrophication community states. The preservation pathway that leads to primary pyrite linings and their long-term preservation is indicative of permanently-limited depths of $O_2$ penetration induced by bioirrigation that can be difficult to detect on the basis of trace fossils and ichnofabric only. Pyrite-lined valves thus represent a unique type of alteration that contrasts with other types of alteration (including the frequency of stained valves with nanopyritic inclusions) whose incidence increases with residence time in the TAZ. Our analyses indicate that obrution by episodic events is not necessary for preservation of well-preserved pyrite-lined valves and that the key factor is low bioirrigation, although still conditioned by relatively high background sedimentation rates that are typical of prodeltas. We suggest that the increase in the frequency of valves with pyrite below the mixed layer at 80-90 cm at the Po prodelta and at 12-20 cm at the Isonzo prodelta represents a temporal signal of the decline in the rate of bioirrigation in muddy sediments of the northern Adriatic Sea driven by a late 20th century increase in the frequency of hypoxia that delayed the recovery of infaunal communities. Although the rates of pyrite formation can also vary over long time scales owing to long-term changes in seawater chemistry (Leavitt et al., 2013; Algeo et al., 2015) and depend on the supply of organic matter and iron availability (Goldhaber et al., 1977; Berner and Raiswell, 1983; Berner, 1984; Berner and Westrich, 1985; Kershaw et al., 2018; Wignall et al., 2005, Bond and Wignall, 2010), we hypothesize that the frequency of pyrite-lined shells (belonging to organisms that inhabit oxic sediment zones) can improve inferences about irrigation efficiency of past benthic communities preserved in the fossil record.

**8. Code availability**: The R language source code is available in the Supplement.

**9. Data availability**: The dataset reported with scoring and age data of individual shells from two cores collected at Po, from Panzano, Piran and Brijun is available in a tab-delimited txt file in the Supplement. This file is a data source for analyses shown in the R language code.

**10. Author contribution**: AT designed the study. IG, AH and MZ conducted sampling, and AT, IG, AH, and MB conducted taphonomic scoring. AT conducted analyses, and all authors contributed to writing and revisions.

**11. Competing interests**: The authors declare that they have no conflict of interest.

**12. Acknowledgments**

We thank Carlton Brett, Lidya Tarhan and an anonymous reviewer for critical comments. This work was supported by the
Austrian Science Fund (FWF) [grant number P24901, 2013], by the Slovak Research and Development Agency (APVV
0555-17) and by the Slovak Scientific Grant Agency (VEGA 2/0169-19). Many thanks to the captain of the sampling vessel,
Jernej Sedmak, for his commitment during the sampling campaigns in 2013 and 2014.

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

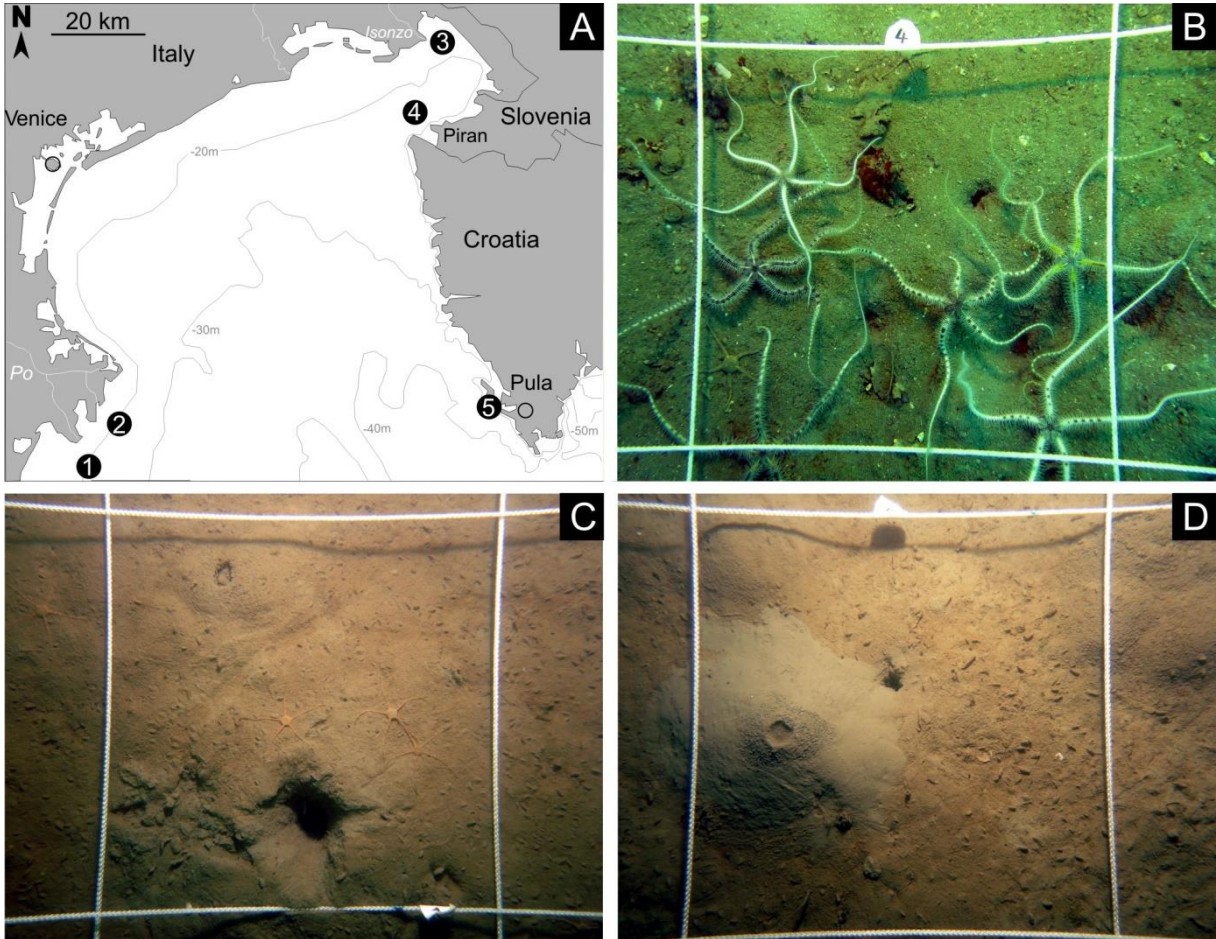

**Figure 1** – A. Geographic map with the locations of five sites analyzed in this study: 1 – Po 4 (with two replicate cores M20 and M21), 2 - Po 3 (with two replicate cores M13 and M14), 3 – Bay of Panzano in the northern Gulf of Trieste (with two sediment cores M28 and M29), 4 – Piran (M53 core) and 5 – Brijuni (M44). B-D. Sea floor photographs taken in 2014 at Piran (B, with abundant *Ophiotrix*), Po 4 (C, with abundant *Ophiura albida*) and Po 3 (D) documenting abundant burrow and mound structures and uneven topography. The size of the square is 25 cm$^2$. The thickness of the mixed layer at Po is ~20 cm on the basis of both $^{14}$C and $^{210}$Pb and also at Brijuni, and ~ 8 cm at Piran, underlain by a shell bed.

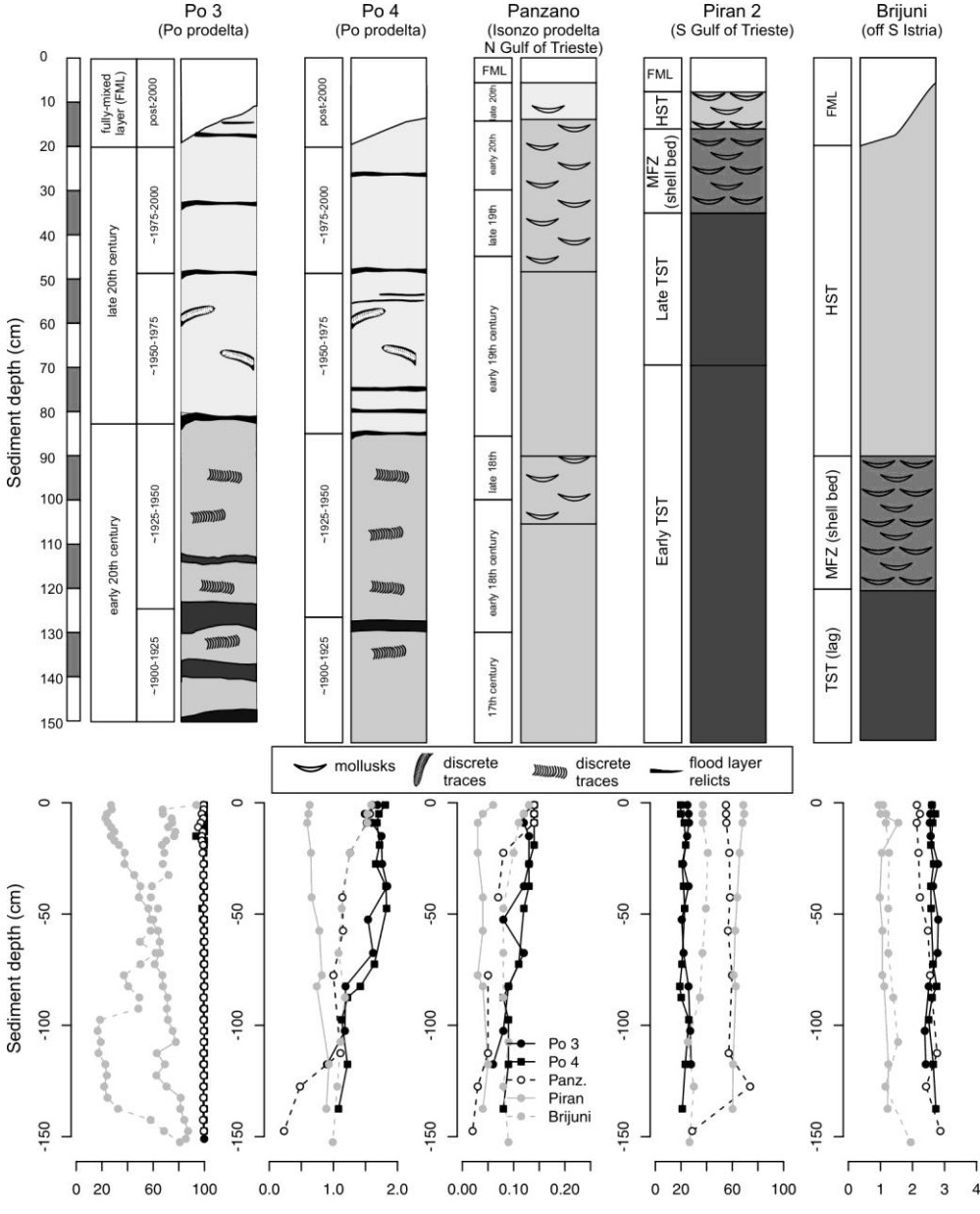

**Figure 2** – Top row: Lithologic sections through sediment cores at five locations, with the highest sedimentation rate (1-2 cm/y) at Po sites, intermediate net sedimentation rate (0.2-0.4 cm/y) in the Bay of Panzano, and very slow sedimentation rate at Piran and Brijuni (~0.01 cm/y), with chronostratigraphic and sequence stratigraphic subdivisions. TST – transgressive systems tract, MFZ – maximum flooding zone, HST – highstand systems tract, FML – fully-mixed layer (its thickness varies within cores depending on differences between the estimates based on [210]Pb profiles and downcore changes in median shell age). Bottom row: Stratigraphic trends in grain size, in the weight percent of total organic carbon (TOC), total nitrogen (TN), CaCO$_3$ (based on total inorganic carbon), and the weight percent of iron also discriminate between sites with higher sedimentation (Po and Panzano) and sites with slower sedimentation (Piran and Brijuni).

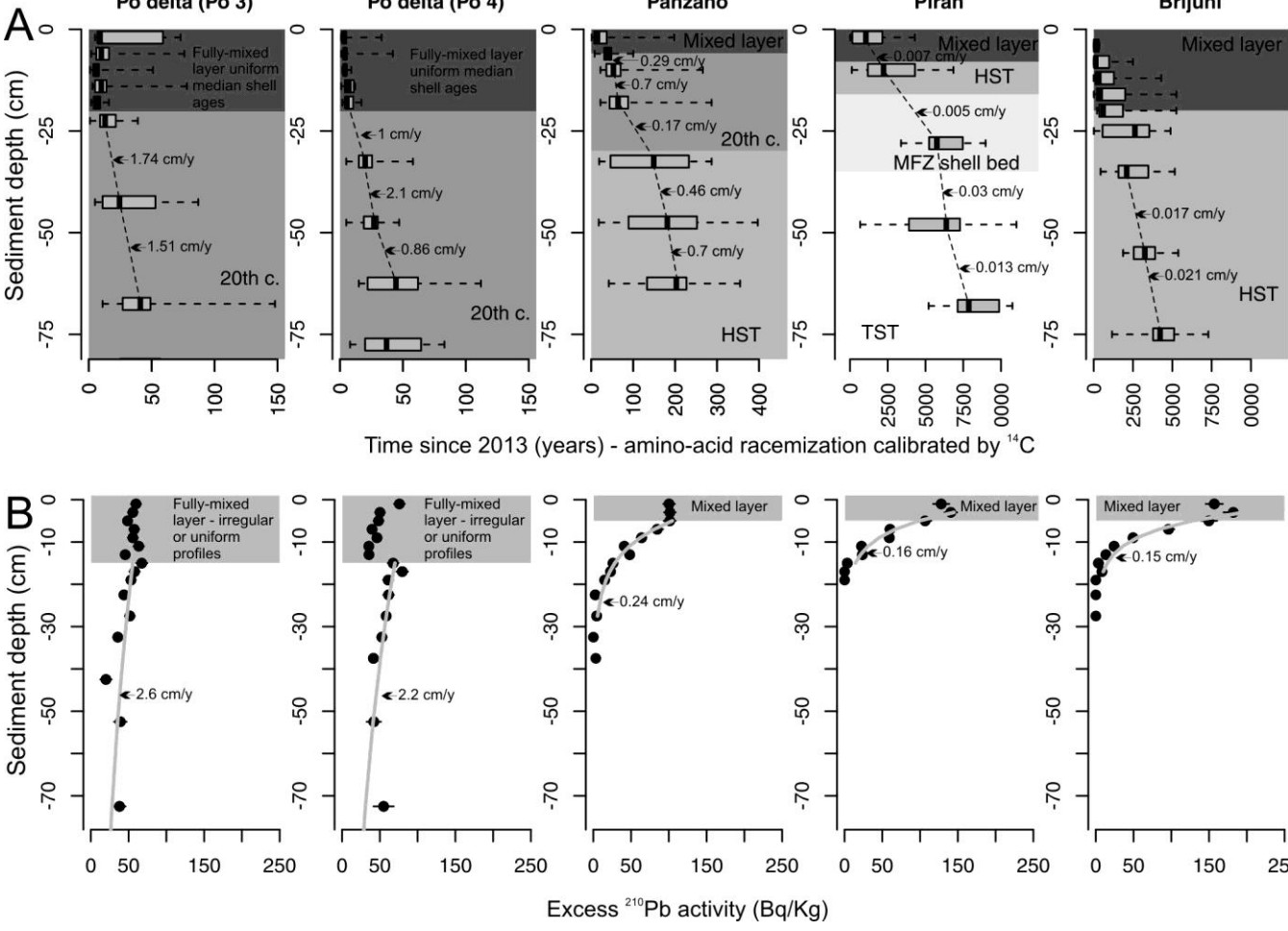

**Figure 3** – Downcore changes in shell ages (A) and the excess in [210]Pb profiles (B) that form the basis for inferences about the depth of the mixed layer (absence of downcore changes in median shell age in the uppermost parts of cores and irregular or uniform segments of the [210]Pb excess) and the sedimentation rate (thickness deposited over a duration defined by differences in median ages, and the slope of the [210]Pb segments below the mixed layer in gray color). We note that some variability in these estimates is also affected by biomixing when the thickness of sediments over which the deposition is measured is too low relative to the thickness of the mixed layer, leading to overestimation of the sedimentation rate.

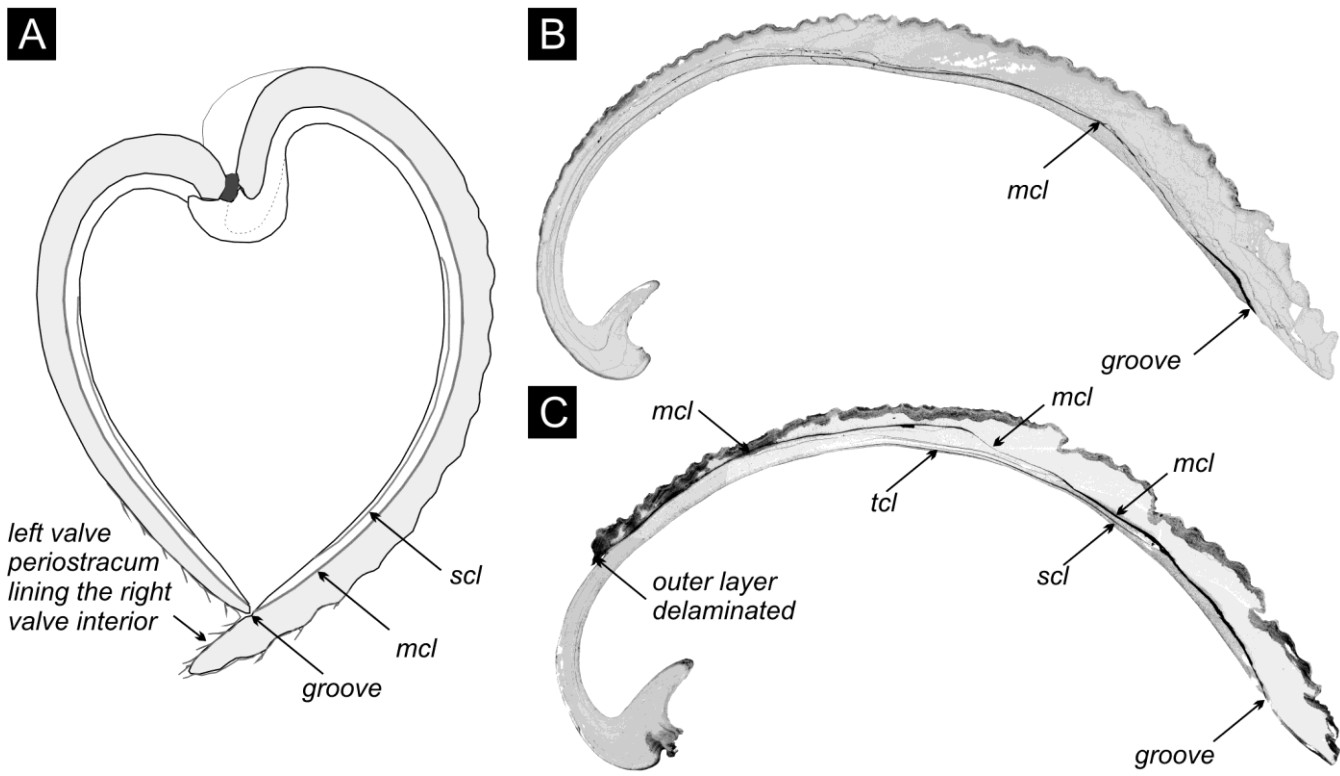

**Figure 4** – A. The cross-section of the articulated shell of *V. gibba*, adapted from Yonge (1946) and Lewy and Samtleben (1979), shows that the internal conchiolin layer (mcl) separates both valves into an inner and an outer layer, and the inner layer can contain another secondary conchiolin layer (scl). The left valve fits into a groove with the conchiolin layer on the interior of the right valve. Periostracum on the left valve extends beyond its length and covers the interior of the right valve. B. The section through the right valve with a 10-100 μm-thick cavity (in black) initially filled by the single (main) conchiolin layer. Po 3 (core M14) at 40-45 cm. C. The section through the right valve with 10-100 μm-thick cavities present within the main and other conchiolin layers (mcl – main conchiolin layer, scl - secondary conchiolin layer, tcl - tertiary conchiolin layer)). Po 3 (core M13) at 65-70 cm.

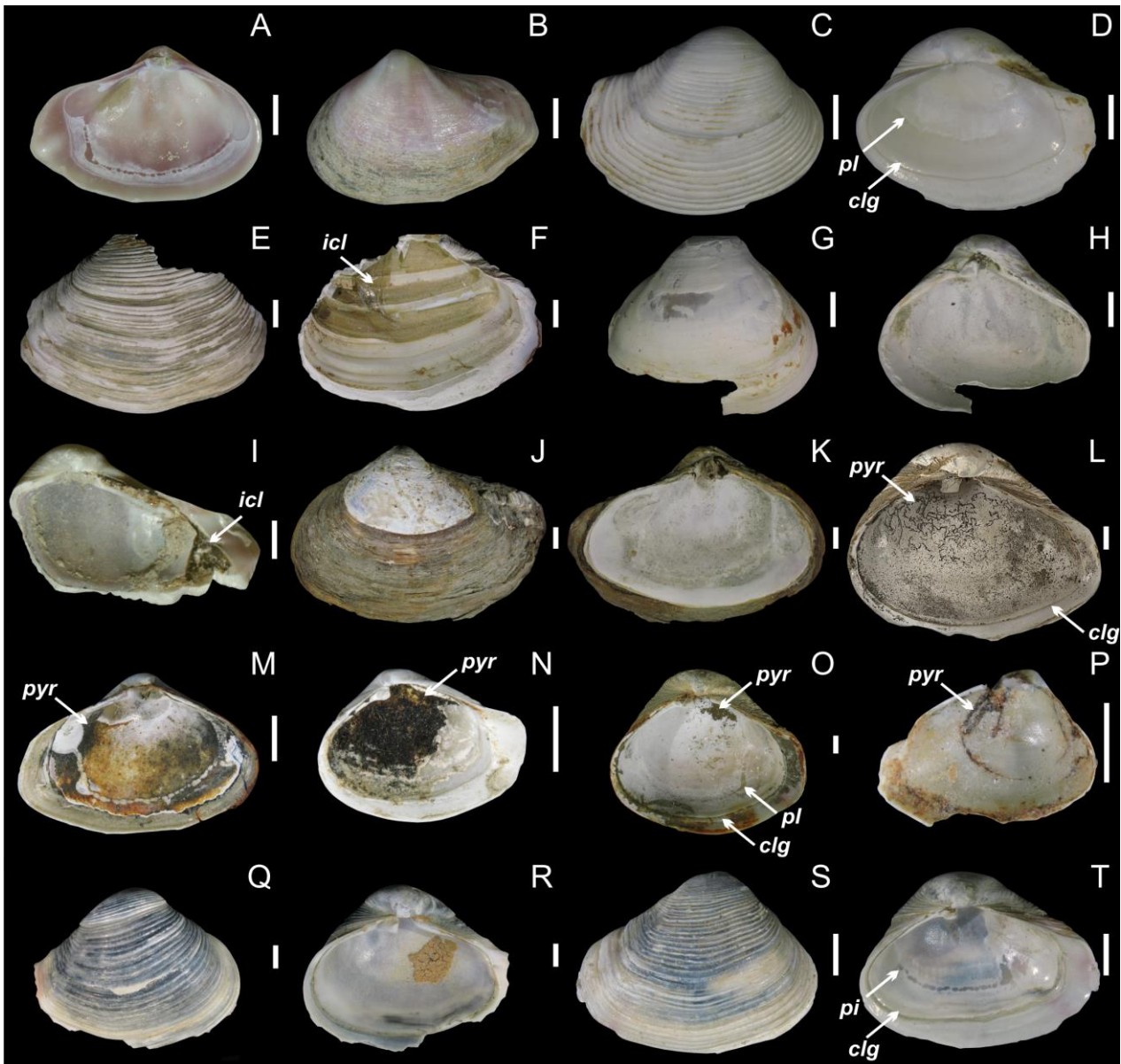

**Figure 5** - *Varicorbula gibba* from Po and Panzano (characterized by high sedimentation rates), showing the variability in external and internal preservation, with well-preserved valves with (A-B) or without periostracum (C-D) and with relicts of the main conchiolin layer (E-F). The interiors of some specimens are lined by pyritic framboids that form isolated and dispersed framboids (K), strings (L), and patchy or continuous coatings on the conchiolin layer (M-P). These specimens tend to be associated with brownish grains and streaks and with surface internal dissolution. Some specimens have blueish color (Q-T). A-B. Left valve, M28-65-70 cm-29,C-D. Right valve, M21-90-95 cm-1. E-F. Outer layer of the right valve, M13-65-70 cm-25. G-H. Inner layer of the right valve, M13-20-25 cm-8, I. Internal conchiolin layer exposed in a fragment, M20-120-125 cm. J-K. Left valve with periostracum and finely dispersed pyrite framboids on interior, M21-2-4 cm-2, L. Interior of the right valve with strings of pyrite framboids, M21-90-95 cm, M. Interior of the left valve lined by pyrite framboids, M28-110-115 cm-22, N. Interior of the right valve almost continuously lined by pyrite framboids, M28-110-115 cm-14, O. Interior of the right valve lined by patches of pyrite framboids, M21-30-35 cm-29, P. Interior of the left valve with strings of oxidixed pyrite framboids, M28-125-130 cm-21, Q-R. Blue-stained right valve, M28-85-90 cm-1. S-T. Blue-stained right valve, M21-105-110 cm-4. Note: clg – conchiolin groove, pl – pallial line, pyr – framboidal pyrite, icl - internal conchiolin layer. Scale bar: 1 mm.

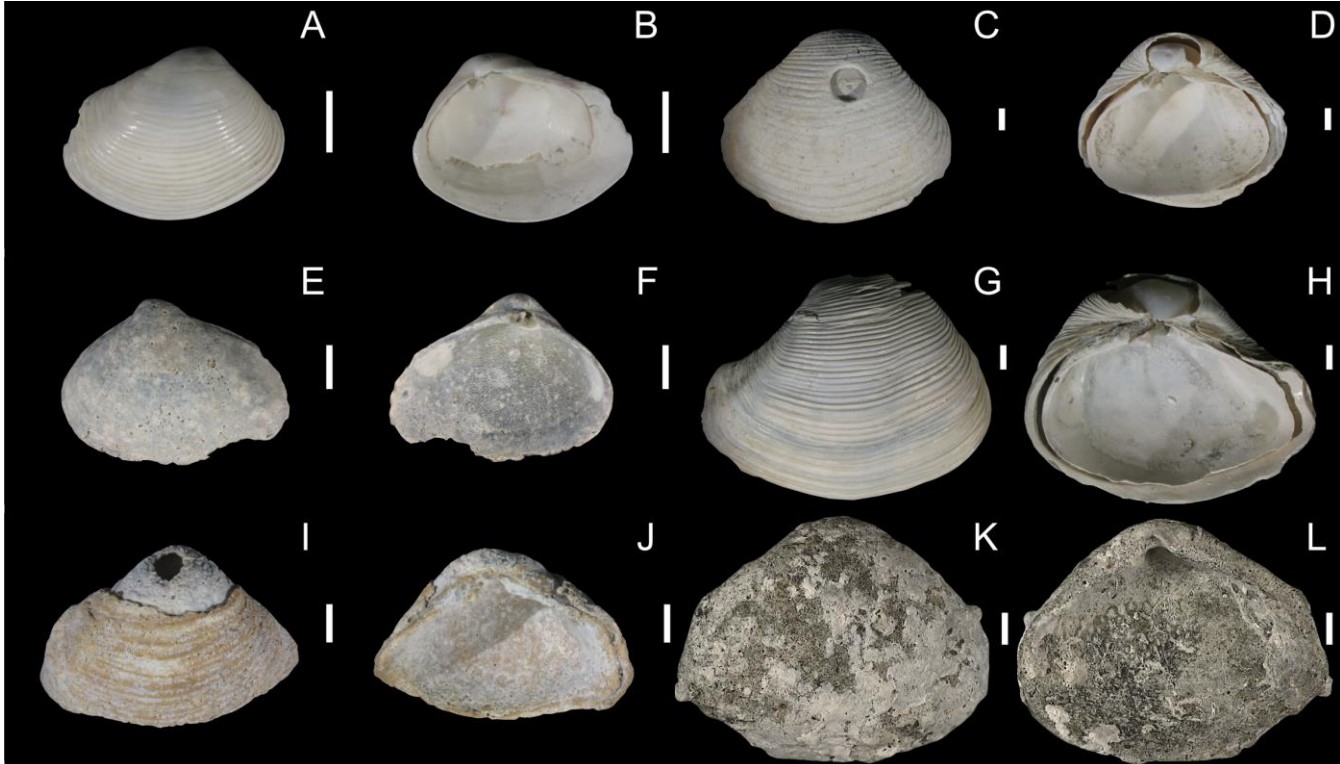

**Figure 6** – *Varicorbula gibba* from a Piran and Brijuni (sites with slow sedimentation rates) with worn, bored, encrusted and stained valves. A-B. M53-0-2 cm-7, C-D. M53-0-2 cm-6, E-F. M53-0-2 cm-4, G-H. M53-100-105 cm-1, I-J. M53-0-2 cm-2, K-L. M44-140-145 cm. Scale bar: 1 mm.

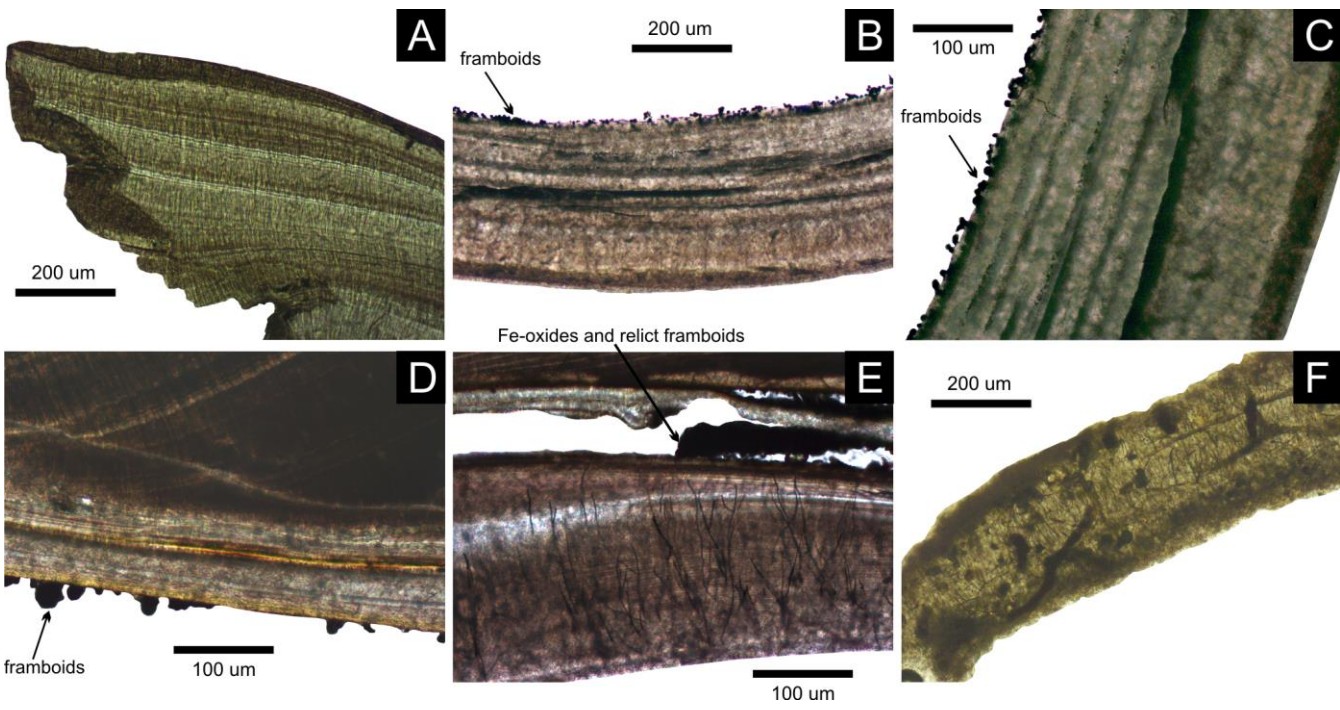

**Figure 7 –** Thin-section photographs showing pyrite framboids on the inner surface of *V. gibba* and preservation of more altered valves. A – A pristine right valve not lined by pyrite framboids, M13-65-79 cm-29. B-C - Internal surface of well-preserved right valve lined by pyrite framboids (arrows), M21-90-95 cm. D - Internal surface of well-preserved right valve lined by pyrite framboids, M14-40-45 cm. E – Internal conchiolin layer within the right valve, with a dark mixture of carbonate nodules and pyrite rimmed and replaced by Fe oxides, M20-120-125 cm. F – Strongly bored valve filled micrite and stained by nanopyritic inclusions, M44-140-145 cm.

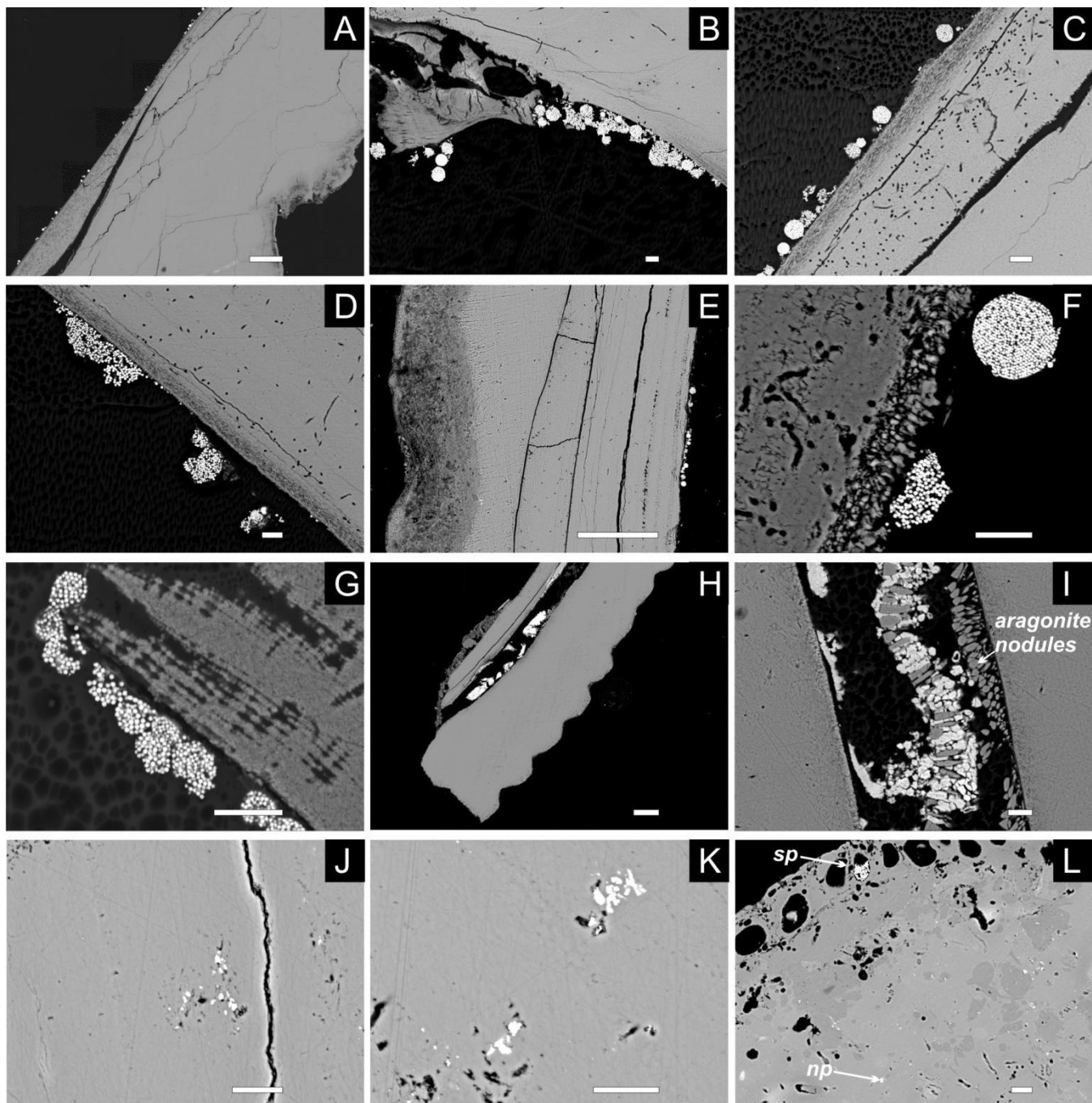

**Figure 8** - Backscattered electron images showing pyrite framboids on *V. gibba* valve interiors and in the main conchiolin layer in well-preserved valves and pyrite framboids and nanopyrite in microborings or microfractures in highly altered valves. A-D – Pyrite framboids attached to the interior surface, Po 3 (M14)-40-45 cm. E-F – Pyrite framboids attached to the interior surface, Po 4 (M21)-90-95 cm. G-I – Pyrite framboids on the interior and framboids (located between aragonite nodules) rimmed by Fe oxides in the main conchiolin layer, Po 4 (M20)-120-125 cm. J-K – a well-preserved specimen without microborings but with blueish staining contain nanopyritic inclusions (aggregation of micro-sized microcystals), Piran (M53)-100-105 cm-1. L – a dark-stained specimen with intense microborings, encrusters and cementation of pores and borings filled by framboids (sp - secondary pyrite lining), by nanopyritic inclusions (np), and by high-Mg calcite (darker infills), Brijuni (M44)-140-145 cm-2. Scale bars: A, E, H – 100 μm, B-D, F, G, I-L - 10 μm.

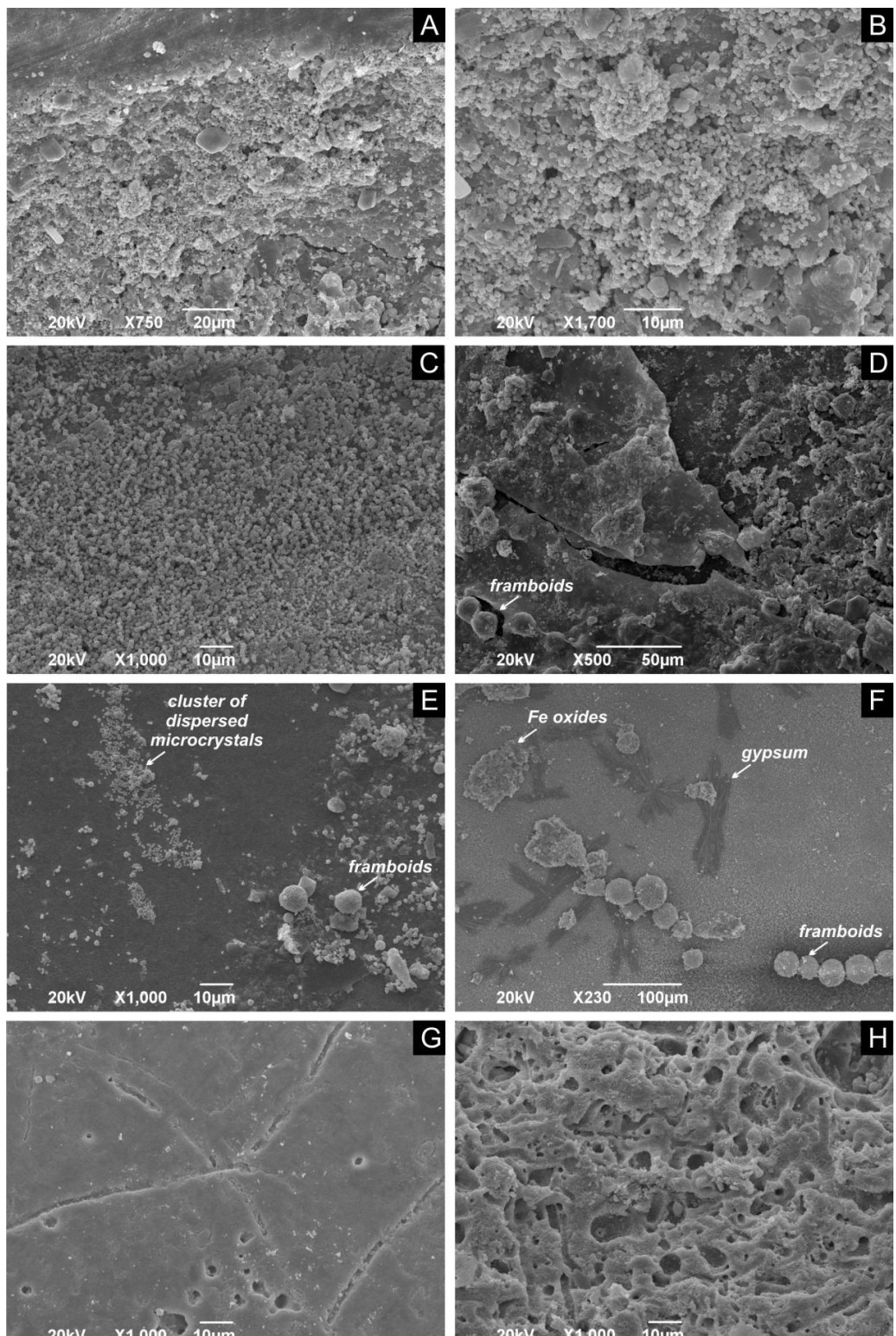

**Figure 9** – Internal surfaces of *V. gibba* with isolated and clustered framboids on well-preserved valves from Po and with intense borings on valves from Brijuni. A-B – Dispersed pyrite microcrystals and isolated framboids clustered around the groove at which the internal conchiolin crops out. C – Dispersed pyrite microcrystals close to the groove. D – Sheets of organics underlain by framboids on the interior of the right valve close to the ventral margin. E – Clusters of dispersed microcrystals and isolated framboids on the well-preserved ventral valve margin. F – Strings of pyrite framboids, irregular flakes of Fe oxides, and small gypsum crystals. G, H – moderate (G) and strong bioerosion (H). A-B - Po4-30-35 cm-28, D-F - Po4-30-35 cm-28, G - Brijuni-M44-70-75 cm-3, H - Brijuni-M44-70-75 cm-1.

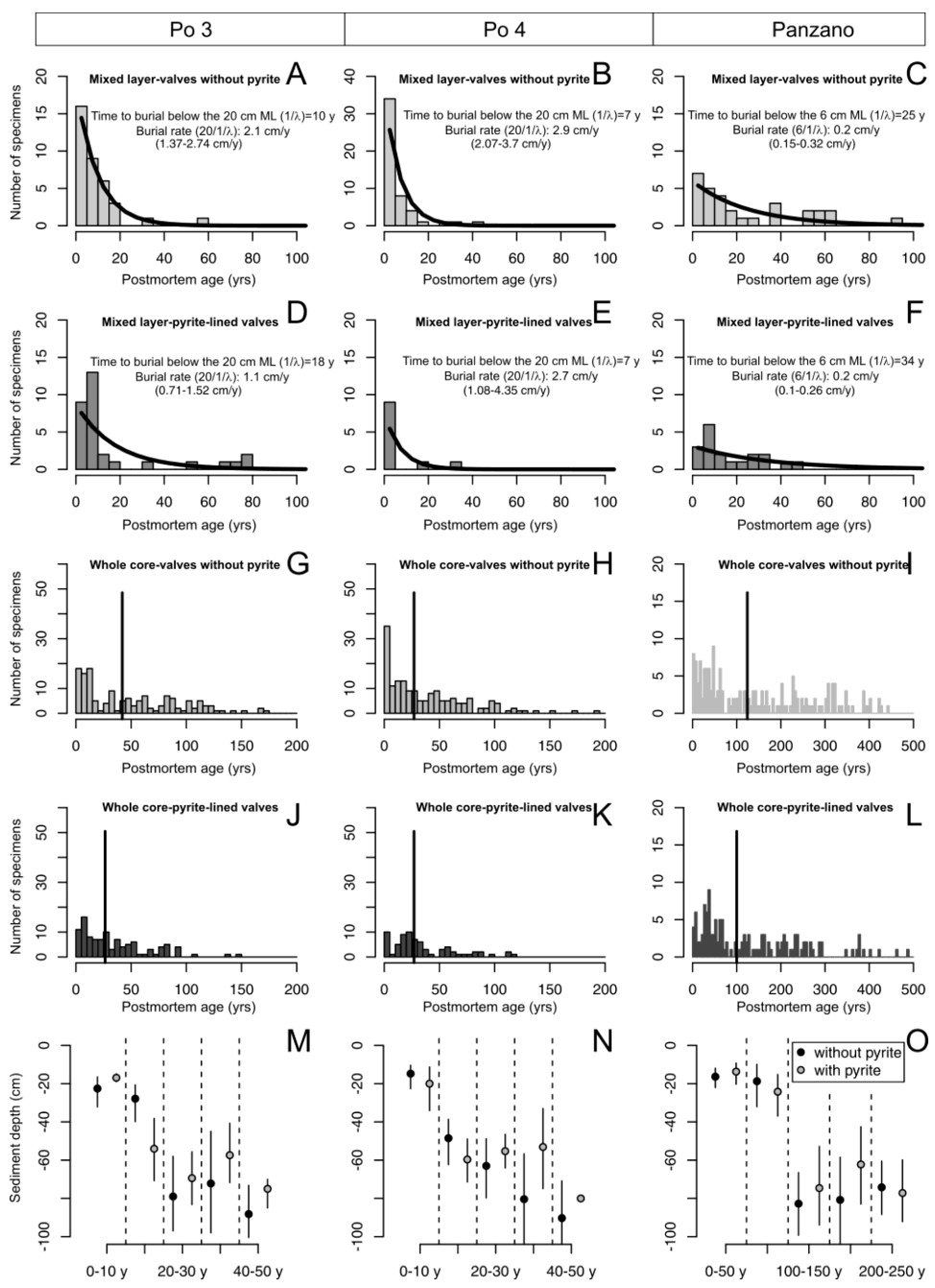

**Figure 10** – A-L. Mixed-layer (A-F) and whole core distributions (G-L) of postmortem ages of *V. gibba* from Po 3, Po 4, and Panzano with valves with and without pyrite show that first, even the youngest valves less than 5-10 years old have pyritic linings. Second, the mixed-layer distributions of valves with and without pyrite show similar right-skewed shapes well-fitted by a simple exponential model (black lines) that indicate that both types of valves are buried below the mixed

layer at a similar rate (burial rate corresponds to inverse of the mean age of the distribution, with 95% confidence intervals in parentheses). The burial rate estimated on this basis is congruent both with downcore changes in median shells ages and in [210]Pb-based estimates of sedimentation rate at Po and Panzano, indicating that the steepness of mixed-layer age distributions primarily correspond to burial rates. At the scale of whole cores (G-L), pyrite-lined valves exhibit a mode at ~25 years at Po 4 and a mode at ~50 years at Panzano. M-O. Mean sediment depths of valves with (gray) and without pyrite (black) of the

same age are similar. Both age and depth distributions indicate a scenario in which pyrite formation is not associated with deeper burial and occurs in reduced microenvironments in near-surface sediment zones.

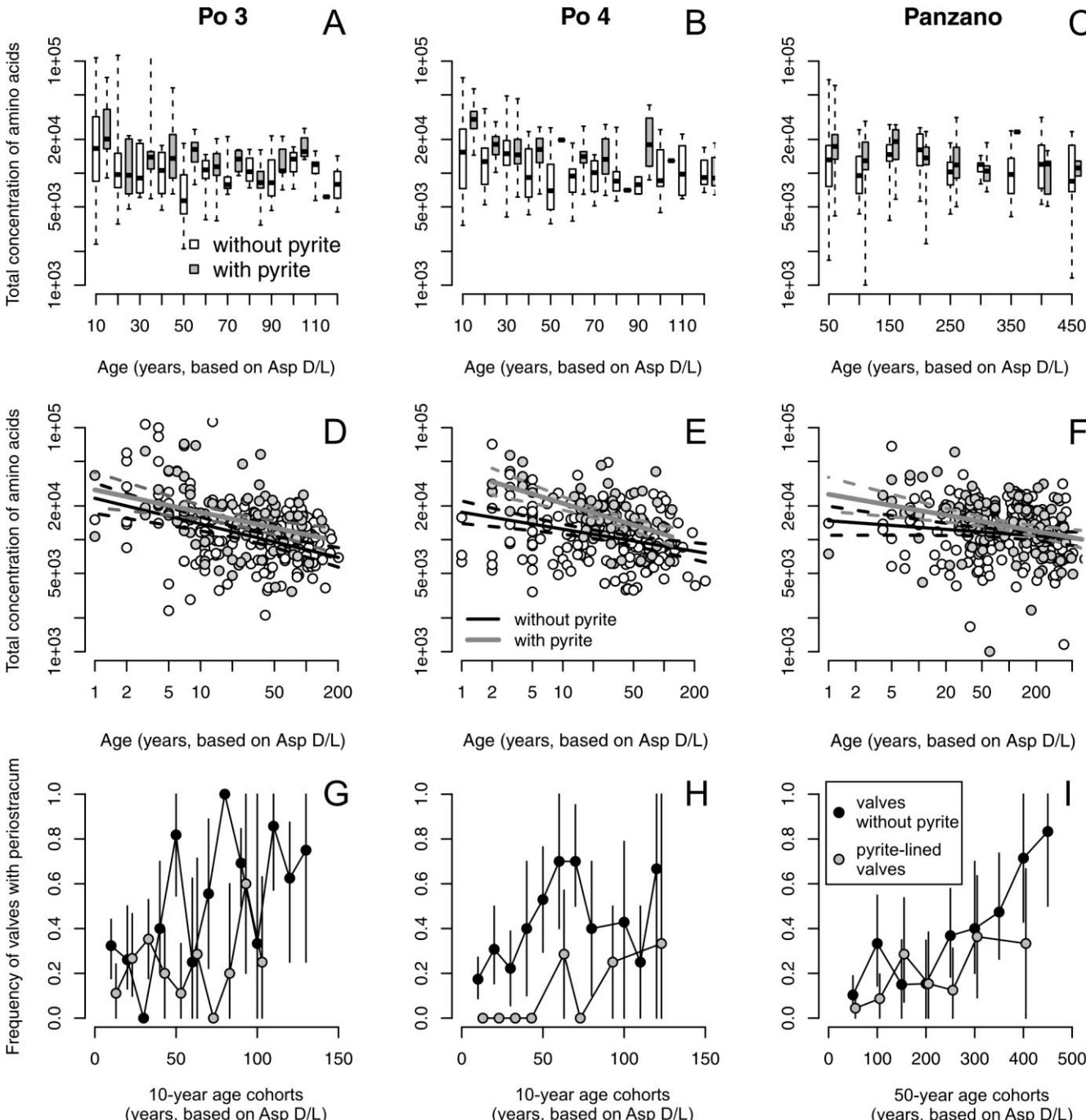

**Figure 11** – A-C. Equally-old valves of *Varicorbula gibba* lined by pyrite framboids tend to have higher concentrations of amino acids (visualized by boxplot pairs of valves with pyrite in gray boxes and without pyrite in white boxes), with age cohorts aggregated to 10 years at Po and 50 years at Panzano. D-F. Valves lined by pyrite framboids possess higher intercepts of the linear dependence of log-transformed amino acid concentrations on log-transformed postmortem age than valves without pyrite. G-I. Valves lined by pyrite framboids show higher mean frequency in preservation of periostracum or conchiolin layer than valves not lined by pyrite, with age cohorts aggregated to 10 years at Po and 50 years at Panzano.

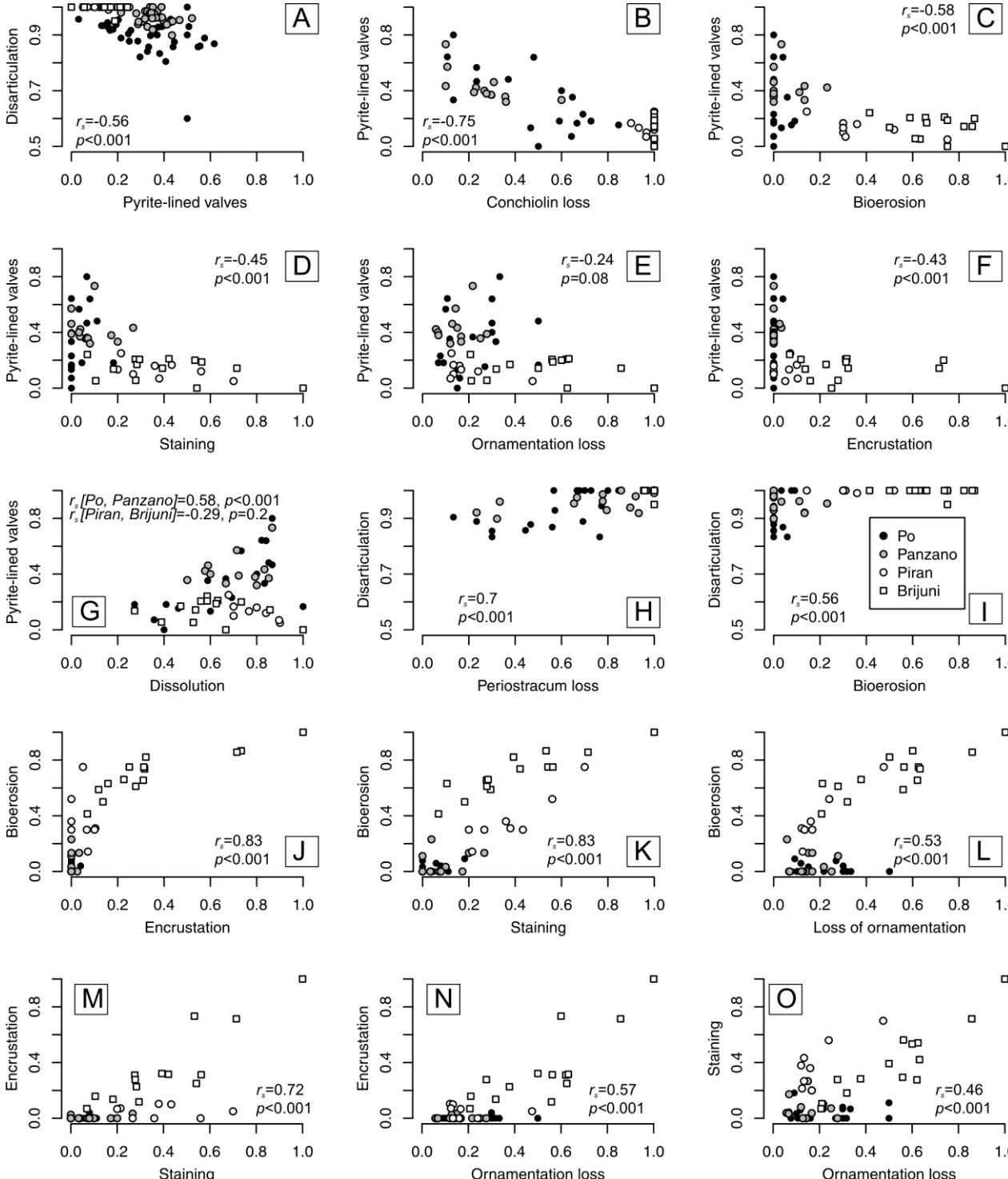

**Figure 12.** A-F. Negative relationships between per-increment frequencies of pyrite-lined valves and frequencies of other types of alteration ($r_s$ values refer to Spearman rank correlations). G. The relation between frequencies of pyrite-lined valves and fine-scale dissolution depends on the site location and the pathway that ultimately leads to dissolution (pyrite oxidation at Po and Panzano or bioerosion at other sites). Valves with pyrite commonly show internal fine-scale dissolution but are still associated with some surviving portions of periostracum or conchiolin layer. However, high frequencies of dissolved specimens are associated with low pyrite preservation as specimens from Piran and Brijuni are bored and dissolved but rarely associated with pyrite grains. H-L. Pairwise positive relationships between frequencies of disarticulation, conchiolin loss, bioerosion, encrustation, staining, and loss of ornamentation. M-O. Pairwise positive relationships between frequencies of encrustation, staining, and ornamentation loss.

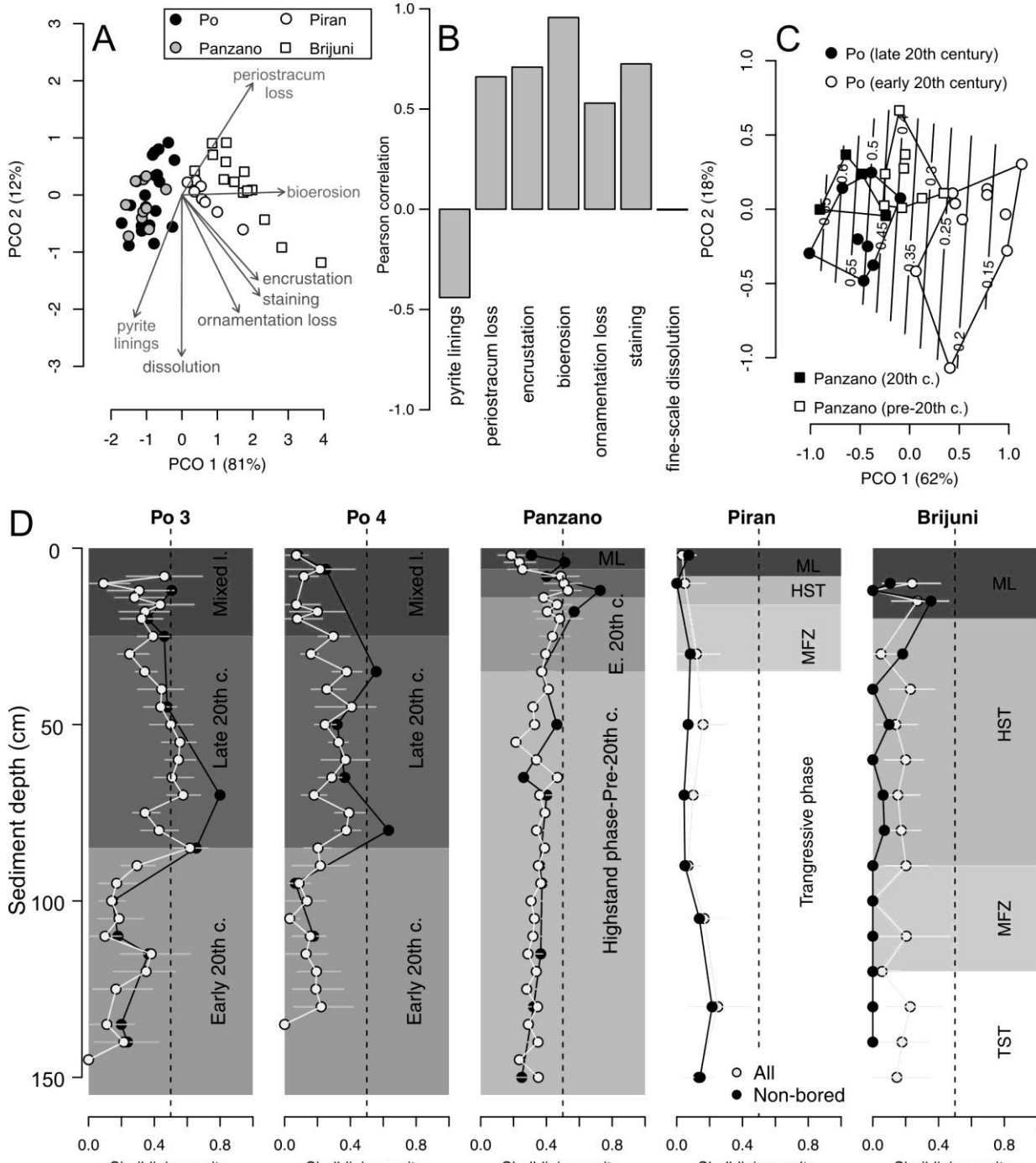

**Figure 13** – A. Principal coordinate analysis (PCO) based on *V. gibba* alteration (using seven variables) in increments from all sites (excluding increments from the mixed layer). B. Correlations between PCO axis 1 and eight alteration variables show that pyrite varies inversely with other alteration variables. C. Principal coordinate analysis of increments Po and Panzano, showing the separation between the latest HST (white circles) and the late 20[th] century increments (black circles), mainly determined by the frequencies of pyrite-lined specimens (contours). D. Stratigraphic changes in the frequencies of all pyrite-lined valves (gray circles) of *V. gibba* peak at 40-50% in the upper part of cores at Po 3 and Po 4 and at 50% at ~10-20 cm at Panzano. The frequencies of pyrite-lined valves are typically less than 20% at Piran and Brijuni where sedimentation rates are slow. The frequencies of pyrite-lined valves at Piran and Brijuni further decline when the frequencies of pyrite-lined valves are limited to valves not affected by bioerosion (black circles), magnifying the contrast between well-preserved pyrite-lined valves at sites with high sedimentation (primary linings) and variably-preserved pyrite-lined valves at sites with slow sedimentation (with some pyrite corresponding to secondary linings).

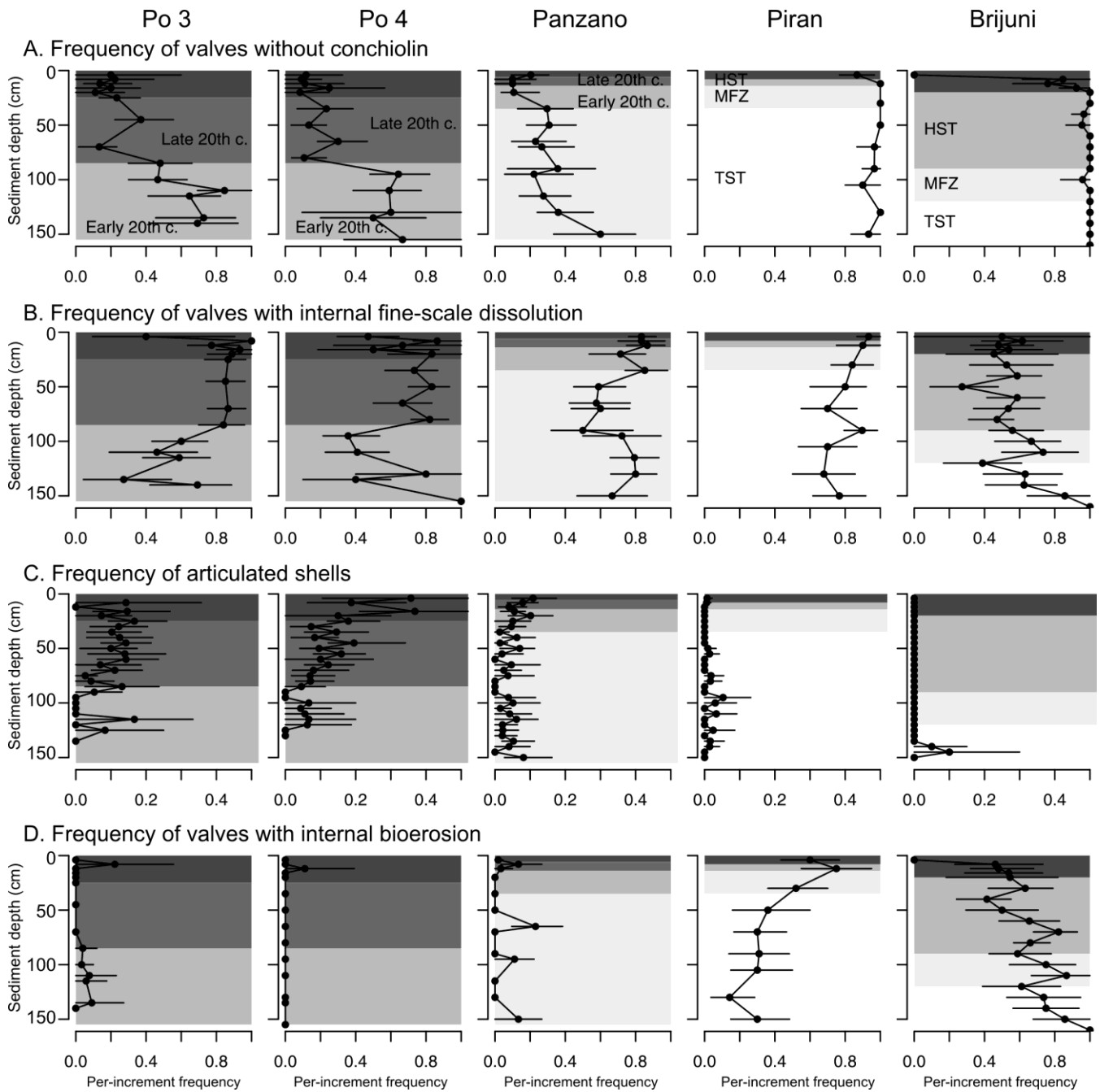

**Figure 14** – Stratigraphic changes in the per-increment frequency of valves of *Varicorbula gibba* affected by the loss of conchiolin (A, external periostracum and internal conchiolin), by internal surficial fine-scale dissolution (B), characterized by the presence of articulation (C), and affected by internal bioerosion (D). TST – transgressive systems tract, MFZ – maximum flooding zone, HST – highstand systems tract.

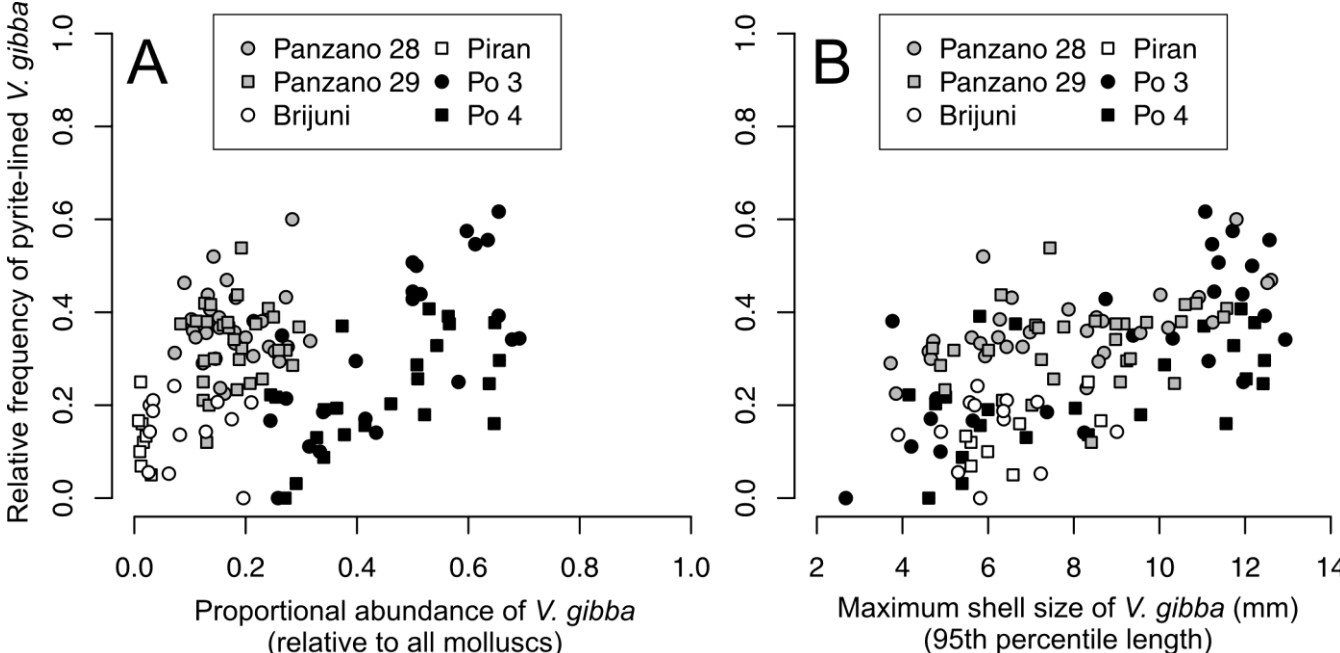

**Figure 15** – A. The relationship between the frequency of *V. gibba* valves lined by pyrite framboids and *V. gibba* proportional abundance per increment is positive at the Po stations (black symbols). Within-site relationships are positive whereas regional-scale relationships are more heterogeneous and noisy owing to between-site differences in time averaging. B. The relationship between the frequency of *V. gibba* valves lined by pyrite framboids and *V. gibba* maximum shell size per increment is positive within all sites. These relationships are interpreted to reflect the response of ecological and taphonomic processes as responding together to higher hypoxia frequency and a narrower extent of aerobic respiration in sediment (but with continuing bio-mixing), possibly reduced bioirrigation during the prolonged recovery, leading to smaller food and space limitation and thus to higher abundance and size of *V. gibba* in the wake of hypoxic events and simultaneously to higher frequency of pyrite linings.

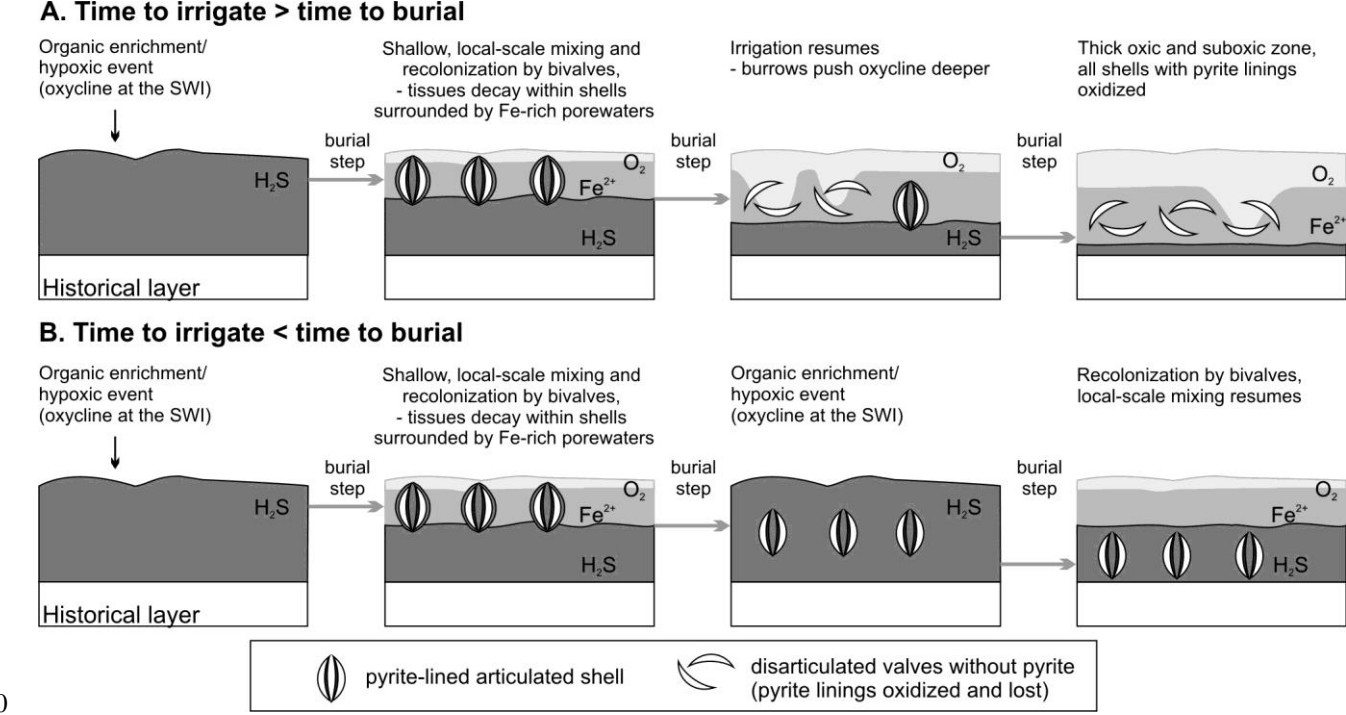

**Figure 16** – Two pathways characterized by differences in the frequency of hypoxic events (leading to the local extinction of infaunal organisms) and by differences in recovery rate of infaunal communities with high irrigation potential. As bivalves colonize and die at shallow sediment zones, both pathways promote the initial formation of reduced pockets, surrounded by iron-rich porewaters induced by local-scale mixing during the early recovery phase. A. In the aftermath of hypoxia, pyrite

linings in reduced pockets are formed initially in mixed sediments, but these are recycled under later recovery of irrigating organisms that oxidize deeper sediment zones and contribute to disintegration of initially articulated shells. B. If the frequency of hypoxic events is high, the oxycline remains permanently shallow because the recovery of infauna with high bioirrigation potential remains patchy, and some subset of shells with pyrite linings is thus not oxidized. Therefore, shells with pyrite linings can escape from the mixed layer into the subsurface stratigraphic record in B. In the historical layer,

valves permanently remain in reducing conditions.

**Table S1** – The results of fitting age-frequency distributions (for valves with and without pyrite linings) from the mixed layer at three sites to a simple disintegration model (with temporally-constant loss rate by disintegration and/or burial from the mixed layer; independent estimates of net sedimentation rates indicate that this parameter primarily corresponds to burial rate) and to a sequestration model with three parameters. AICc - Akaike Information criterion corrected for sample size.

| | Simple disintegration model (1 parameter) | | | | Sequestration model (3 parameters) | | | | |
|---|---|---|---|---|---|---|---|---|---|
| | Sample size (n) | lambda (loss from mixed layer) | Neg. log-likelihood | AICc | lambda1 (early loss from mixed layer) | lambda2 (late loss from mixed layer) | tau (sequestration) | Neg. log-likelihood | AICc |
| Po 3-without pyrite | 36 | 0.1026 | 118.0 | 117.7 | 0.1176 | 0.0457 | 0.0037 | 238.1 | 242.2 |
| Po 3-with pyrite | 31 | 0.0558 | 120.5 | 116.8 | 0.1324 | 0.0233 | 0.0078 | 243.1 | 240.4 |
| Po 4-without pyrite | 50 | 0.1441 | 146.9 | 146.5 | 0.1674 | 0.0680 | 0.0061 | 295.8 | 299.5 |
| Po 4-with pyrite | 11 | 0.1358 | 33.0 | 32.5 | 0.2222 | 0.0723 | 0.0206 | 68.4 | 74.4 |
| Panzano-without pyrite | 30 | 0.0396 | 126.9 | 126.9 | 0.0421 | 0.0393 | 0.0963 | 255.9 | 260.7 |
| Panzano-with pyrite | 21 | 0.0296 | 94.9 | 91.7 | 0.0588 | 0.0080 | 0.0013 | 192.0 | 190.8 |