# Peer review of "Pyrite-lined shells as indicators of inefficient bioirrigation in the Holocene-Anthropocene stratigraphic record"

_Biogeosciences, 2021_

## Referee Comment (RC3)

[referee-annotated manuscript omitted]

---

## Author Comment (AC2)

**REVIEWER 2**
This is an original manuscript on a highly relevant topic. I am sure that we will soon witness an explosion in this style of papers. This study is definitely timely. My comments are all minor and aimed towards improving the clarity of the argument, pointing in places to additional background literature. The authors highlight the role of deltaic systems on sedimentary dynamics, noting the higher sedimentation rates in the prodelta in comparison with areas in the northwest Adriatic Sea. Some cores are coming from prodelta settings (Po and Isonzo) and others from areas off strandplains. Deltaic settings are characterized by a complex array of stressors. It is clear that sedimentation rate is a first-rate controlling factor in this area. However, what about other potential factors, such as freshwater discharge, hypoxia, or substrate consolidation in connection with a deltaic source? Some of these (e.g. hypoxia) are assessed through the text, but a better articulation with the deltaic context would be advisable.

**Response: We thank the reviewer for all comments. Benthic communities in deltaic environments of the northern Adriatic Sea are affected by a combination of natural and anthropogenic impacts, although it seems that the present-day states are mainly determined by the effects of eutrophication, hypoxia and trawling, and tend to be dominated by species adapted to sediment disturbance (as *V. gibba*) or by mobile species. However, other limits on distribution – salinity, turbidity, and whether the location is below or above the thermocline – are also important. In the Discussion, we have expanded on the ecological (as opposed to the taphonomic) effect of sedimentation rate as follows:** "*High sedimentation rates typical of deltaic environments can limit mixing and irrigation via high substrate instability (MacEachern et al., 2005; Bhattacharya et al., 2020), and benthic communities at Po Delta are affected by short-term seasonal variability in sediment input and reworking (Ambrogi et al., 1990; Paganelli et al., 2012). However, water depths at ~20 m at Po prodelta are probably largely beyond the reach of proximal deposition of thicker flood deposits (Tesi et al., 2012) and thus less affected by substrate instability, and tend to be mainly limited by the frequency of hypoxic events (Crema et al., 1991; Simonini et al., 2004; Tomašových et al., 2020).*"

A discussion on other influences on pyritization would be useful as well. For example, the higher abundance of pyritized shells is present in nearshore areas where restricted circulation may have been associated with lower oxygen content. Also, these are areas with higher amounts of organic carbon and iron in the fine-grained sediment. In particular, bioturbation is strongly affected by the interplay of these parameters.

**Response: We have revised the manuscript so that it is clear that the focus is on linings formed by clustered pyrite framboids (rather than on pyritization in general). Although the oxygen depletion (owing to limited water circulation or other causes) is certainly one of the key factors in reducing bioirrigation and in fostering the preservation of pyrite linings by reducing its potential to re-oxidation, persistence of bio-mixing can induce the formation of suboxic zones with dissolved iron (especially in organic-rich muds) – this factor is probably necessary to avoid iron limitation and thus to ensure the confinement of H$_2$S produced during the decay of organic tissues. To clarify our reasoning, we have significantly expanded our Discussion of conditions that are needed for the formation and preservation of pyrite linings (persisting sediment mixing and the lack of intense and deep bioirrigation) as follows:**
"*The preferential occurrence of pyrite linings in valves with periostracum and with higher concentrations of amino acids and the negative correlation of pyrite-lined valves with the frequencies of other types of alteration at the scale of increments indicate that pyrite framboids*

[revised manuscript text omitted]

*Depending on the frequency and duration of hypoxic events or other disturbances limiting benthic fauna, two preservation scenarios can be envisioned (Fig. 16): (i) If the frequency of hypoxic events is low and infaunal organisms efficiently mixing and irrigating sediments rapidly recover, articulated shells will disarticulate and disintegrate when exposed to scavengers, borers and degradation of organics in TAZ. For example, maceration of the conchiolin layer triggers delamination of valves into inner and outer layers (Fig. 5E-H). In such conditions, the thickness of the aerobic zone within the mixed layer will increase (Fig. 16A). First, pyrite framboids will not nucleate on valves anymore if the labile biomass and microbes coating the decaying tissues were degraded during the earlier phase of decomposition. Second, bacterial sulfate reduction in microenvironments surrounded by iron-dominated pore waters triggers initial formation of pyrite linings, oxidation induced by bioirrigation will catch up with pyrite-lined shells prior to their deep burial and thus inhibit their subsurface sequestration. (ii) If the frequency of hypoxic events is high relative to the recovery time of the burrowing community with efficient bioirrigators, some subset of pyrite-lined shells can remain preserved if recovery of bioirrigation-inducing burrowers is slow or communities with infrequent bioirrigators are locked by hysteresis effects in an alternative stable state even when hypoxic conditions abated, and $O_2$ penetration depths thus remain close to the sediment-water interface (Fig. 16B). However, the recovery of deposit and detritus feeders that still mix sediments support the formation of the ferruginous suboxic zone (van de Velde and Meysman, 2016), thus allowing the precipitation of pyrite framboids at location of decaying molluscs. These conditions with reduced bioirrigation and limited iron recycling are typical of eutrophied environments (Karlson et al., 2007; Lehtoranta et al., 2009). We suggest that such conditions with limited bioirrigation but persisting mixing characterize present-day benthic ecosystems inhabiting muddy sediments in the northern Adriatic Sea. If the*

*nutrient-fueled eutrophication or other sources of sediment organic enrichment lead to permanent anoxia of bottom waters and high sulfate reduction, the concentrations of pyrite framboids within shells will be prohibited because sulfide production by bacterial sulfate reduction in organic-rich sediments can exceed in-situ availability of reactive iron oxides, $H_2S$ diffuses and precipitates elsewhere (Raiswell and Berner, 1985; Schenau et al., 2002; Raiswell et al., 2008; Farrell et al., 2009). Iron limitation can be driven by iron bounded to framboids linked to disseminated organic matter under high sediment organic enrichment but also to release of ferrous iron from sediments to water column (if water column is not sulfidic, Pakhomova et al., 2007). In addition, as mentioned above, mixing of sediments by burrowers that underlies the iron-based redox shuttle is aborted under anoxic conditions, and iron-limitation in pore waters is thus further enhanced by the lack of bioturbation. The absence of pyrite linings in anoxic sediments documented in the deep-time stratigraphic record (e.g., Brett et al., 1997) indicates that in marine environments with persistent bottom-water anoxia, the window for the early and rapid formation of pyrite linings (e.g., within shells of nektonic groups as cephalopods that fell on the anoxic seafloor) will be closed when organic matter degradation in surface sediments leads to the excess of hydrogen sulfide."*

There are various papers published on this topic during the last fifteen years or so. I suggest, for example, to check MacEachern, J. A., Bann, K. L., Bhattacharya, J. P., 2005. Ichnology of deltas: Organisms' responses to the dynamic interplay of rivers, waves, storms and tides. In: Giosan, L., Bhattacharya, J. P. (Eds.), River Deltas: Concepts, Models, and Examples. SEPM Special Publication, 83, 49–85. Also of relevance is: Bhattacharya, J.P., Howell, C.D., MacEachern, J.A. and Walsh, J.P., 2020. Bioturbation, sedimentation rates, and preservation of flood events in deltas. Palaeogeography, Palaeoclimatology, Palaeoecology, 560, p.110049. In short, the proposed interpretation relies heavily on sedimentation rates, but bringing other parameters to the discussion would be important to reflect more adequately the complex dynamics of deltaic systems.

**Response: As we have mentioned above, although the effect of sedimentation rate is important, it is probably not sufficient for formation of pyrite linings (i.e., clusters of framboids associated with the decay of larger organic tissues, as opposed to disseminated pyrite). We think that the condition of reduced bioirrigation is probably still needed for subsurface preservation of pyrite linings. It can be also argued that the condition of iron limitation is in fact best resolved when the sediment is just mixed but not intensely irrigated.**
**In the Discussion, we have added: "***However, although high background sedimentation rate (or rapid episodic burial) is to some degree a necessary condition for preservation of pyrite linings, we argue below that it is not a sufficient condition, and that potential for reoxidation must be also reduced by disturbances (such as hypoxia) limiting the functioning of irrigating infauna.***"**

In line 61, the classic paper in this regard is: Bromley, R.G. and Ekdale, A.A., 1986. Composite ichnofabrics and tiering of burrows. Geological magazine, 123(1), pp.59-65. **Thanks, we have added this reference.**

---

## Author Comment (AC3)

**REVIEWER 3**
This paper presents an exceptionally well documented case study of a phenomenon rarely seen in modern environments but common in the deep time fossil record: pyrite formation within shells. The authors present a wealth data on the relative frequency and association of pyrite with other taphonomic indicators of residence time and also provide data on the actual ages of the shells based on amino-acid racemization. The results provide a powerful case for the development and preservation of pyrite in closed spaces of dead organisms and shows that pyrite linings are most frequent in areas of higher sedimentation rates and slow mixing or bioirrigation. The authors even present data that indicate an upward increase in pyrite formation within shells of the later 20[th] century that parallels evidence for increased eutrophication owing to anthropogenic activity. In fact, the presence of pyrite may be a more sensitive indicator of sluggish rates of bioirrigation than ichnofabrics. The paper is extremely well written, well organized and thoroughly referenced. And supported by a large data set and thorough statistical analysis. I noted only a few minor errors in the text and references, which are marked on the pdf. I also have the following queries, mainly out of interest in the subject; I refer to relevant line numbers. The authors may wish to comment on them.

Line 75. Here and elsewhere, throughout, the term "microniches" is not quite right; perhaps "microenvironments" is preferable.
**Response: We thank the reviewer for all critical comments. We have replaced microniches with microenvironments (although "microniche" is frequently used in titles of several references about this topic).**

183, see also 613.: this paragraph well explains the tight closure of Varicorbula valves. But is there a reason that Varicorbula shells remain closed after death instead of splaying open at the hinge ligaments as do most bivalves? Does the ligament groove serve as a sort of locking devise? This may explain why they are most frequently pyrite lined. Perhaps a brief discussion of comparative taphonomy of these clams would be useful here.
**Response: We assume that these shells were located in the uppermost sediment zones in the upper 3 cm after their death, i.e., their disarticulation would be partly acting against sediment at their living locations. Although we did not perform experiments to check how much does the groove and the periostracum overlap increases the strength of articulation, it seems that these traits, including the option that the groove locks valves to some degree after the death, can provide some buffer against disarticulation. In addition, *V. gibba* possesses a relatively small internal ligament that, as in other members of the superfamily Myoidea, generates a small opening moment that is not sufficient to open valves against the sediment pressure (Trueman, 1954, Yonge, 1982).**
**We have added this information in the Discussion as follows:** *"Therefore, tightly-articulated shells of V. gibba with low opening moments of a small internal ligament that can be insufficient to open valves against the sediment pressure (as in other members of the superfamily Myoidea, Trueman, 1954; Yonge, 1982) and with internal conchiolin layers and can be intrinsically susceptible to the formation of reducing conditions."*

180s and 480s: Alternatively, do these clams frequently perish within their burrows as opposed to many that seem to rise to the surface during mortality. Under such conditions, burial within sediment may not require obrution, but I do not think this applies to most other articulated and closed bivalves let alone completely articulated multi-element skeletons.

**Response: It would be great to know an answer to this question – laboratory experiments showed that during an anoxic/hypoxic event,** *V. gibba* **emerges to the surface, as most other invertebrates. However, when the source of mortality is not related to oxygen depletion, and is rather related to predation (qualitative observations indicate that this is one source of mortality as many articulated shells are also drilled), then we assume that the scenario with the initial location occurring in the sediment is possible. As we mentioned above, several intrinsic traits and/or this within-sediment effect could ultimately reduce probability of disarticulation and thus lead to relatively high frequencies of articulated shells, although it is difficult to disentangle these effects at this stage (as many of these shells are in the uppermost cm, we think that articulation cannot be explauned by rapid burial to deeper sediment levels by some episodic event).**

319-320: how is it that pyrite formation does not set in until after 10 years in the sediments? If, as assumed the development of sulfides is associated or at least initiated with decay of organic matter contained within the enclosed spaces of shell cavities. But one would guess that such OM should be largely gone after just a year or so. Is the issue that the initial sulfides are monosulfide gels that only later recrystallize to recognizable pyrite?
**Response: This part refers to median ages, not to minimum ages, and the mode is younger than 10 years - primarily indicating yearly scales of pyrite formation – in other words, the pyrite formation can most likely occur almost immediately after the death of V. gibba. In Discussion, we have added: "***Age distributions of valves with and without pyrite linings show right-skewed shapes in the mixed layer at prodelta sites, with median age equal to 7-10 (without pyrite) and 7-18 years (with pyrite linings) at Po (in the upper 20 cm) and to 11 (without pyrite) and 15 years (with pyrite linings) at Panzano (in the upper 6 cm), respectively (Fig. 10A-F). They are dominated by recentmost cohorts younger than 10 years.***"**

349: it seems odd that shells with periostracum preservation should be negatively correlated with pyrite linings as one would suspect both might indicate higher rates of burial and reduced decay.
**Response: This was our mistake, the frequencies of pyrite-lined valves correlate positively with the frequencies of valves with periostracum correlate positively (negatively with the frequencies of valves without any periostracum). We have revised this sentence as follows by revising "without any relicts":** *"First, per-increment frequencies of pyrite-lined valves rank correlate negatively with the frequencies of disarticulated shells (Fig. 11A), with the frequencies of valves without any relicts of periostracum or conchiolin layer (Fig. 11B-C),..."*

385: this result: increased borings in the HST vs. TST seems paradoxical as sedimentation rates are normally predicted to increase into the later HST. Any explanation?
**Response: The temporal dynamic in accommodation/sediment supply ratio in the NE Adriatic with low sediment input (with the exception of sites close to river input) and sediment winnowing (and shedding towards deeper basinal part) is complex and sediments deposited during the latest highstand phase were at some locations affected by strong winnowing (related to the counterclockwise current system that fully developed in the northern Adriatic Sea during the highstand phase), leading to very thin/condensed sediments (and the input of clastic was partly compensated by in situ production of heterozoan carbonate particles). At regional scales, the highstand phase is thus represented by a thin condensed wedge – that is locally even thinner than the transgressive portion. However, the transgressive units tend to show retrogradation at large scales, whereas the**

**thin HST units tend to show aggradation. In Methods, we have added this information that can clarify this complexity:** *"We note that the net sedimentation rates at Piran and Brijuni (affected by negligible clastic sediment input and by significant contribution of in situ heterozoan carbonate production) were very slow and did not increase relative to the condition during the transgressive phase (it did not lead to the formation of prograding sediment bodies during the highstand phase)."*

Is it also possible that anthropogenic activity is related to increased rates of runoff and therefore, faster burial of organisms?

**Response: This scenario is possible to some degree at Pirand and Brijuni where the 210Pb profiles uppermost 10-20 cm show that the 20[th] century sediments , but long-term estimates of sediment accumulation rates do not indicate this scenario at Po and at Panzano. In Methods, we have expanded our description of temporal variability in sedimentation rate within sites as follows:** *"Upcore changes in median shell age show that sedimentation rates moderately oscillated through time and do not show any increase in the uppermost levels that correspond to the late 20[th] century. The within-site variability in sedimentation rates is smaller than the marked variability among sites. Sedimentation rates fluctuated between ~1 and 2.4 cm/y during the 20[th] century at Po sites (Tomašových et al., 2018) and between ~0.2-1 cm/y over the past 500 years at Panzano (Gallmetzer et al., 2017; Tomašových et al., 2017)."*

Lines 443-445. This is a very interesting finding in line with observations of iron sulfide blackened shells in ancient assemblages in which there is a strong positive correlation between other taphonomic indicators of long residence time and darkening:  Kolbe, S., Zambito, IV, J.J., Brett, C.E., Wise, J.L., Wilson, R.D., 2011. Brachiopod shell discoloration as an indicator of taphonomic alteration in the deep-time fossil record, Palaios 26: 682-692.

**Response; We have added to the Discussion:** *"A similar relationship with darkened specimens enriched in minerals possessing iron sulfides being more damaged by disarticulation, bioerosion and encrustation  in contrast to specimens less affected by discoloration was observed by Kolbe et al. (2011) in Ordovician brachiopods."*

480-onward. The result that shells of the same age may or may not show pyritization does not seem necessarily to support the contention that the pyritization occurred gradually during normal burial rather than during burial events. It is certainly possible that pulses of burial could entomb not only live or recently dead individuals, but many others that had died and decayed prior to the event. Pyrite would be localized in those that were buried with soft tissues intact whereas the "dead shells" would show little tendency toward sulfate reduction or pyrite formation.

**Response: We expect that this scenario with burial pulses would actually generate two distinct age distributions of valves with and without pyrite – with valves without pyrite characterized by older age (and thus slower burial rate), i.e., they spend longer time close to the sediment-water interface or within the mixed layer than living or recently-dead shells. To clarify our reasoning, we have added this statement in the Discussion:** *"A single episodic burial pulse that mixes living or recently-dead shells (with decaying biomass and potential for framboid formation) with older shells (without biomass) will generate one age distribution of valves with pyrite linings dominated by younger cohorts and another age distribution of valves without pyrite linings dominated by older cohorts whereas Under mixing, age distributions will be smoothed by stochastic movement of valves, but age distributions of valves wit pyrite linings*

*should be steeper and their median ages should be still lower, in contrast to our observations both at Po and Panzano."*

**When episodic burial occurs frequently, temporal accuracy of dating methods can affect this inference. However, this scenario with episodic burial would also propagate to generally higher sedimentation rate in the upper parts of sediment cores at Po and Panzano where pyrite-lined valves are more frequent – but net sedimentation rates do not increase upwards.**

**We ultimately prefer the scenario without rapid burial (and we find it useful to attempt to use the alternative represented by low bioirrigation, even when conceptually these two scenarios are not mutually not exclusive) because individual flood events occur at decadal scales (thus infrequent relative to ages of pyrite-lined valves that seem to form annualy), their frequency did not change during the 20$^{th}$ century, and the mixing that still occurs would make the initial phase of rapid burial ultimately unimportant because shells reworked up would be oxidized. We have revised and expanded this part in the *Discussion as follows:* "*Distinct layers deposited by major decadal floods preserved in cores at Po prodelta (Tesi et al., 2012; Tomašových et al., 2018) may have triggered episodic burial of benthic communities, but similar flood-event layers were not detected at Panzano. However, first, the major discharge events that lead to the deposition of flood deposits occur at decadal scales at Po prodelta (seven events during the 20$^{th}$ century, Zanchettin et al., 2008), whereas age distributions of pyrite-lined valves indicate that pyrite linings form almost every year. The frequency of flood events did not increase during the 20$^{th}$ century, in contrast to the increasing frequency of pyrite-lined valves. Second, pyrite-lined valves younger than 10 years old occur in the uppermost 5-10 cm of the mixed layer close to the sediment-water interface. Their high abundance in the uppermost zones indicates that they do not represent transient valves that were just recently exhumed from deeper zones. Although mixing after the deposition of thin flood layers can rework some subset of valves upward, exhumation of initially-buried valves to oxic conditions would lead to removal of pyrite linings. Third, equally-old valves with and without pyrite linings occur at similar depths and age distributions of pyrite-lined valves and valves without pyrite at Po and Panzano are similar (Fig. 9A-F), indicating that valves with and without pyrite were buried below the mixed layer at the similar rate. A single episodic burial pulse that mixes living or recently-dead shells (with decaying biomass and potential for framboid formation) with older shells (without biomass) will generate one age distribution of valves with pyrite linings dominated by younger cohorts and another age distribution of valves without pyrite linings dominated by older cohorts whereas Under mixing, age distributions will be smoothed by stochastic movement of valves, but age distributions of valves wit pyrite linings should be steeper and their median ages should be still lower, in contrast to our observations both at Po and Panzano. Although multimodal whole-core age distributions and stratigraphic changes in abundance indicate that input rates of V. gibba were not constant over the duration of core deposition at all sites (especially increasing in abundance after~1950 AD, Tomašových et al., 2018), the burial-rate parameter based on the exponential model (assuming the temporally-constant input of V. gibba) can be realistic when based on the topcore increments characterized by yearly to decadal time averaging at Po and Panzano (i.e., these increments were deposited after the mid-20$^{th}$ century increase in dominance by V. gibba). The pyrite framboids thus did not preferentially form within valves that were rapidly buried deeper in sediments (either by new**

*sediment deposition or by burrowers) as predicted by the obrution scenario and rather precipitated in near-surface sediment levels naturally inhabited by V. gibba."*

559 - this observation (of loss of Fe from sediments under persistent anoxia) is very important, as it may help explain the absence of pyritized fossils in truly black, laminated sediments that we have documented repeatedly (see Brett, C.E., Dick, V.B. and Baird, G.C.,1991. Comparative taphonomy and paleoecology of Middle Devonian dark gray and black shale facies from western New York. In Landing, E. and Brett, C.E., eds., Dynamic Stratigraphy and Depositional Environments of the Hamilton Group in New York Pt. II. State Museum Bulletin 469, p. 5â‘36.) One implication is that, while eutrophication and increased organic matter may enhance pyrite formation up to a point (because of lower benthic $O_2$ and reduced bioirrigation); too much OM may shut down the process.

**Response: We note that another subset of this dynamic is that mixing of particles by burrowers (i.e., the subset of bioturbation that does not necessarily irrigate the sediment) enhances the development of the suboxic iron-rich zone. To follow this comment, we have expanded the Discussion in 5.2 as follows: At the end of section 5.2., we have revised our discussion of anoxic scenario, as follows:** *"If the nutrient-fueled eutrophication or other sources of sediment organic enrichment lead to permanent anoxia of bottom waters and high sulfate reduction, the concentrations of pyrite framboids within shells will be prohibited because sulfide production by bacterial sulfate reduction in organic-rich sediments can exceed in-situ availability of reactive iron oxides, $H_2S$ diffuses and precipitates elsewhere (Raiswell and Berner, 1985; Schenau et al., 2002; Raiswell et al., 2008; Farrell et al., 2009). Iron limitation can be driven by iron bounded to framboids linked to disseminated organic matter under high sediment organic enrichment but also to release of ferrous iron from sediments to water column (if water column is not sulfidic, Pakhomova et al., 2007). In addition, as mentioned above, mixing of sediments by burrowers that underlies the iron-based redox shuttle is aborted under anoxic conditions, and iron-limitation in pore waters is thus further enhanced by the lack of bioturbation. The absence of pyrite linings in anoxic sediments documented in the deep-time stratigraphic record (e.g., Brett et al., 1997) indicates that in marine environments with persistent bottom-water anoxia, the window for the early and rapid formation of pyrite linings (e.g., within shells of nektonic groups as cephalopods that fell on the anoxic seafloor) will be closed when organic matter degradation in surface sediments leads to the excess of hydrogen sulfide."*

570: forgive my ignorance, but I do not understand your use of **hysteresis** here. Perhaps explain a bit.
**Response: Hysteresis corresponds to non-linear ecosystem dynamic that generates alternative stable states under the same environmental conditions. We have added parenthetically into the Discussion: "**... *can remain in a hysteresis state (non-linear ecosystem dynamic that generates alternative community states occur under the same environmental conditions, Duarte et al., 2015). For example, ecosystems subjected to eutrophication shift to a new state under some specific nutrient loading, but when recovering from eutrophication, the pre-eutrophication state is not established until the nutrient loading is reduced to a much lower level relative to the level that forced the shift initially. Some observations of recentmost oligotrophication in the northern Adriatic Sea (Mozetic et al.; Djakovac et al., 2012, 2015)*

*coupled with the continuing dominance of V. gibba in soft-bottom habitats indicate that the soft-botom communities, initially disturbed by high frequency of hypoxic events, is to some degree in hysteresis (Tomašových et al., 2020)."*

577: it is not intuitive that large size would be associated with lowered oxygen levels (resulting from eutrophication); in many cases low oxygen has been attributed to stunting and diminutive organisms. Apparently in Varicorbula the stunting affect is offset by increased growth. Presumably this is well documented; a reference might be useful.
**Response: The increase in size reflects the indirect effect of ecological release coupled with higher tolerance of V. gibba to short-term hypoxic or anoxic events. V. gibba is also negatively affected by anoxia, but this species is able to grow to larger sizes in the wake of those events, in contrast to conditions without any hypoxic events.**

613: but why do these bivalves not open shortly after death? If they are not buried rapidly (episodically) then how do they come to remain tightly closed?
**Response: Of course, shells buried themselves during their life, but we refer to the absence of additional postmortem burial. We assume that these shells were located in the uppermost sediment zones in the upper 3 cm after their death, i.e., at their living locations. *V. gibba* possesses a relatively small internal ligament that, as in other members of the superfamily Myoidea, generates a small opening moment that is not sufficient to open valves against the sediment pressure (Trueman, 1954, Yonge, 1982).**

Line 635: this also contrasts with nanopyrite infillings that seem to increase together with other signs of degradation with residence time.
**Response: yes, we have modified this sentence to**:*"Pyrite-lined valves thus represent a unique type of alteration that contrasts with other types of alteration (including the frequency of stained valves with nanopyritic inclusions) whose incidence increases with residence time in the taphonomic active zone"*

**Illustrations** are fine and require no modification.

**References** I have gone through and highlighted all that appear to be correctly cited; most are fine but there are a number of minor problems:
The following are **slightly out of alphabetical order:** Briggs et al. 1991., Boyle, Bjereskov, Degobbis. Meysman, Schieber 2012 (should come before Schieber and Baird)
**Response: Fixed**

**Wrong date**
Brush et al. 2020 (2021 intext) **Fixed to 2021**
Kralje et al. 2019 (2020 in text) **Fixed to 2019**
Slagter et al. 2021 (2020 intext) **Fixed to 2021**
**No date** (in ref list) Hunda et al. [2006] **Fixed**

**In reference list but apparently not referenced in text**:
Aller, 1982 **Added to main text**
Arcon et al. 1999 **Added to main text**
Clark ad Lutz, 1980 **Removed**

Eliott et al. 2007 **Added to main text as Elliott**
Faganeli et al 1985 **Added to main text**
Palinkas et al. 2007 **Added to main text as Palinkas and Nittrouer 2007**
Powell and Stanton 1985 **Removed**

**Cited in text but not in reference list:**
Alvizi et al. 2016 **Added to main text**
Stanton and Powell, 1985 **Removed from the main text**
Wignall et al., 2010 **Added to main text as Bond and Wignall 2010**

**Need to designate a and b references in text:**
Tomašovych 2019 a vs. b
**Response: Fixed**

**Overall:** this is an excellent paper, which needs only minor revision for publication. The paper provides much new data and reaches important conclusions regarding the little known phenomenon of early diagenetic pyritization; it has important implications for sedimentation rates and rates of bioirrigation, for the taphonomy of exceptional fossils, and potentially for conservation paleobiology and evidence for anthropogenic effects. The results will be of interest to geochemists, sedimentologists, taphonomists, paleobiologists and perhaps conservation paleobiologists. I strongly recommend publication with slight correction.

---

## Author Response (AR1)

Dear dr. Aninda Mazumdar,

below we respond to the comments of three reviewers (our comments in bold, text citations in italics) for the manuscript bg-2021-153. We have followed all recommendations of reviewers and revised the manuscript accordingly. We have shortened the title to reduce redundancy as follows: "Pyrite-lined shells as indicators of inefficient bioirrigation in the Holocene-Anthropocene stratigraphic record". We have also added Table S2 with original data and the R language script in the Supplement.

Thanks you very much for your consideration

Adam Tomasovych, on behalf of the coauthors

**REVIEWER 1**
In this manuscript, Tomašových et al. assess the preservation, pyritization and age and depth distribution of valves of the hypoxia-tolerant bivalve Varicorbula gibba in Adriatic Sea sediments. The authors compare these V. gibba taphonomic data to sedimentary and radiometric proxies for mixed layer depth and sedimentation rate, as well as sedimentary biogeochemical data. They conclude that sedimentation rate likely played a strong role in enhancing long-term pyritization of V. gibba valves (i.e., by limiting extents of oxidation in the uppermost sediment pile, and by shuttling valves below the mixed layer on relatively rapid time scales). The authors also observe that V. gibba valves with pyrite linings appear to be more prevalent in the portions of cores corresponding to the late 20$^{th}$ century, correlative with increases in seasonal hypoxic events and episodes of eutrophication. They therefore conclude that these hypoxic episodes, by deleteriously impacting the local infaunal community (particularly bioirrigators), directly resulted in valve-associated reducing microniches and decreased potential for reoxidation of valve pyrite linings, allowing burial of pyritized valves to outpace bioirrigator-mediated reoxidation in sites characterized by high sedimentation rates. With this work, the authors lend new insights into the role played by bioirrigation (and feedbacks between sedimentation and bioirrigation) and eutrophification in pyritization, with important implications for how pyritized fossils in the stratigraphic record can be used to reconstruct not only the taphonomy of body fossil assemblages but also changes in the extent and timescales of bioturbation. This manuscript represents an impressive body of work with potentially important implications for not only reconstructing environmental-ecological-taphonomy feedbacks on historic time scales but also in the deep-time stratigraphic record. I am therefore supportive of publication, and below I highlight a few relatively minor aspects that I have confidence the authors should be able to readily address in a revised version of the manuscript:
**Response: We thank the reviewer for taking the time to read the manuscript and for offering her comments and constructive criticisms.**

I would have liked to have seen more extensive discussion, up front (e.g., in the Introduction, the Methods or a new section of its own) of the ecology of Varicorbula gibba—for instance, whether it is infaunal, semi-infaunal or epifaunal; feeding ecology; seasonal variation in abundance; and ecological relationships to other local taxa. We are not told until near the end of the manuscript (l. 580-581) that V. gibba is among the assemblage of shallowly burrowing detritivores and deposit-feeders that have been previously documented at Adriatic prodelta sites such as Po and Isonzo. However, the ecology of V. gibba seems very relevant to the

manuscript's consideration of impacts of hypoxia and eutrophification on local benthic communities. For instance, is it an opportunistic taxon? Does its relative abundance in benthic communities actually increase under conditions of hypoxia—or in spite of being hypoxia-tolerant, are impacts of hypoxia on V. gibba, on the whole, deleterious (as they are inferred to be for the bioirrigating community)? This in turn could impact some of the authors' assumptions regarding rates of V. gibba input to the sediment pile.

**Response: We have followed this advice. We have added more information about the burrowing depth, feeding habits, lifespan and abundance of *Varicorbula gibba* into the Methods as follows (this information is then also used in the Discussion):**
*"We target specimens of the bivalve V. gibba because this species is common in all cores. V. gibba is a small-sized (<15 mm), shallow-infaunal, poorly mobile bivalve that has short siphons and thus lives in the upper 3 cm of sediment (Faresi et al., 2012), with the posterior margin located at or slightly below the sediment-water interface. It feeds on suspended phytoplankton but also exploits benthic diatoms, bacteria and organic detritus at the sediment-water interface (Yonge 1946). This species has higher tolerance to reduced oxygen levels relative to other molluscs and invertebrate groups (Holmes and Miller, 2006; Riedel et al., 2012) and can survive for several days and even weeks in anoxic conditions (Christensen 1970). In the northern Adriatic Sea, it increases in abundance in the wake of short-term, seasonal anoxic or hypoxic events to more than 1,000 individuals/m$^2$ (Hrs-Brenko 2006, Nerlović et al. 2011), and grows to 7-8 mm during the first year, achieving the maximum size of ~15 mm in two years (Hrs-Brenko 2003). It is classified as an opportunistic species in the assessments of benthic ecosystem health in the Mediterranean Sea (Simboura and Zenetos, 2002; Moraitis et al. 2018). Soft-bottom molluscan assemblages at water depths below the seasonal thermocline in the northern Adriatic became dominated by this species during the late 20$^{th}$ century Sea (Tomašových et al., 2018)."*

The authors explore various models for the distribution of V. gibba valves at depth (and from that exercise conclude that a model of uniform loss from the mixed layer for pyritized and non-pyritized valves is most parsimonious), but this appears to be premised upon an assumption of invariant input rates. If V. gibba abundance (or relative abundance) varies with bottom-water redox state and degree of nutrient loading this may not be a valid assumption, however. In addition to providing further detail on V. gibba ecology earlier in the manuscript, the authors should discuss these assumptions and the extent to which they can be constrained or justified.
**Response: Yes, estimates of loss rates on the basis of age distributions depend on the assumption that the temporal input of shells to death assemblages is relatively constant. However, although whole-core age distributions and stratigraphic changes in abundance of V. gibba indicate that input rates were not constant over the duration of core deposition, with abundance changes occurring at multi-decadal scales, this assumption is important over the time scale of residence time of shells in the mixed layer. The residence time is rather short at Panzano and Po (less than few decades, shorter than the time scale of the recentmost shift that occurred in the mid-20th century), and we thus assume that this assumption is not violated. To clarify this in the Results, we have added:** *"Although multimodal whole-core age distributions and stratigraphic changes in abundance indicate that input rates of V. gibba were not constant over the duration of core deposition at all sites (especially increasing in abundance after~1950 AD, Tomašových et al., 2018), the burial-rate parameter based on the exponential model (assuming the temporally-constant input of V. gibba) can be realistic when based on the topcore increments*

*characterized by yearly to decadal time averaging at Po and Panzano (i.e., these increments were deposited after the mid-20th century increase in dominance by V. gibba)."*

Similarly, in the manuscript's discussion of rates of "loss" of valves from the mixed layer (e.g., l. 201-204 and elsewhere), I suggest replacing use of the term "loss" (which is interpretive and connotates a null model that other processes—for instance, additive processes that may variable impact pyritized vs. non-pyritized valves, or those of different age 'cohorts'—do not lead to differences in abundance or distribution) and instead phrasing this in terms of relative abundance or distribution—at least prior to the Discussion section of the manuscript.

**Response: We have updated our description and terminology about "loss" rates. The term/parameter (lambda in the simple exponential model, reflecting the steepness of an age distribution) is just used initially in the Methods where the models are introduced. This parameter descriptively encompasses all mechanisms that lead to loss of valves from the mixed layer (by disintegration, transport, burial), regardless of whether they will be lined by pyrite. However, as independent estimates of sedimentation rates (210Pb-based and based on downcore changes in median ages) are equivalent to the value of "loss" parameter at Po and Panzano, we removed "loss rate" and just refer to this parameter as "burial rate" in the main text. To clarify this in Methods in section 3.2, we have added:** *"We estimate burial rates of valves below the mixed layer by fitting age distributions of valves with and without pyrite linings from Po and Panzano to two models that assume that the input of dead shells to the death assemblage is constant during their residence in the mixed layer (and thus over the duration of time averaging). The parameters estimated by these models are related to burial rate and depend on the steepness of age distributions in the mixed layer. However, they are also determined by the disintegration rate within the mixed layer (Tomašových et al., 2014). First, a simple model with temporally-constant loss rate of valves ($\lambda$) from the mixed layer (with loss occurring by disintegration and/or burial) predicts that age distributions can be well-fitted by the exponential distribution (disintegration-burial model). Second, a more complex (sequestration) model where loss rate declines from $\lambda_1$ to $\lambda_2$ with postmortem age at some sequestration rate $\tau$ predicts that the resulting age distributions are heavy-tailed, typically owing to exhumation of older valves to the sediment surface, with $\lambda_1$ corresponding to the disintegration rate of young valves, $\lambda_2$ corresponding to the reduced disintegration rate of older valves, and $\tau$ to the sequestration rate that can correspond to burial rate (Tomašových et al., 2014). The Akaike Information Criterion corrected for small sample size shows similar support for these two models at Po and Panzano both in valves with and without pyrite linings, with either a lower AICs for the simple model or a slightly higher support for the sequestration model that does not exceed 2-3 units, and small differences between $\lambda_1$ and $\lambda_2$ in the sequestration model. The $\lambda$ parameter estimated by the simple model at Po and at Panzano is similar both to $^{210}$Pb-based estimates of sedimentation rate and to $^{14}$C shell-based estimates (1-2 cm/y at Po and 0.2 cm at Panzano). Therefore, we infer that this $\lambda$ parameter at Po and Panzano corresponds to the burial rates and can be used to compare burial rates of valves with and without pyrite linings."*

I would also have liked to have seen additional information on sedimentation rates and mixed layer depths, specifically how these were constrained—given the importance of each of these to the authors' conclusions. For instance, it would be good to include the $^{210}$Pb data, which do not appear to currently be part of any of the figures or tables."

**Response: The excess activity in $^{210}$Pb data and shell ages calibrated by radiocarbon for all cores analyzed in this study were published in our former studies (with source data) that are referenced in Methods. To explain the data sources better, first, in Methods, we have added:** "*Core geochronology, the estimates of sedimentation rates, and the depth of the mixed layer are based (i) on the profiles in the excess activity of $^{210}$Pb and (ii) on the stratigraphic distribution of median bivalve shell ages based on amino acid racemization (AAR) calibrated by $^{14}$C published formerly in studies devoted to individual sites (at Panzano in Tomašových et al., 2017, at Po in Tomašových et al., 2018, at Piran in Mautner et al., 2018 and Tomašových et al., 2019b, and at Brijuni in Schnedl et al., 2018 and Gallmetzer et al., 2019).*"

**Second, to directly show the 210Pb profiles, age homogenization, and downcore changes in median shell ages, we have added a new figure 3 that summarizes these data in the upper parts of cores and thus provides information about the depth of the mixed layer and about net sedimentation rates based on both methods mentioned above. We have added this information about the definition of the depth of the mixed layer to Methods:** "*The estimates of sedimentation rates based on the slope of the $^{210}$Pb profiles below the mixed layer are similar to those based on downcore changes in median shell ages at both Po sites and at Panzano (Fig. 3). However, the $^{210}$Pb segments located below the fully-mixed layer at Piran and Brijuni are still steepened by biomixing (and thus overestimate sedimentation rates), as is typical of conditions when the rate of biomixing exceeds the rate of sedimentation (Johannessen and Macdonald 2012). The thickness of the surface well-mixed layer, based on homogeneity of median per-increment shell ages (amino acid racemization calibrated by $^{14}$C, Fig. 3A) is 20 cm at Po and at Brijuni, 5 cm at Panzano, and 8 cm at Piran (where a coarse skeletal shell bed occurs at 8-35 cm below the seafloor). The cores at Piran and Brijuni can be subdivided to units deposited during the transgressive phase characterized by rapid increase in accommodation space (transgressive systems tract, TST), during the time of maximum ingression (maximum flooding zone, MFZ), and during the highstand phase, characterized by very slow increase in accommodation space (highstand systems tract, HST, prior to the 20$^{th}$ century). The uppermost zones contain a mixture of late-highstand and 20$^{th}$ century sediments (Fig. 2). We note that the net sedimentation rates at Piran and Brijuni (affected by negligible clastic sediment input and by significant contribution of in situ heterozoan carbonate production) were very slow during the highstand phase and did not increase relative to the condition during the transgressive phase (i.e., it did not lead to the formation of prograding sediment bodies during the highstand phase). The core at Panzano captures about 500 years and the 20$^{th}$ century sediments occur in the upper 35 cm. The cores at Po consist of sediments deposited during the early and the late 20$^{th}$ century (and the earliest 21$^{st}$ century), as described by Gallmetzer et al. (2017) and Tomašových et al. (2018, 2019b). The thickness of the surface, well-mixed layer, based on the vertical extent of uniform or irregular segments of profiles in $^{210}$Pb excess is ~16 cm at Po, and 6 cm at Panzano, Piran, and Brijuni (Fig. 3B). With the exception of Brijuni, the estimates of the mixed-layer depths based on the $^{210}$Pb profiles and on the $^{14}$C-based shell age profiles are thus similar. At Brijuni, the 6 cm-thick mixed layer based on the $^{210}$Pb profiles relative to the 20 cm-thick*

*mixed layer based on $^{14}C$ probably reflects the shorter (multi-decadal) half-life of $^{210}Pb$ relative to the longer time needed to mix the upper 20 cm of sediment and their bioclasts."*

[Figure]

***Figure 3*** *– Downcore changes in shell ages (A) and the excess in $^{210}Pb$ profiles (B) that form the basis for inferences about the depth of the mixed layer (absence of downcore changes in median shell age in the uppermost parts of cores and irregular or uniform segments of the $^{210}Pb$ excess) and the sedimentation rate (thickness deposited over a duration defined by differences in median ages, and the slope of the $^{210}Pb$ segments below the mixed layer in gray color). We note that some variability in these estimates is also affected by biomixing when the thickness of sediments over which the deposition is measured is too low relative to the thickness of the mixed layer, leading to overestimation of the sedimentation rate.*

The authors state that sedimentation rates, although variable between the different sites and cores, appear to have been largely invariant throughout the deposition of individual cores (e.g., l. 266). It would be good to see that data upon which that assessment is based, as well as further discussion by the authors of whether this is surprising or expected for the prodeltaic sediments of their study sites over the hundred- to thousand-year time scales recorded by these cores.

**Response: Although over longer time scales, prograding and lobe-switching delta at Po led to high variability in sedimentation as documented over the past several centuries**

**(and is probably typical of other major prodeltas), shorter temporal duration preserved in our cores (~100 years at Po) were still characterized by relatively constant sedimentation rate. We have replaced "invariant" by "moderately-oscillating", with the key point that the variability in net sedimentation rates within cores at Po and Panzano was not trending and was smaller than variability between the cores. This within-core variability still led to decadal-scale residence times at Po and Panzano (as opposed to longer, millenial-scale residence times at Piran and Brijuni).**

**As mentioned above, the within-core variability was documented in our former studies based on downcore changes in median shell ages of 5 cm increments – we refer to these references in our statement now inserted into the Methods (see above), and the new figure shows this variability in the upper part of cores. In Methods, to better explain this variability, we have added:** *"Upcore changes in median shell age show that sedimentation rates moderately oscillated through time and do not show any increase in the uppermost levels that correspond to the late 20[th] century. The within-site variability in sedimentation rates is smaller than the marked variability among sites. Sedimentation rates fluctuated between ~1 and 2.4 cm/y during the 20[th] century at Po sites (Tomašových et al., 2018) and between ~0.2-1 cm/y over the past 500 years at Panzano (Gallmetzer et al., 2017; Tomašových et al., 2017). Sedimentation rates at Piran and Brijuni were persistently one or two orders of magnitude lower both during the transgressive and highstand phase (~0.01-0.02 cm/y) than at Po and Panzano (Tomašových et al., 2019b, 2021), and we thus refer to Po and Panzano as sites with high and to Piran and Brijuni as sites with low sedimentation rate. The estimates of sedimentation rates based on the slope of the $^{210}$Pb profiles below the mixed layer are similar to those based on downcore changes in median shell ages at both Po sites and at Panzano (Fig. 3)."*

**We have also added to the caption of Figure 3 that some variability in these estimates is also affected by biomixing when the thickness of sediments over which the deposition is measured is too low relative to the thickness of the mixed layer, leading to overestimation of sedimentation rate.**

There is also some ambiguity in the authors' discussion of the role of organic matter in fostering precipitation of pyrite linings on V. gibba valves. Pyritization is, as the authors acknowledge, typically limited by the supply of organic matter (as well a requiring a redox interface between iron and sulfate reduction at the localized supply of organic matter). However, something the authors do not directly discuss (though they perhaps allude to this in l. 529-532) but which is, in contrast, discussed by some of the studies they cite (e.g., Raiswell et al., 1993, Marine Geology; Farrell et al., 2009, Geology; as well as Raiswell et al., 2008, AJS) is that the presence of abundant disseminated organic matter in the sedimentary matrix tends to be detrimental to extensive pyritization of macroorganism carcasses. So although the hypoxic conditions fostered by eutrophification may, in the case of their Adriatic sediment samples, have played an important role in the development of a shallow redoxcline and thus pyrite precipitation on V. gibba, high rates of organic matter delivery to the seafloor are unlikely to foster extensive and exceptional fossilization of macroorganism carcasses via, for instance, pyrite templating or replacement in geologic analogues. In other words, early diagenetic precipitation of pyrite framboids on V. gibba valves under these conditions does not necessarily equate to exceptional pyritization—particularly given the abundance of sedimentary organic matter noted by the authors. The authors should therefore temper their

discussion of how their findings bear upon understanding of pathways of exceptional fossilization via pyritization, and incorporate discussion of these caveats.

**Response: We have adjusted the Discussion according to the comment, i.e., extensive organic enrichment will lead to the excess in H₂S, not confining it to the location of decay. We have thus removed the statements that may indicate that sediment organic enrichment is enhancing the formation of pyrite framboids within shells. First, although organic enrichment is associated with seasonal hypoxic events, it is rather the oxygen depletion that negatively affects bioirrigator activities. Second, sediment mixing (i.e., the second component of bioturbation, in addition to irrigation) by weakly-irrigating organisms still occurs in soft-bottom habitats, providing one mechanism for enrichment of porewaters in dissolved iron in the suboxic zone. This is also confirmed empirically (we mention the importance of iron reduction as observed in pore-water studies in the Setting) - even under sediment organic enrichment observed in the northern Adriatic Sea, uppermost sediment zones still show some iron reduction that provides dissolved iron for rapid formation of linings. We have added to Discusion this information at the beginning of section 5.2: "***Such organic enrichment can lead to porewater sulfidization and to exhaustion of highly reactive iron from porewaters, but organic-rich sites at Po prodelta still show high concentrations of dissolved iron in the uppermost few cm (Barbanti et al., 1995). Sediment mixing by (weakly-irrigating) infauna tends to counteract the exhaustion of dissolved iron and the potential buildup of H₂S in porewaters because burrowers transfer particles with iron oxides from the sediment-water interface to reducing conditions in subsurface zones (Faganeli and Ogrinc, 2009). In contrast to bioirrigation, mixing alone also contributes to higher oxygen consumption, thus further reducing the exposure of sedimentary particles to O₂ (van de Velde and Meysman, 2016).***"**

**At the end of section 5.2., we have revised our discussion of anoxic scenario, as follows:**
*"If the nutrient-fueled eutrophication or other sources of sediment organic enrichment lead to permanent anoxia of bottom waters, the concentrations of pyrite framboids within shells can be prohibited because mixing of sediments by burrowers that underlies the iron-based redox shuttle is aborted and sulfide production by bacterial sulfate reduction in organic-rich sediments can exceed availability of reactive iron oxides, with H₂S diffusing away (Raiswell and Berner, 1985; Schenau et al., 2002; Raiswell et al., 2008; Farrell et al., 2009), or ferrous iron can be released from sediments to the water column if the water column is not sulfidic (Pakhomova et al., 2007). The absence of pyrite linings in anoxic sediments documented in the deep-time stratigraphic record (e.g., Brett et al., 1997) indicates that in marine environments with persistent bottom-water anoxia, the window for the early and rapid formation of pyrite linings (e.g., within shells of nektonic groups such as cephalopods that fell on the anoxic seafloor) will be closed when the suboxic zone is not induced by biomixing and organic matter degradation in surface sediments leads to the excess of hydrogen sulfide (Middelburg and Levin, 2009)."*

Similarly, obrution in the typical sense need not involve deep burial, but rather rapid burial (and the associated 'smothering' of benthic communities).
**Response: Yes, we add "rapid (burial)" as a characteristic that is necessary for obrution. However, we stress that deep (rather than just shallow) burial by obrution is necessary to ensure that pyrite linings escape the re-oxidation that would follow in the wake of the obrution event (by re-colonizing burrowers) and thus for the long-term preservation and**

**transition of pyrite into the subsurface stratigraphic record. When mentioning the obrution pathway in the Discussion, we have added:** "*These early-sequestration conditions can be attained (i) when freshly-dead shells are episodically buried under a deposition of new sediment (obrution, Brett et al., 1997) sourced by river floods or storms, and can decay in reducing conditions beyond the reach of burrowers; the thickness of the obrution deposits needs to be sufficiently high so that pyrite is not reoxidized by burrowing infauna later...*"

**In section 5.2, we have expanded on our argument that rapid burial (with the exception of initial burial of bivalves to their living position below the sediment-water interface) is probably not necessary to explain the preservation of pyrite linings as follows:** "*The first scenario with obrution is frequently invoked in the deep-time stratigraphic record because it explains the short exposure of organic remains to $O_2$ and their rapid sequestration below the mixed layer into the historical layer (Brett et al., 2012a). Distinct layers deposited by major decadal floods preserved in cores at Po prodelta (Tesi et al., 2012; Tomašových et al., 2018) may have triggered episodic burial of benthic communities with V. gibba, but similar flood-event layers were not detected at Panzano. However, first, the major discharge events that lead to the deposition of flood deposits occur at decadal scales at Po prodelta (seven events during the 20[th] century, Zanchettin et al., 2008), whereas age distributions of pyrite-lined valves indicate that pyrite linings form continuously at yearly scales. The frequency of flood events did not increase during the 20[th] century, in contrast to the increasing frequency of pyrite-lined valves in the upper parts of cores at Po and Panzano. Second, pyrite-lined valves younger than 10 years old occur at high abundance in the uppermost 5-10 cm of the mixed layer close to the sediment-water interface, indicating that they do not represent transient valves that were just recently exhumed from deeper zones. Third, equally-old valves with and without pyrite linings occur at similar depths and their age distributions are similar (Fig. 9A-F), indicating that valves with and without pyrite were buried below the mixed layer at a similar rate. A single episodic burial pulse that mixes living or recently-dead shells (with decaying biomass and potential for framboid formation) with older shells (without biomass) will generate one age distribution of valves with pyrite linings dominated by younger cohorts and another age distribution of valves without pyrite linings dominated by older cohorts. Age distributions of valves with pyrite linings generated by such episodic burial should be steeper and their median ages should be lower relative to valves without linings, in contrast to our observations both at Po and Panzano. We thus suggest that the pyrite framboids did not preferentially form within valves that were rapidly buried deeper in sediments (either by new sediment deposition or by burrowers) as predicted by the obrution scenario and rather precipitated in near-surface sediment zones naturally inhabited by V. gibba.*"

Particularly if the redoxcline (due to hypoxic conditions) is located in the uppermost centimeters of the sediment pile, an "obrution scenario" and a "hypoxia-mediated reduced bioirrigation scenario" are entirely compatible, and should not be discussed as diametrically opposed alternative models (e.g., as in l. 475-484, l. 622-627).
**Response: Yes, these two scenarios are mutually not exclusive and can be complementary. However, in the non-obrution scenario, when infaunal shells die** *in situ* **and start decaying, then the obrution is not needed for the initial precipitation of linings formed by pyrite framboids. To ensure that such linings are not re-oxidized, the irrigation needs to remain limited (even when sediments are still affected by mixing of weakly-irrigating organisms), and there can be environmental conditions that generate such conditions (frequent disturbance and/or hysteresis after initial disturbance). This**

**scenario can be especially useful for depositional environments where storms or floods are not necessarily apparent or expected.**

**In the Discussion, first, we have thus simplified the two pathways and we have added that they can be complementary: (1) obrution (that would need to be sufficiently deep to avoid subsequent re-oxidation) and (2) without obrution, initiated just under a shallow burial of shells in their living position, but associated with subsequently low bioirrigation, as follows:** *"These early-sequestration conditions .... can be attained (i) when freshly-dead shells are episodically buried under a deposition of new sediment (obrution, Brett et al., 1997) sourced by river floods or storms, and can decay in reducing conditions beyond the reach of burrowers; the thickness of the obrution deposits needs to be sufficiently high so that pyrite is not reoxidized by burrowing infauna later (Allison, 1988; Brett et al., 2012a, b; Schiffbauer et al., 2014) and/or (ii) when biomass of decaying infaunal organisms is tightly enclosed within shells (as in V. gibba) or within burrows (Thomsen Vorren, 1984; Hansen et al., 1996), and the oxygen penetration by bioirrigation is shallow or intermittent (as can be typical of poorly-permeable fine-grained sediments), generating reducing microenvironments even without obrution (Jorgensen, 1977). Although these two pathways can be complementary, they may also act independently."*

**Second, when assessing their roles at our stations, we ultimately suggest that pyrite linings were not triggered by rapid (shallow or deep) obrution and that it is the lack of irrigation that is a crucial condition for final preservation of pyrite linings in soft-bottom deposits of the northern Adriatic Sea on the basis of age and depth distribution of pyrite-lined valves. Even if rapid burial would occur, the shallow burial alone would not protect linings from re-oxidization in well-irrigated sediments. In the Discussion we write:** *"However, first, the major discharge events that lead to the deposition of flood deposits occur at decadal scales at the Po prodelta (seven events during the 20[th] century, Zanchettin et al., 2008), whereas age distributions of pyrite-lined valves indicate that pyrite linings form continuously at yearly scales. The frequency of flood events did not increase during the 20[th] century, in contrast to the increasing frequency of pyrite-lined valves. Second, pyrite-lined valves younger than 10 years old occur in the uppermost 5-10 cm of the mixed layer close to the sediment-water interface. Their high abundance in the uppermost zones indicates that they do not represent transient valves that were just recently exhumed from deeper zones. Although mixing of sediments by burrowers after the deposition of thin flood layers can rework some subset of valves upward, significant exhumation of initially-buried valves to oxic conditions would lead to the removal of pyrite linings. Third, equally-old valves with and without pyrite linings occur at similar depths and their age distributions at Po and Panzano are similar (Fig. 9A-F), indicating that valves with and without pyrite were buried below the mixed layer at a similar rate."*

On a more minor note, previous studies of pyritization have suggested that relatively more recalcitrant organics may preferentially undergo pyritization (or more rapid pyritization) (e.g., Briggs et al., 1991, Geology; Raiswell et al., 1993, Marine Geology); the authors should therefore take care to not oversimplify fossil pyritization as targeting solely the most labile tissues.
**We have followed this and have removed "labile" and just refer to "organic tissues".**

l. 196: Although D/L amino acid ratios are of course broadly used, given that this paper may attract a broad audience (including those who do not commonly employ organic geochemical methods), I suggest providing further detail here.

**Response: We have added this statement into Methods, with references:** "*Ratios of D- and L-isomers (D/L) of eight amino acids (aspartic, glutamic, serine, alanine, valine, phenylalanine, isoleucine, and leucine), and their concentrations in valves of V. gibba were measured at Northern Arizona University using reverse-phase high-pressure liquid chromatography (RP-HPLC) and the procedures of Kaufman and Manley (1998). D/L ratios measure the extent of razemization and thus represent a geochronological tool (Kosnik et al., 2008; Allen et al., 2013).*"

l. 351-352: Please state here how sites were partitioned into "high-" and "low-" sedimentation rates (e.g., what range of values were used for this categorization).

**Response: We have revised our definition and characterization of sedimentation rates in Methods, where we have initially assigned the five stations effectively to these three categories, with the final split defined at the end:** "*The two Po cores are characterized by high sedimentation rate (~1-2 cm/y), the Panzano core was deposited under intermediate sedimentation rate (~0.2-0.4 cm/y, occassionally up to 1 cm/y), and the Piran (a 25 cm-thick skeletal shell bed occurs at 8 cm below the seafloor; Tomašových et al., 2019b) and Brijuni sites (a 20 cm-thick sandy mud in the core-top overlies a coarse bryozoan-rich molluscan muddy sand Tomašových et al., 2021) are sediment-starved (~0.01-0.03 cm/y). Upcore changes in median shell age show that sedimentation rates moderately oscillated through time and do not show any increase in the uppermost levels that correspond to the late 20th century. The within-site variability in sedimentation rates is smaller than the marked variability among sites. Sedimentation rates fluctuated between ~1 and 2.4 cm/y during the 20th century at Po sites (Tomašových et al., 2018) and between ~0.2-1 cm/y over the past 500 years at Panzano (Gallmetzer et al., 2017; Tomašových et al., 2017). Sedimentation rates at Piran and Brijuni were persistently one or two orders of magnitude lower both during the transgressive and highstand phase (~0.01-0.02 cm/y) than at Po and Panzano (Tomašových et al., 2019b, 2021), and we thus refer to Po and Panzano as sites with high sedimentation rate and to Piran and Brijuni as sites with low sedimentation rate.*"

**As pyrite linings are frequent at Po and Panzano stations and not at sediment-starved stations at Piran and Brijuni, we generally refer in the main text to Po and Panzano as "high-sedimentation" sites (i.e., above ~0.2 cm/y) and opposed them with "slow-sedimentation" sites at Piran and Brijuni. Although Po and Panzano stations still differ in sedimentation rate, we think that this subdivision still helps with the summarizing the pattern of pyrite linings.**

Figure 9: the burial rates calculated for A) and D) (without and with pyrite linings, respectively for Po 3) seem substantially different (by a factor of 2). It would be good to see additional interrogation of the grounds on which it was determined that these are essentially indistinguishable.

**Response: In Methods, we have referred now to two types of estimation of sedimentation rates – based on downcore changes in median shell age and based on 210Pb profiles. The estimates in figure 9 (now figure 10), however, refer to the third method that is fully based on the shape of age distributions (and is congruent with other methods used to determine sedimentation rates). We have added estimates of 95% confidence intervals**

**into Figure 10 to show that the within-site estimates of burial rate for valves with and without pyrite linings are comparable.**

**Technical Corrections**:
l. 97: "…may not be surprising…" **Fixed**
l. 171-172: This sentence contains two separate notations of the mixed layer depth at Brijuni—perhaps a typo? **Fixed**
l. 388: increments **Fixed**
l. 469: reach **Fixed**

For Figure 11 in particular (and, to a lesser extent, some of the other figures), the plots are so closely packed together that it is a little challenging to read the axis labels and attribute these to the appropriate plots. Could the panel components be spaced slightly further apart? **Response: We have spaced the insets and re-arranged the axis labels in Figure 11 and in other figures.**

Figure 15: For the A) label, is this supposed to be > (not <)? **Yes, fixed**

**REVIEWER 2**
This is an original manuscript on a highly relevant topic. I am sure that we will soon witness an explosion in this style of papers. This study is definitely timely. My comments are all minor and aimed towards improving the clarity of the argument, pointing in places to additional background literature. The authors highlight the role of deltaic systems on sedimentary dynamics, noting the higher sedimentation rates in the prodelta in comparison with areas in the northwest Adriatic Sea. Some cores are coming from prodelta settings (Po and Isonzo) and others from areas off strandplains. Deltaic settings are characterized by a complex array of stressors. It is clear that sedimentation rate is a first-rate controlling factor in this area. However, what about other potential factors, such as freshwater discharge, hypoxia, or substrate consolidation in connection with a deltaic source? Some of these (e.g. hypoxia) are assessed through the text, but a better articulation with the deltaic context would be advisable.
**Response: We thank the reviewer for all comments. Benthic communities in deltaic environments of the northern Adriatic Sea are affected by a combination of natural and anthropogenic impacts, although it seems that the present-day states are mainly determined by the effects of eutrophication, hypoxia and trawling, and tend to be dominated by species adapted to sediment disturbance (as *V. gibba*) or by mobile species. However, other limits on distribution – salinity, turbidity, and whether the location is below or above the thermocline – are also important. In the Discussion, we have expanded on the ecological (as opposed to the taphonomic) effect of sedimentation rate as follows: "***High sedimentation rates typical of deltaic environments can limit mixing and irrigation via high substrate instability (MacEachern et al., 2005; Bhattacharya et al., 2020), and benthic communities at the Po Delta are affected by short-term seasonal variability in sediment input and reworking (Ambrogi et al., 1990; Paganelli et al., 2012). However, habitats deeper than ~20 m at the Po prodelta are largely beyond the reach of proximal deposition of thicker flood deposits (Palinkas and Nittrouer, 2007; Tesi et al., 2012) and thus less affected by substrate instability, and tend to be mainly limited by the frequency of hypoxic events (Crema et al., 1991; Simonini et al., 2004; Tomašových et al., 2020).***"**

A discussion on other influences on pyritization would be useful as well. For example, the higher abundance of pyritized shells is present in nearshore areas where restricted circulation may have been associated with lower oxygen content. Also, these are areas with higher amounts of organic carbon and iron in the fine-grained sediment. In particular, bioturbation is strongly affected by the interplay of these parameters.

**Response: We have revised the manuscript so that it is clear that the focus is on linings formed by clustered pyrite framboids (rather than on pyritization in general). The oxygen depletion (owing to limited water circulation or other causes) is certainly one of the key factors in reducing bioirrigation and in fostering the preservation of pyrite linings by reducing its potential to re-oxidation. Persistence of bio-mixing can also induce the formation of suboxic zones with dissolved iron (especially in organic-rich muds) – this factor is probably necessary to avoid iron limitation and thus to ensure the confinement of H$_2$S produced during the decay of organic tissues. To clarify our reasoning, we have significantly expanded our Discussion of conditions that are needed for the formation and preservation of pyrite linings (persisting sediment mixing and the lack of intense and deep bioirrigation) as follows:**

[revised manuscript text omitted]

There are various papers published on this topic during the last fifteen years or so. I suggest, for example, to check MacEachern, J. A., Bann, K. L., Bhattacharya, J. P., 2005. Ichnology of deltas: Organisms' responses to the dynamic interplay of rivers, waves, storms and tides. In: Giosan, L., Bhattacharya, J. P. (Eds.), River Deltas: Concepts, Models, and Examples. SEPM Special Publication, 83, 49–85. Also of relevance is: Bhattacharya, J.P., Howell, C.D., MacEachern, J.A. and Walsh, J.P., 2020. Bioturbation, sedimentation rates, and preservation of flood events in deltas. Palaeogeography, Palaeoclimatology, Palaeoecology, 560, p.110049. In short, the proposed interpretation relies heavily on sedimentation rates, but bringing other parameters to the discussion would be important to reflect more adequately the complex dynamics of deltaic systems.

**Response: As we have mentioned above, although the effect of sedimentation rate is important, it is probably not sufficient for formation of pyrite linings (i.e., clusters of framboids associated with the decay of larger organic tissues, as opposed to disseminated pyrite). We think that the condition of reduced bioirrigation is probably still needed for subsurface preservation of pyrite linings. It can be also argued that the condition of iron limitation is in fact best resolved when the sediment is just mixed but not intensely irrigated. In the Discussion, we have added***: "****Although high background sedimentation rate (or rapid episodic burial) is a necessary condition for preservation of pyrite linings, we argue below that it is not a sufficient condition, and that potential for reoxidation must be also reduced by disturbances (such as hypoxia), which limit the functioning of irrigating infauna."*

In line 61, the classic paper in this regard is: Bromley, R.G. and Ekdale, A.A., 1986. Composite ichnofabrics and tiering of burrows. Geological magazine, 123(1), pp.59-65. **Thanks, we have added this reference.**

**REVIEWER 3**
This paper presents an exceptionally well documented case study of a phenomenon rarely seen in modern environments but common in the deep time fossil record: pyrite formation within shells. The authors present a wealth data on the relative frequency and association of pyrite with other taphonomic indicators of residence time and also provide data on the actual ages of the shells based on amino-acid racemization. The results provide a powerful case for the development and preservation of pyrite in closed spaces of dead organisms and shows that pyrite linings are most frequent in areas of higher sedimentation rates and slow mixing or bioirrigation. The authors even present data that indicate an upward increase in pyrite formation within shells of the later 20[th] century that parallels evidence for increased eutrophication owing to anthropogenic activity. In fact, the presence of pyrite may be a more sensitive indicator of sluggish rates of bioirrigation than ichnofabrics. The paper is extremely well written, well organized and thoroughly referenced. And supported by a large data set and thorough statistical analysis. I noted only a few minor errors in the text and references, which are marked on the pdf. I also have the following queries, mainly out of interest in the subject; I refer to relevant line numbers. The authors may wish to comment on them.

Line 75. Here and elsewhere, throughout, the term "microniches" is not quite right; perhaps "microenvironments" is preferable.
**Response: we have replaced microniches with microenvironments (although "microniche" is frequently used in titles of several references about this topic).**

183, see also 613.: this paragraph well explains the tight closure of Varicorbula valves. But is there a reason that Varicorbula shells remain closed after death instead of splaying open at the hinge ligaments as do most bivalves? Does the ligament groove serve as a sort of locking devise? This may explain why they are most frequently pyrite lined. Perhaps a brief discussion of comparative taphonomy of these clams would be useful here.
**Response: We assume that these shells were located in the uppermost sediment zones in the upper 3 cm after their death, i.e., their disarticulation would be partly acting against sediment at their living locations. Although we did not perform experiments to check how much does the groove and the periostracum overlap increases the strength of articulation, it seems that these traits, including the option that the groove locks valves**

to some degree after the death, can provide some buffer against disarticulation. In addition, *V. gibba* possesses a relatively small internal ligament that, as in other members of the superfamily Myoidea, generates a small opening moment that is not sufficient to open valves against the sediment pressure (Trueman, 1954, Yonge, 1982). We have added this information in the Discussion as follows: *"The tightly-articulated shells of V. gibba can be intrinsically susceptible to the formation of reducing conditions owing to (i) the overlapping periostracum, (ii) the internal groove that can lock valves to some degree, and (iii) low opening moments of a small internal ligament that can be insufficient to open valves against the sediment pressure (as in other members of the Myoidea, Trueman, 1954; Yonge, 1982)."*

180s and 480s: Alternatively, do these clams frequently perish within their burrows as opposed to many that seem to rise to the surface during mortality. Under such conditions, burial within sediment may not require obrution, but I do not think this applies to most other articulated and closed bivalves let alone completely articulated multi-element skeletons. **Response: It would be great to know an answer to this question – laboratory experiments showed that during an anoxic/hypoxic event, *V. gibba* emerges to the surface, as most other invertebrates. However, when the source of mortality is not related to oxygen depletion, and is rather related to predation (qualitative observations indicate that this is one source of mortality as many articulated shells are also drilled), then we assume that the scenario with the initial location occurring in the sediment is possible. As we mentioned above, several intrinsic traits and/or this within-sediment effect could ultimately reduce probability of disarticulation and thus lead to relatively high frequencies of articulated shells, although it is difficult to disentangle these effects at this stage (as many of these shells are in the uppermost cm, we think that articulation cannot be explained by rapid burial to deeper sediment levels by some episodic event).**

319-320: how is it that pyrite formation does not set in until after 10 years in the sediments? If, as assumed the development of sulfides is associated or at least initiated with decay of organic matter contained within the enclosed spaces of shell cavities. But one would guess that such OM should be largely gone after just a year or so. Is the issue that the initial sulfides are monosulfide gels that only later recrystallize to recognizable pyrite? **Response: This part refers to median ages, not to minimum ages, and the mode is younger than 10 years - primarily indicating yearly scales of pyrite formation – in other words, the pyrite formation can most likely occur almost immediately after the death of *V. gibba*. In the Discussion, we have added: "***Age distributions of valves with and without pyrite linings show right-skewed shapes in the mixed layer at prodelta sites, with median age equal to 7-10 (without pyrite) and 7-18 years (with pyrite linings) at Po (in the upper 20 cm) and to 11 (without pyrite) and 15 years (with pyrite linings) at Panzano (in the upper 6 cm), respectively (Fig. 10A-F). The increments in the mixed layer are thus dominated at these sites by recentmost cohorts younger than ~10 years."***

349: it seems odd that shells with periostracum preservation should be negatively correlated with pyrite linings as one would suspect both might indicate higher rates of burial and reduced decay. **Response: This was our mistake, the frequencies of pyrite-lined valves correlate positively with the frequencies of valves with periostracum (negatively with the frequencies of valves without any periostracum). We have revised this sentence as follows by revising "without any relics":** *"First, per-increment frequencies of pyrite-lined*

*valves rank correlate negatively with the frequencies of disarticulated shells (Fig. 12A), with the frequencies of valves without any relicts of periostracum or conchiolin layer (Fig. 12B-C),..."*

385: this result: increased borings in the HST vs. TST seems paradoxical as sedimentation rates are normally predicted to increase into the later HST. Any explanation?
**Response: The temporal dynamic in accommodation/sediment supply ratio in the NE Adriatic with low sediment input (with the exception of sites close to river input) and sediment winnowing (and shedding towards deeper basinal part) is complex and sediments deposited during the latest highstand phase were at some locations affected by strong winnowing (related to the counterclockwise current system that fully developed in the northern Adriatic Sea during the highstand phase), leading to very thin/condensed sediments (and the input of clastic was partly compensated by in situ production of heterozoan carbonate particles). At regional scales, the highstand phase is thus represented by a thin condensed wedge – that is locally even thinner than the transgressive portion. However, the transgressive units tend to show retrogradation at large scales, whereas the thin HST units tend to show aggradation. In Methods, we have added this information that can clarify this complexity:** *"The net sedimentation rates at Piran and Brijuni (affected by negligible clastic sediment input and by significant contribution of in situ heterozoan carbonate production) were very slow during the highstand phase and did not increase relative to the condition during the transgressive phase (i.e., prograding sediment bodies did not form during the highstand phase)."*

Is it also possible that anthropogenic activity is related to increased rates of runoff and therefore, faster burial of organisms?
**Response: This scenario is possible to some degree at Pirand and Brijuni where the 210Pb profiles of the uppermost 10-20 cm show that the 20th century sediments were deposited relatively rapidly, but long-term estimates of sediment accumulation rates do not indicate this scenario at Po and at Panzano. In Methods, we have expanded our description of temporal variability in sedimentation rate within sites as follows:** *"Upcore changes in median shell age show that sedimentation rates moderately oscillated through time and do not show any increase in the uppermost levels that correspond to the late 20th century. The within-site variability in sedimentation rates is smaller than the marked variability among sites. Sedimentation rates fluctuated between ~1 and 2.4 cm/y during the 20th century at Po sites (Tomašových et al., 2018) and between ~0.2-1 cm/y over the past 500 years at Panzano (Gallmetzer et al., 2017; Tomašových et al., 2017)."*

Lines 443-445. This is a very interesting finding in line with observations of iron sulfide blackened shells in ancient assemblages in which there is a strong positive correlation between other taphonomic indicators of long residence time and darkening: Kolbe, S., Zambito, IV, J.J., Brett, C.E., Wise, J.L., Wilson, R.D., 2011. Brachiopod shell discoloration as an indicator of taphonomic alteration in the deep-time fossil record, Palaios 26: 682-692.
**Response: Thanks, we have updated and added this information to the Discussion as follows:** *"A similar relationship with darkened specimens enriched in minerals possessing iron sulfides being more damaged by disarticulation, bioerosion and encrustation in contrast to specimens less affected by discoloration was observed by Kolbe et al. (2011) in Ordovician brachiopods."*

480-onward. The result that shells of the same age may or may not show pyritization does not seem necessarily to support the contention that the pyritization occurred gradually during normal burial rather than during burial events. It is certainly possible that pulses of burial could entomb not only live or recently dead individuals, but many others that had died and decayed prior to the event. Pyrite would be localized in those that were buried with soft tissues intact whereas the "dead shells" would show little tendency toward sulfate reduction or pyrite formation.

**Response: We expect that this scenario with burial pulses would actually generate two distinct age distributions of valves with and without pyrite – with valves without pyrite characterized by older age (and thus slower burial rate), i.e., they spend longer time close to the sediment-water interface or within the mixed layer than living or recently-dead shells. To clarify our reasoning, we have added this statement in the Discussion:** *"A single episodic burial pulse that mixes living or recently-dead shells (with decaying biomass and potential for framboid formation) with older shells (without biomass) will generate one age distribution of valves with pyrite linings dominated by younger cohorts and another age distribution of valves without pyrite linings dominated by older cohorts. Age distributions of valves with pyrite linings generated by such episodic burial should be steeper and their median ages should be lower relative to valves without linings, in contrast to our observations both at Po and Panzano. We thus suggest that the pyrite framboids did not preferentially form within valves that were rapidly buried deeper in sediments (either by new sediment deposition or by burrowers) as predicted by the obrution scenario and rather precipitated in near-surface sediment zones naturally inhabited by V. gibba."*

**When episodic burial occurs frequently, temporal accuracy of dating methods can affect this inference. However, this scenario with episodic burial would also propagate to a generally higher sedimentation rate in the upper parts of sediment cores at Po and Panzano where pyrite-lined valves are more frequent – but net sedimentation rates do not increase upwards.**

**We ultimately prefer the scenario without rapid burial (and we find it useful to attempt to use the alternative represented by low bioirrigation, even when conceptually these two scenarios are not mutually exclusive) because individual flood events occur at decadal scales (thus infrequent relative to ages of pyrite-lined valves that seem to form annually), their frequency did not change during the 20th century, and the mixing that still occurs would make the initial phase of rapid burial ultimately unimportant because shells that get reworked up would be oxidized. We have revised and expanded this part in the** ***Discussion as follows:*** *"Distinct layers deposited by major decadal floods preserved in cores at Po prodelta (Tesi et al., 2012; Tomašových et al., 2018) may have triggered episodic burial of benthic communities with V. gibba, but similar flood-event layers were not detected at Panzano. However, first, the major discharge events that lead to flood deposits occur at decadal scales at Po prodelta (seven events during the 20th century, Zanchettin et al., 2008), whereas age distributions of pyrite-lined valves indicate that pyrite linings form continuously at yearly scales. The frequency of flood events did not increase during the 20th century, in contrast to the increasing frequency of pyrite-lined valves in the upper parts of cores at Po and Panzano. Second, pyrite-lined valves younger than 10 years old occur at high abundance in the uppermost 5-10 cm of the mixed layer close to the sediment-water interface, indicating that they do not represent transient valves that were just recently exhumed from deeper zones. Third, equally-old valves with and without pyrite linings occur at similar depths and their age*

*distributions at Po and Panzano are similar (Fig. 9A-F), indicating that valves with and without pyrite were buried below the mixed layer at a similar rate."*

559 - this observation (of loss of Fe from sediments under persistent anoxia) is very important, as it may help explain the absence of pyritized fossils in truly black, laminated sediments that we have documented repeatedly (see Brett, C.E., Dick, V.B. and Baird, G.C.,1991. Comparative taphonomy and paleoecology of Middle Devonian dark gray and black shale facies from western New York. In Landing, E. and Brett, C.E., eds., Dynamic Stratigraphy and Depositional Environments of the Hamilton Group in New York Pt. II. State Museum Bulletin 469, p. 5â€'36.) One implication is that, while eutrophication and increased organic matter may enhance pyrite formation up to a point (because of lower benthic $O_2$ and reduced bioirrigation); too much OM may shut down the process.

**Response: We note that another subset of this dynamic is that mixing of particles by burrowers (i.e., the subset of bioturbation that does not necessarily irrigate the sediment) enhances the development of the suboxic iron-rich zone. To follow this comment, we have expanded the Discussion in 5.2 as follows: At the end of section 5.2., we have revised our discussion of anoxic scenario, as follows:** *"If the nutrient-fueled eutrophication or other sources of sediment organic enrichment lead to permanent anoxia of bottom waters, the concentrations of pyrite framboids within shells can be prohibited because mixing of sediments by burrowers that underlies the iron-based redox shuttle is aborted and sulfide production by bacterial sulfate reduction in organic-rich sediments can exceed availability of reactive iron oxides, with $H_2S$ diffusing away (Raiswell and Berner, 1985; Schenau et al., 2002; Raiswell et al., 2008; Farrell et al., 2009), or ferrous iron can be released from sediments to the water column if the water column is not sulfidic (Pakhomova et al., 2007). The absence of pyrite linings in anoxic sediments documented in the deep-time stratigraphic record (e.g., Brett et al., 1997) indicates that in marine environments with persistent bottom-water anoxia, the window for the early and rapid formation of pyrite linings (e.g., within shells of nektonic groups such as cephalopods that fell on the anoxic seafloor) will be closed when (a) the suboxic zone is not induced by biomixing and (b) organic matter degradation in surface sediments leads to the excess of hydrogen sulfide (Middelburg and Levin, 2009)."*

570: forgive my ignorance, but I do not understand your use of **hysteresis** here. Perhaps explain a bit.
**Response: Hysteresis corresponds to non-linear ecosystem dynamic that generates alternative stable states under the same environmental conditions. We have added parenthetically into the Discussion: "***...If subsequent recovery of bioirrigation-inducing burrowers is slow or interrupted by another hypoxic event or if communities with infrequent bioirrigators are locked by hysteresis effects in an alternative stable state even when hypoxic conditions abated (Kemp et al., 2009; Duarte et al., 2015),..."*

577: it is not intuitive that large size would be associated with lowered oxygen levels (resulting from eutrophication); in many cases low oxygen has been attributed to stunting and diminutive organisms. Apparently in Varicorbula the stunting affect is offset by increased growth. Presumably this is well documented; a reference might be useful.
**Response: The increase in size reflects the indirect effect of "ecological release" from predation and competition coupled with higher tolerance of V. gibba to short-term hypoxic or anoxic events (in contrast to smaller tolerance of potential predators or**

**competitors). V. gibba is also negatively affected by anoxia, but this species is able to grow to larger sizes in the wake of those events after they are over, in contrast to conditions without any hypoxic events. We have added in the Methods: "***In the northern Adriatic Sea, it increases in abundance in the wake of short-term, seasonal anoxic or hypoxic events to more than 1,000 individuals/m$^2$ (Hrs-Brenko 2006, Nerlović et al. 2011), and grows to 7-8 mm during the first year, achieving the maximum size of ~15 mm in two years (Hrs-Brenko 2003).*"

613: but why do these bivalves not open shortly after death? If they are not buried rapidly (episodically) then how do they come to remain tightly closed?
**Response: Yes, shells buried themselves during their life, but we refer to the absence of additional postmortem burial. We assume that these shells were located in the uppermost sediment zones in the upper 3 cm after their death, i.e., at their living locations. *V. gibba* possesses a relatively small internal ligament that, as in other members of the superfamily Myoidea, generates a small opening moment that is not sufficient to open valves against the sediment pressure (Trueman, 1954, Yonge, 1982).**

Line 635: this also contrasts with nanopyrite infillings that seem to increase together with other signs of degradation with residence time.
**Response: yes, we have modified this sentence to**:*"Pyrite-lined valves thus represent a unique type of alteration that contrasts with other types of alteration (including the frequency of stained valves with nanopyritic inclusions) whose incidence increases with residence time in the taphonomic active zone"*

**Illustrations** are fine and require no modification.

**References** I have gone through and highlighted all that appear to be correctly cited; most are fine but there are a number of minor problems:
The following are **slightly out of alphabetical order:** Briggs et al. 1991., Boyle, Bjereskov, Degobbis. Meysman, Schieber 2012 (should come before Schieber and Baird)
**Response: Fixed**

**Wrong date**
Brush et al. 2020 (2021 intext) **Fixed to 2021**
Kralje et al. 2019 (2020 in text) **Fixed to 2019**
Slagter et al. 2021 (2020 intext) **Fixed to 2021**
**No date** (in ref list) Hunda et al. [2006] **Fixed**

**In reference list but apparently not referenced in text**:
Aller, 1982 **Added to main text**
Arcon et al. 1999 **Added to main text**
Clark ad Lutz, 1980 **Removed**
Eliott et al. 2007 **Added to main text as Elliott**
Faganeli et al 1985 **Added to main text**
Palinkas et al. 2007 **Added to main text as Palinkas and Nittrouer 2007**
Powell and Stanton 1985 **Removed**

**Cited in text but not in reference list:**
Alvizi et al. 2016 **Added to main text**

Stanton and Powell, 1985 **Removed from the main text**
Wignall et al., 2010 **Added to main text as Bond and Wignall 2010**

**Need to designate a and b references in text:**
Tomašovych 2019 a vs. b
**Response: Fixed**

**Overall:** this is an excellent paper, which needs only minor revision for publication. The paper provides much new data and reaches important conclusions regarding the little known phenomenon of early diagenetic pyritization; it has important implications for sedimentation rates and rates of bioirrigation, for the taphonomy of exceptional fossils, and potentially for conservation paleobiology and evidence for anthropogenic effects. The results will be of interest to geochemists, sedimentologists, taphonomists, paleobiologists and perhaps conservation paleobiologists. I strongly recommend publication with slight correction.